# Frost flowers and sea-salt aerosols over seasonal sea-ice areas in north-western Greenland during winter–spring

Keiichiro Hara[1], Sumito Matoba[2], Motohiro Hirabayashi[3], and Tetsuhide Yamasaki[4]

[1] Department of Earth System Science, Faculty of Science, Fukuoka University, Japan
[2] Institute of Low Temperature Science, Hokkaido University, Japan
[3] National Institute of Polar Research, Tokyo, Japan
[4] Avangnaq, Osaka, Japan

Correspondence to: K. Hara (harakei@fukuoka-u.ac.jp)

**Abstract.** Sea-salts and halogens in aerosols, frost flowers, and brines play an important role in atmospheric chemistry in polar regions. Simultaneous sampling and observations of frost flowers, brine, and aerosol particles were conducted around Siorapaluk in northwestern Greenland during December 2013 – March 2014. Results show that water-soluble frost flower and brine constituents are sea salt constituents (e.g., $Na^+$, $Cl^-$, $Mg^{2+}$, $K^+$, $Ca^{2+}$, $Br^-$, and iodine). Concentration factors of sea-salt constituents of frost flowers and brine relative to seawater were 1.14–3.67. Sea-salt enrichment of $Mg^{2+}$, $K^+$, $Ca^{2+}$, and halogens ($Cl^-$, $Br^-$, and iodine) in frost flowers is associated with sea-salt fractionation by precipitation of mirabilite and hydrohalite. High aerosol number concentrations correspond to the occurrence of higher abundance of sea-salt particles in both coarse and fine modes, and blowing snow and strong winds. Aerosol number concentrations, particularly in coarse mode, are increased considerably by release from the sea-ice surface under strong wind conditions. Sulfate depletion by sea-salt fractionation was found to be slight in sea-salt aerosols because of the presence of non-sea-salt $SO_4^{2-}$. However, coarse and fine sea-salt particles were found to be rich in Mg. Strong Mg enrichment might be more likely to proceed in fine sea-salt particles. Mg-rich sea-salt particles might be released from the surface of snow and slush layer (brine) on sea-ice and frost flowers. Mirabilite-like and ikaite-like particles were identified only in aerosol samples collected near new sea-ice areas. From the field evidence and results from earlier studies, we propose and describe sea-salt cycles in seasonal sea-ice areas.

## 1 Introduction

Frost flowers on sea-ice are ice crystals that contain brine and sea salts. They appear often during winter–spring on the surfaces of new and young sea ice in polar regions. Frost flowers play an important role as an interface among the atmosphere, sea ice, and ocean. Key conditions for formation and growth of frost flowers are (1) a cold atmosphere (below -15 °C) and (2) weak–calm winds (Perovich and Richter-Menge, 1994; Martin et al., 1995, 1996; Style and Worster, 2009; Roscoe et al., 2011). Vertical gradients of temperature and relative humidity near the sea-ice surface are crucially important for the appearance of frost flowers. Frost flowers can be formed by condensation of water vapor on nodules on new and young sea ice. Results of earlier studies (e.g., Domine et al., 2005; Douglas et al., 2012) show that water vapor is initially supplied to the atmosphere with sublimation or evaporation occurring from the warm sea-ice surface. Strong vertical gradients of air temperature above the sea-ice surface engender supersaturation of water vapor near the sea-ice surface; then the gradients induce condensation of water vapor (i.e., frost flower formation). The concentrated seawater (i.e. brine) is present on new and young sea ice. Consequently, brine with high salinity migrates upwardly and gradually on frost flowers (Perovich and Richter-Menge, 1994; Martin et al., 1996; Roscoe et al., 2011). Under cold conditions, solutes with lower solubility in brine can be precipitated in and on sea ice and frost flowers depending on the temperature. Earlier investigations (e.g., Marion, 1999; Koop et al., 2000; Diekmann et al., 2008, 2010; Geilfus et al., 2013) have revealed that several salts can be precipitated at -2.2 °C (ikaite, $CaCO_3 \bullet 2H_2O$), -8.2 °C (mirabilite, $Na_2SO_4 \bullet 10H_2O$), -15 °C (gypsum, $CaSO_4$), -22.9 °C (hydrohalite, $NaCl \bullet 2H_2O$), -28 °C ($NaBr \bullet 5H_2O$), -33 °C (sylvite, KCl), -36 °C ($MgCl_2 \bullet 12H_2O$), and -53.8 °C (antarcticite, $CaCl_2 \bullet 6H_2O$). The salt precipitation

occurring here at lower temperatures causes changes to sea-salt ratios in brine and frost flowers, known as sea-salt fractionation. Actually, early works showed that sea-salt ratios in frost flowers differed from those in brine and seawater (Rankin and Wolff, 2000; Rankin et al., 2002; Alvarez-Aviles et al., 2008; Douglas et al., 2012). It has been considered that salt precipitation proceed in brine, because sea-salt ratios cannot be changed in occurrence of sea-salt fractionation on frost flowers.

Frost flowers have a fine structure. Earlier field and laboratory experiments indicated frost flowers as less fragile, even under strong winds, in spite of their fine structure (Obbard et al., 2009; Roscoe et al., 2010). In addition, results of model studies have implied that blowing snow contributes importantly to atmospheric halogen chemistry (Yang et al., 2010; Abbatt et al., 2012; Lieb-Lappen and Obbard, 2015). Because of the lower number density of aerosol particles in polar regions, especially

in Antarctic regions, emission of sea-salt particles from sea-ice areas is an important aerosol source during winter–spring (e.g., Wagenbach et al., 1998; Rankin et al., 2000; Hara et al., 2004, 2011, 2012, 2013). Actually, sea-salt particles released from sea-ice areas are dispersed from the boundary layer to the free troposphere (up to ca. 4 km) over Syowa Station, Antarctica through vertical motion by cyclone activity (Hara et al., 2014), and into the interior (Dome F Station and Concordia Station) of the Antarctic continent (Hara et al., 2004; Udisti et al., 2012). Vertical transport of the sea-salt particles originating from

sea ice can act as a supply of cloud condensation nuclei (CCN) and ice nuclei (IN) in the upper boundary layer – free troposphere (Twohy and Poellet, 2005; Wise et al., 2012; DeMott et al., 2016). Because of the horizontal transport of sea-salt particles into the Antarctic plateau, $Na^+$ records in ice cores taken in the inland area have been used recently as a proxy of the sea-ice extent (e.g., Wolff et al., 2003, 2006).

As described above, sea-salt fractionation proceeds on new and young sea ice. For that reason, sea-salt ratios in sea-salt particles (or aerosols) released from sea-ice areas differ from those of the bulk seawater ratio (Hara et al., 2012, 2013). For instance, Mg-rich sea-salt particles and aerosol particles containing $MgCl_2$ and $MgSO_4$ were identified to a remarkable degree at Syowa Station during winter–spring because precipitation of mirabilite and hydrohalite engenders Mg-enrichment in sea-salt particles (Hara et al., 2012, 2013). Because the relative humidity of Mg-salt deliquescence, such as that of $MgCl_2$, is lower

than that of NaCl (e.g., Kelly and Wexler, 2005), sea-salt fractionation can engender modification of aerosol hygroscopicity, which is closely related to phase transformation, heterogeneous reactions, and abilities to act as cloud condensation nuclei and ice nuclei. According to results of laboratory experiments conducted by Koop et al. (2000), $Br^-$ can be enriched in frost flowers by sea-salt fractionation. Reportedly, $Br^-$ enrichment occurs slightly in frost flowers in the Weddell Sea, Antarctica (Rankin et al., 2002). Slight $Br^-$ enrichment to $Na^+$ has also been observed in a few samples collected at Barrow, Alaska, although no $Br^-$

enrichment was detected in samples of frost flowers and brine (Douglas et al., 2012). Additionally, results of some earlier studies have described non-significant $Br^-$ enrichment in frost flowers at Barrow and Hudson Bay (Alvarez-Aviles et al., 2008; Obbard et al., 2009). Therefore, many issues remain with respect to sea-salt and halogen chemistry of aerosols and frost flowers. They demand further measurement results and discussion of them. In addition to sea-salt fractionation, sea-salt ratios in frost flowers and aerosols can be altered gradually by heterogeneous reactions in a process known as sea-salt modification.

Furthermore, frost flowers have large specific surface area: 63–299 $cm^2$ $g^{-1}$ (mean, 162 $cm^2$ $g^{-1}$) at Hudson Bay, Canada (Obbard et al., 2009) and 185 (+80, -50) $cm^2$ $g^{-1}$ at Barrow, Alaska (Domine et al., 2005). Because of the larger surface area, earlier studies have assessed the potential of frost flowers for use as reaction sites (e.g., Kaleschke et al., 2004, references in Abbatt et al., 2012). Sea-salt modification in sea-salt aerosols, and sea salts in and on frost flowers and sea ice can act as potential sources of gaseous reactive species. For instance, gaseous reactive halogen species (e.g., $Br_2$, HOBr, Br, and BrO)

induce depletion of ozone and mercury near the surface in both polar regions during the polar sunrise (Barrie et al., 1988; Schroeder et al., 1998; Foster et al., 2001; Ebinghaus et al., 2002).

To elucidate the atmospheric impact of fractionated sea-salt particles and their relation between sea-salt particles in the atmosphere and frost flowers on sea ice, one must ascertain (1) the chemical properties (e.g., concentrations, ratios, and pH) of frost flowers and brine, and (2) the physical and chemical properties of aerosols (e.g., size distribution, constituents, and mixing states) above seasonal sea ice with frost flowers. Despite their importance, simultaneous observations and measurements of aerosols and frost flowers over seasonal ice areas with frost flower appearance have not been reported for polar regions, although sampling and observations of frost flowers have been conducted in the Arctic (e.g., Alvaraz-Aviles et al., 2008; Douglas et al., 2012) and Antarctica (Rankin and Wolff, 2000; Rankin et al., 2002). Using data from simultaneous measurements and sampling of aerosols, frost flowers, and brine around northwestern Greenland during winter–spring, this study was conducted to elucidate sea-salt cycles in seasonal sea-ice areas and their related phenomena, such as sea-salt fractionation on the sea-ice surface including frost flowers, brine and snow, their aging processes, and the release of fractionated sea-salt particles to the atmosphere.

## 2 Samples and Analysis

### 2.1 Sampling sites and conditions

Figure 1 shows the locations where simultaneous observations of frost flowers, brine, seawater, and aerosols were made on a new and young sea-ice area in Robertson fjord near Siorapaluk in northwestern Greenland from mid-December, 2013 through mid-March, 2014. In the fjord, the open sea surface appears during summer. Sea ice formed gradually from October, 2013 in the fjord. Sea ice flowed out several times from the fjord by the action of sea tides, heaving, and strong winds (Fig. 1) before and during the measurements taken for this study. Moreover, sea-ice with different ages was present in the fjord because locations of sea-ice breaks differed in each case, as presented in Fig. 1. For the present study, we define sea-ice ages in the fjord as new, young, old, and very old, depending on the sea-ice age. We chose Sites I–III (new – young sea ice; less than 1 cm – ca. 35 cm thickness) as sampling sites of aerosols, frost flowers, brine, and seawater. Site I was approximately 2 km distant from Siorapaluk. The sea ice on Site I flowed during 10–12 February, 2014 (shown as a red broken line); then it refroze. Several days to a week prior (approximately mid-February), an open-lead appeared around Site II. Thereafter, new sea ice was formed. The refrozen lead width was 2–6 m near sampling sites. The sea surface appeared off Siorapaluk on approximately 1 March by sea-ice breakage and strong winds (shown by the blue broken line). Then, new sea ice formed again (photographs, Fig. 1). On 3 and 5 March, we traversed along the new sea-ice area (blue broken line in Fig. 1) to observe the sea-ice conditions and the appearance of frost flowers on the sea-ice. Then, we chose sampling sites (IIIa and IIIb) that were safely accessible. The new sea ice at Sites IIIa and IIIb (less than 1 cm thickness) was a few days old. We accessed sampling sites with frost flowers on foot and by dog sledge from Siorapaluk.

### 2.2 Sampling of frost flowers, brine, snow, and seawater

Sampling of frost flowers, brine (slush), and seawater was conducted from 20 February, 2014 through 3 March, 2014 on sea ice near Siorapaluk. Frost flowers were taken from the sea-ice surface using a clean stainless steel shovel. During the campaign, snowfall and blowing snow occurred occasionally. On the frost flower and slush layer at Sites I and II, snow was present, slightly, to the extent to that the fine structure of frost flowers was identified clearly. All bodies of the frost flowers above the sea-ice surface were collected if the forms of frost flowers were matured and dry. The frost flowers were collected with the coating brine water if the form of frost flowers was young and coated with brine water. When snow was present on the frost flower and slush layer, some frost flower samples can contain slightly snowy pieces because of the difficulty of segregation. Brine samples were collected by shaving off a thin layer of the sea-ice surface and coating the brine water in proximity to the frost flowers. With the exception of new sea-ice at Site III with thickness of a few centimeters, slush was sampled on sea-ice where frost flowers were formed. Because it was difficult to collect only brine from the slush layer, the slush layer was sampled

as "brine samples" in this study. Snow on sea-ice was also taken using a clean stainless steel shovel from the location with snow accumulation (< 3 cm depth) without frost flowers at Sites I and II. Pieces of frost flowers, brine (slush) samples, and snow were moved into each polyethylene bag (Whirl-pak; Nasco). Using a dropper, seawater samples were collected in polypropylene bottles from a crack in the sea ice or a hole we made in the sea ice. All samples were melted at ambient temperature. The $H^+$ concentration (i.e., pH) of the liquid sample was measured using a portable pH meter (B-212; Horiba Instruments Ltd.). Then residue of the sample was transferred to polypropylene bottles. The samples in the bottles were kept in colder conditions below -20 °C in Greenland. Then, samples were unfrozen during transport (ca. 3 days) to our laboratory in Japan because of the difficulty of carrying frozen samples in an airplane. After the bottled liquid samples were transported to Japan, all were kept frozen in a cold room until chemical analyses were conducted.

## 2.3 Aerosol sampling and measurements

Aerosol measurements and direct sampling were conducted over seasonal sea ice around Siorapaluk, Greenland from 17 December, 2013 through 7 March, 2014. Aerosol number concentrations were measured at flow rate of 2.83 L min$^{-1}$ using a portable optical particle counter (OPC, KR12A; Rion Co. Ltd.). The measurable size range was $D_p$ >0.3, >0.5, >0.7, >1.0, >2.0, and >5.0 μm. The OPC packed in an insulator box was set at ca. 1 m above the sea-ice surface by a tripod. Aerosol number concentrations were recorded every 23–25 s, corresponding to 1 L of air volume, during aerosol direct sampling. Details of OPC measurements were presented by Hara et al. (2014).

Direct aerosol sampling was done using a two-stage aerosol impactor. Carbon-coated collodion thin films supported by a Ni micro-grid (square-300 mesh; Veco Co.) were used as sample substrates in this study. The cut-off diameters (aerodynamic diameter) of the impactor were 2.0 and 0.2 μm at a flow rate of ca. 1.2 L min$^{-1}$. The impactor was set at ca. 1 m above the seasonal sea-ice surface, similarly to OPC measurement. Direct aerosol sampling was conducted for 10–15 min depending on the aerosol number concentration. Aerosol samples were kept in polyethylene capsules immediately after aerosol measurements and sampling. The polyethylene capsules with aerosol samples were packed into polyethylene zipper bags. All bags with aerosol samples were put into an airtight box together with a desiccant (Nisso-Dry M; Nisso Fine Co., Ltd.) until analysis at room temperature to prevent humidification that can engender morphological change and efficient chemical reactions, as described by Hara et al. (2002, 2005, 2013, 2014). All aerosol samples described as a result of this study were analyzed and observed within one year after sampling.

Meteorological data (air temperature, relative humidity, air pressure, wind direction, and wind speed) were measured using an auto-weather station (HOBO U30-NRC Weather Station; Onset Computer Corp.), which was set on the coast near Siorapaluk and ca. 1 km away from the village. Meteorological data were recorded to the data logger of the auto-weather station with time resolution of 5 min. A thermorecorder (TR-7Wf; T and D Corp.) and thermosensor (TR1106; T and D Corp.) were used for measurements of the temperatures of seawater, slush layer (brine) on sea ice, base of frost flowers on the slush layer, and the atmosphere above the top of the frost flowers.

## 2.4 Sample analysis

### 2.4.1 Analysis of frost flower, brine, snow, and seawater

Re-frozen samples of frost flower, brine, snow, and seawater were melted at ambient temperature before chemical analyses. Concentrations of ion species ($Na^+$, $K^+$, $Mg^{2+}$, $Ca^{2+}$, $Cl^-$, $NO_3^-$ and $SO_4^{2-}$) in frost flowers, brine and seawater were measured using ion chromatography (ICS 2100; Thermo Fisher Scientific Inc.) after $10^3$-fold dilution by ultrapure water, whereas those in snow were determined without dilution. A guard column (IonPac CG12; Thermo Fisher Scientific Inc.), column (IonPac CS12; Thermo Fisher Scientific Inc.), and a 20 mM $CH_3SO_3H$ eluent for the cation measurement, and guard column (IonPac

AG14; Thermo Fisher Scientific Inc.), column (IonPac AS14; Thermo Fisher Scientific Inc.) and 3.5 mM NaOH eluent were used for anion measurements. Concentrations of $Br^-$ in frost flower and brine were measured using an ion chromatograph – mass spectrometer (IC-MS) after $10^6$-fold dilution using ultrapure water. Ionic contents in samples were separated using ion chromatography with a guard column (IonPac AG11-HC; Thermo Fisher Scientific Inc.), a column (IonPac AS11-HC; Thermo Fisher Scientific Inc.), and gradient KOH eluent (4–36 mM), and were injected into a mass spectrometer (6100 series single quadrupole LC/MS; Agilent Technologies Inc.). The detection limit of $Br^-$ in IC-MS was 0.9 ng $L^{-1}$. Iodine concentrations in frost flower and brine were measured after $10^3$-fold dilution using ultrapure water with an inductively coupled plasma-mass spectrometer (ICP-MS, 7700 series single quadrupole ICP-MS; Agilent Technologies Inc.). The RF power and flow of carrier Ar gas were, respectively, 1550 W and 0.80 L min$^{-1}$ in ICP-MS analysis. The detection limit of iodine used for this study was 17 ng $L^{-1}$. Samples of frost flowers and brine were injected into ICP-MS after melting. Because ICP-MS can provide only the elemental concentrations in samples, iodide ($I^-$) and iodate ($IO_3^-$) were not divided in this study. Details of IC-MS analytical procedures are described elsewhere by Hirabayashi et al. (in preparation for publication). We calculated concentration factors (CF-X) with molar concentrations of chemical species (X) in samples divided by the concentration of X in seawater taken at Siorapaluk to evaluate the concentration processes of the chemical species during the formation and aging of frost flowers. Analytical errors of respective analytical methods were estimated from the reproducibility of determination of standard solutions with concentrations similar to those of the field samples.

### 2.4.2 Analysis of individual aerosol particles

Individual aerosol particles on the sample substrate were observed and analyzed for this study using a scanning electron microscope equipped with an energy-dispersive X-ray spectrometer (SEM-EDX, Quanta FEG-200F, FEI, XL30; EDAX Inc.). The analytical conditions were 20 kV accelerating voltage and 30 s counting time. Details of analytical procedures were described by Hara et al. (2013, 2014). We analyzed 1261 particles in coarse mode (mean, 41 particles per sample) and 6337 particles in fine mode (mean, 192 particles per sample). In this study, most aerosol-sampled areas on the substrates were analyzed in coarse mode. Although we attempted to analyze as many coarse particles as possible, the lower aerosol number concentrations in coarse mode limit the number of the aerosol particles analyzed in this study.

## 3. Results

### 3.1 Meteorological conditions during the campaign

Figure 2 depicts time variations of air temperature, relative humidity, and wind speed during our measurement period. The air temperature was -34.2 – +1.8 °C. Colder conditions below -20 °C occurred on days of year (DOY) = 47.5–58.6 (February 17–28, 2014) during our intensive sampling and observations of aerosols and frost flowers. Because of an approaching cyclone, several strong wind events occurred during the campaign. Particularly strong winds on DOY = 39.7–41.2 (9–11 February) caused breaks and outflows of sea ice from the front of Siorapaluk (ca. 1 km distant). Then the seawater started refreezing immediately after recovery of the weather. Sea ice off Siorapaluk (ca. 5–6 km distant) broke and flowed out again off Siorapaluk by storm conditions on DOY = 58.9–60.3 (February 28 – March 1). Sea ice formed after recovery of the weather conditions. Frost flowers appeared on the new sea ice in both cases.

### 3.2 Conditions of frost flowers and sea-ice

Sea ice conditions where frost flowers formed can be categorized as three types for this study: young ice (Site I), younger ice in refrozen leads between young ice (Site II), and new sea ice (Sites IIIa and IIIb). Figure 3 depicts photographs of frost flowers observed at Site I, categorized as young ice (Fig. 3a) on 20 February. The sea-ice surface was partly covered with thin snow. Frost flowers were formed patchily on the ice surface with little or no snow cover. The sea-ice surface underneath frost flowers

was wetted by brine (sherbet-like i.e., slush layer). Figure 3b presents a close up photograph of frost flowers observed on 22 February at the same site as that observed on 20 February. Salt crystals deposited on the branches of frost flower crystals were identified. Frost flowers at Site I were covered completely with snow after the storm on 28 February – 1 March. Figure 3c shows frost flowers observed at Site II (Fig. 1). The sea-ice surface underneath frost flowers was wet by brine. Figure 3d shows frost flowers observed at Site IIIa (Fig. 1) on 4 March. The site was at the edge of sea ice, close to open water. The frost flower diameter was approximately 5 mm. The sea-ice surface was covered with brine water (Fig. 3e). The frost flower crystals were partially submerged in the brine water. During our campaign (December, 2013 – March, 2014), frost flowers were absent on old and very old sea-ice. Although snow covered dry surface of old and very old sea-ice patchily before the storm, bare sea-ice without a slush layer was apparent after the storm (Fig. 3f).

## 3.3 Concentrations of sea salts in frost flowers, brine, seawater, and snow on sea ice

Figure 4 depicts relations among the respective constituents of frost flowers, brine, snow, and seawater found in this study. Logarithmic plots of Fig. 4 are shown in Figure S1. Concentrations of $Na^+$ in frost flowers and brine were 48–154 mmol $L^{-1}$, which greatly exceeded the concentration of seawater collected at Siorapaluk ($Na^+$, 42.5 mmol $L^{-1}$). The ratios of $Cl^-/Na^+$ in seawater collected at Siorapaluk were similar to ratios reported in the literature (e.g., Lide, 2005; Millero et al., 2008), although the $SO_4^{2-}/Na^+$ ratio at Siorapaluk was similar to the ratio in Millero et al. (2008) and slightly higher than the ratios in Lide (2005), as shown in Figure S2. Moreover, the ratios of $K^+/Na^+$ and $Ca^{2+}/Na^+$ in seawater at Siorapaluk were slightly lower (ca. 20%) than those of the literature values. Differences among the ratios found at Siorapaluk and those of the literature (e.g., Lide, 2005; Millero et al., 2008) were larger than our analytical errors (less than 5–6%), as estimated by reproducibility of the determination of standard samples with concentrations similar to those of the analyzed samples of frost flowers, brines, and snow. In this study, seawater ratios at Siorapaluk were used in the following analysis and discussion except for $Br^-$ and I, which were referred from earlier reports (e.g., Lide, 2005; Millero et al., 2008; Millero, 2016).

The concentration factors of $Na^+$ ($CF_{Na}$) in frost flowers and brine were 1.14–3.67, which roughly approximated results reported from previous studies (e.g., Rankin et al., 2002; Alvarez-Aviles et al., 2008). By contrast, the $Na^+$ concentrations in snow samples collected on sea-ice were 1–2 orders lower than that of seawater. The $Na^+$ concentrations of fresh snow on sea-ice were lower than 0.1 mmol $L^{-1}$. By contrast, $Na^+$ concentrations were 0.4–3.2 mmol $L^{-1}$ in the aged snow on sea-ice. High correlation among constituents was identified in frost flowers, brine, and snow, as shown below.

Frost flowers:
$[Cl^-] = 1.302 [Na^+] + 2.158 (R^2 = 0.969)$
$[Br^-] = 0.0022 [Na^+] + 0.025 (R^2 = 0.910)$
$[I] = 1.17 \times 10^{-6} [Na^+] + 5.867 \times 10^{-4} (R^2 = 0.791)$
$[Mg^{2+}] = 0.122 [Na^+] – 0.761 (R^2 = 0.982)$
$[K^+] = 0.023 [Na^+] – 0.063 (R^2 = 0.988)$
$[Ca^{2+}] = 0.023 [Na^+] – 0.019 (R^2 = 0.996)$

Brine:
$[Cl^-] = 1.421 [Na^+] – 21.35 (R^2 = 0.964)$
$[Br^-] = 0.0020 [Na^+] + 0.06 (R^2 = 0.962)$
$[I] = 1.10 \times 10^{-6} [Na^+] + 3.25 \times 10^{-5} (R^2 = 0.842)$
$[Mg^{2+}] = 0.114 [Na^+] – 0.8267 (R^2 = 0.982)$
$[K^+] = 0.022 [Na^+] – 0.140 (R^2 = 0.988)$

$[Ca^{2+}] = 0.021\,[Na^+] - 0.023\ (R^2 = 0.972)$

Snow:

$[Cl^-] = 1.315\,[Na^+] + 0.02\ (R^2 = 0.972)$

$[Mg^{2+}] = 0.035\,[Na^+] + 0.02\ (R^2 = 0.751)$

$[K^+] = 0.024\,[Na^+] - 0.002\ (R^2 = 0.997)$

$[Ca^{2+}] = 0.026\,[Na^+] + 2 \times 10^{-5}\ (R^2 = 0.994)$

In this study, the $Br^-$ and $I$ concentrations in the snow samples were not determined. High coefficients of determination imply strongly that these constituents are sea salts (derived from seawater). When the intercepts are close to zero, the slopes in the relations might be close to the ambient molar ratios. Furthermore, the slopes of the relations can be biased positively in cases of contamination/mixing of non-sea-salt species such as minerals and anthropogenic species, which can be deposited onto surfaces of frost flowers, brines, and snows. In contrast, the ratios can be biased negatively in cases of sea-salt fractionation on sea-ice and depletion/release of the continents in frost flowers, brines, and snows into the atmosphere. The molar ratios in frost flowers, brines, and snow are presented in Table 1. With the exception of $Mg^{2+}/Na^+$ in snow and $I/Na^+$ in frost flowers and brine, the molar ratios conform to the slopes. The intercept values and the coefficients of determination in these ratios are, respectively, larger and smaller than the other ratios.

Higher slopes and molar ratios relative to $Na^+$ and considerable $SO_4^{2-}$ depletion were observed clearly in frost flowers (Fig. 4). It is expected that this $SO_4^{2-}$ depletion was caused by mirabilite precipitation. To evaluate the contribution of mirabilite precipitation to changes of the molar ratios in frost flowers, snow, and brine, sea-salt ratios relative to $Na^+$ after mirabilite precipitation were compared. Clear negative slopes (ca. -0.041) for molar concentrations were identified for the relation between $Na^+$ and nss-$SO_4^{2-}$ (not shown), as discussed by Wagenbach et al. (1998) and Hara et al. (2004). The negative slope found in this study resembles the slope for sea-salt aerosols in Antarctica (e.g., Wagenbach et al., 1998; Hara et al., 2004). Sea-salt ratios relative to $Na^+$ after mirabilite precipitation were estimated using the negative slope values (R = -0.041). Assuming that mirabilite precipitation occurred only in sea-salt fractionation, the amount of the depleted $SO_4^{2-}$ ($[SO_4^{2-}]_{SO4\text{-}depleted}$) and the concentration of $Na^+$ ($[Na^+]_{SO4\text{-}depleted}$) after mirabilite precipitation were calculated using the $Na^+$ concentration in seawater or brine ($[Na^+]_{seawater}$) and the following equations.

$$\left[SO_4^{2-}\right]_{SO4\text{-}depleted} = \left[Na^+\right]_{seawater} \times R \quad \cdots(eq1)$$

$$\left[Na^+\right]_{SO4\text{-}depleted} = \left[Na^+\right]_{seawater} - 2 \times \left[SO_4^{2-}\right]_{SO4\text{-}depleted} \quad \cdots(eq2)$$

Then, the sea-salt ratios after mirabilite precipitation were estimated as follows.

$$\left(\frac{[X]}{[Na^+]}\right)_{SO4\text{-}depleted} = \frac{[X]_{seawater}}{[Na^+]_{SO4\text{-}depleted}} \quad \cdots(eq3)$$

In eq. 3, $[X]_{seawater}$ denotes sea-salt concentrations in seawater or brines other than $Na^+$ and $SO_4^{2-}$. The estimated ratios (i.e., $Cl^-/Na^+$, 1.33; $Mg^{2+}/Na^+$, 0.099; $K^+/Na^+$, 0.022; $Ca^{2+}/Na^+$, 0.021) were consistent with the ambient molar ratios in frost flowers and snow samples. This coincidence strongly suggests that mirabilite precipitation cause change of sea-salt ratios in frost flowers and brines.

In addition to mirabilite precipitation, hydrohalite precipitation can change the molar ratios in aerosols and frost flowers (e.g., Marion et al., 1999; Koop et al., 2000; Hara et al., 2012). Next, we attempted to compare the relations of respective sea salts to $Cl^-$ and $Mg^{2+}$ (Supplementary, Figures S3–S5) to understand sea-salt fractionation by precipitation of mirabilite and other salts, and distribution of the fractionated sea-salts on sea-ice. Relations among $Mg^{2+}$, $K^+$, $Ca^{2+}$, and $Cl^-$ in frost flowers well matched those in brine (Figures S3–S4). The molar ratios to $Cl^-$ were not changed by mirabilite precipitation without $Cl^-$ loss by heterogeneous reactions. The relations of sea-salts implies that mirabilite precipitation made important contribution to change of sea-salt ratios in frost flowers. Furthermore, student t-tests were applied to assess differences in the molar ratios among frost flowers, brines, and snow (Table 2). The molar ratios relative to $Na^+$ in frost flowers were significantly higher than those in brines (p<0.01), although $K^+/Cl^-$, $Mg^{2+}/Cl^-$, $Br^-/Cl^-$, and $I^-/Cl^-$ were not significantly different (p>0.1). In comparison between frost flowers and snow, the ratios of $K^+$ and $Ca^{2+}$ to $Na^+$ and $Cl^-$ in frost flowers were significantly lower than those in snow (p<0.01). However, the ratios of $Mg^{2+}/Na^+$ and $Mg^{2+}/Cl^-$ in frost flowers were significantly higher than those in snow (p<0.01). Differences between frost flowers and snow for $Cl^-/Na^+$ (or $Na^+/Cl^-$), $SO_4^{2-}/Na^+$, and $SO_4^{2-}/Cl^-$ were not significant (p>0.1). Although $K^+/Na^+$ and $Ca^{2+}/Na^+$ in snow were higher than those in brine, $Mg^{2+}/Na^+$ and $Mg^{2+}/Cl^-$ in snow were significantly lower than those in brine (p<0.01). The ratios of $SO_4^{2-}$ to $Na^+$ and $Cl^-$ in snow were slightly but significantly lower than those in brine (0.05 < p < 0.1). Between snow and brines, the ratios of $Cl^-/Na^+$ (or $Na^+/Cl^-$) were not significantly different (p>0.1). The statistical analysis indicated that distribution of the fractionated sea-salts was highly heterogeneous in frost flowers, snow and brines.

Figure 4 and Table 1 show that the molar ratios in frost flowers, brine, and snow differed among sampling sites, circumstances such as temperature, and the age of frost flowers (details are discussed later herein). The molar ratios in frost flowers resembled the ratios found by previous studies of frost flowers and aerosols (Rankin et al., 2002; Douglass et al., 2012; Hara et al., 2012). The $Br^-/Na^+$ ratios found from previous investigations, however, differed considerably among frost flower sampling sites (Rankin et al., 2002; Alvarez-Aviles et al., 2008; Douglass et al., 2012). $Br^-$ enrichment in frost flowers was observed in this study and previous studies conducted in the Weddell Sea, Antarctica by Rankin et al. (2002), although Alvarez-Aviles et al. (2008) and Obbard et al. (2009) reported that $Br^-$ was not enriched in frost flowers (similar to the seawater ratio) at Barrow, Alaska or at Hudson Bay, Canada. Furthermore, the slope of $Mg^{2+}$-$Na^+$ in surface snow on sea-ice was lower than the seawater ratio, although fresh snow samples with the $Na^+$ concentration lower than 0.1 mmol $L^{-1}$ were distributed on the seawater ratios.

### 3.4 Aging of frost flowers and brine on sea ice

Figure 5 presents variations of $CF_{Na}$ and molar ratio of $SO_4^{2-}/Cl^-$ in frost flowers at Site I. As described above, aged frost flowers, young frost flowers, and fresh frost flowers were collected respectively at Site I, Site II, Site IIIa, and Site IIIb. The values of $CF_{Na}$ of all samples exceeded 1.0, even on new ice at Sites IIIa and IIIb, indicating that concentrated brine was excluded from sea ice to the sea-ice surface during sea-ice formation before the frost flower formation (Fig. 5a). Some samples with low $CF_{Na}$ at Sites I and II can be contaminated with slight snowfall. The $CF_{Na}$ of frost flowers at Sites IIIa and IIIb was lower than that at Sites I and II, which suggests that sea-salt concentrations and sea-salt ratios of frost flowers varied depending on the age of frost flowers and sea-ice.

To ascertain the changes of sea-salt constituents that occur along with the aging of frost flowers, we attempted to monitor the sea-salt constituents of frost flowers and brines at Site I on 20–28 February 2014. Figure 6 depicts short-term variations of air temperatures measured by AWS ($T_{AWS}$), air temperature above frost flowers ($T_{air}$, ca. 10 cm above the brine/sea-ice surface), temperature at bases of frost flowers ($T_{FF}$), and molar ratios of sea salts ($SO_4^{2-}/Na^+$, $SO_4^{2-}/Cl^-$, $Br^-/Cl^-$, $I^-/Cl^-$, $Mg^{2+}/Cl^-$, $K^+/Cl^-$, and $Ca^{2+}/Cl^-$) in frost flowers and brine. Unfortunately, we did not measure $T_{air}$ and $T_{FF}$ on 20–22 February. $T_{AWS}$ during 20–

28 February was lower than the temperature for mirabilite formation (ca. -9 °C). In addition, $T_{FF}$ was -18.9 – -21.3 °C on 24–28 February. Figure 6a shows that $T_{AWS}$ and $T_{air}$ were lower than -25 °C from 23 February. Sea-salt ratios in brines at Site I were distributed around with seawater ratios. In contrast to the ratios in brines, molar ratios of $SO_4^{2-}/Na^+$ in frost flowers decreased considerably relative to seawater ratios. Although the molar ratios of $SO_4^{2-}/Na^+$ in frost flowers did not change greatly on 26 February, other sea-salt ratios ($Mg^{2+}/Cl^-$, $K^+/Cl^-$, and $Ca^{2+}/Cl^-$) increased simultaneously. In addition, $Na^+/Cl^-$ ratios in frost flowers dropped slightly on 26 February. Molar ratios of $Br^-/Cl^-$ and $I^-/Cl^-$ in frost flowers were higher than the seawater ratio (values in literature) and brine (Figs. 6d–6e). As described above, $Br^-$ and $I^-$ were enriched in frost flowers at Site I. Although ratios of $Mg^{2+}/Cl^-$, $K^+/Cl^-$, $Ca^{2+}/Cl^-$, and $Br^-/Cl^-$ increased on 26–28 February, increases in the ratio of $I^-/Cl^-$ were not clear.

Although the molar ratios of $Mg^{2+}/Cl^-$, $K^+/Cl^-$, and $Ca^{2+}/Cl^-$ cannot change by mirabilite precipitation, these ratios increased on 26 February when $T_{FF}$ dropped approximately to the temperature for hydrohalite precipitation. Because of analytical errors 2–4 times smaller than the difference, it is expected that hydrohalite precipitation results in changes of the molar ratios. To ascertain the contribution of hydrohalite precipitation, we attempted to estimate the sea-salt ratios using the following assumption.

(1) Only mirabilite precipitation occurred on 20–24 February.
(2) Hydrohalite precipitation did not proceed on 20–24 February.
(3) Hydrohalite precipitation started from 26 February.
(4) Sea-salts other than mirabilite and hydrohalite were not fractionated on 20–28 February.

Assuming the occurrence of mirabilite precipitation on 20–24 February, sea-salt ratios in the residual brines are given as shown in eq. 3. Under this assumption, $Mg^{2+}$ was not precipitated. Therefore, $Mg^{2+}/Cl^-$ ratios did not change in mirabilite precipitation, as follows.

$$\left(\frac{[Mg^{2+}]}{[Cl^-]}\right)_{Seawater} = \left(\frac{[Mg^{2+}]}{[Cl^-]}\right)_{SO4\text{-}depleted} \quad \cdots(eq4)$$

Here, $([Mg^{2+}]/[Cl^-])_{seawater}$ and $([Mg^{2+}]/[Cl^-])_{SO4\text{-}depleted}$ respectively denote the molar ratios of seawater collected at Siorapaluk and the residual brine after mirabilite precipitation. When hydrohalite precipitation occurred, the $Cl^-$ concentrations in the residual brine ($[Cl^-]_{hydrohalite\text{-}depleted}$) decreased gradually with hydrohalite precipitation as follows.

$$[Cl^-]_{hydrohalite\text{-}depleted} = [Cl^-]_{seawater} - [Cl^-]_{hydrohalite} \quad \cdots(eq5)$$

Here, $[Cl^-]_{hydrohalite}$ and $[Cl^-]_{seawater}$ respectively stand for the amount of $Cl^-$ in hydrohalite and the $Cl^-$ concentration in seawater or brine. Therefore, $Mg^{2+}/Cl^-$ ratios in the residual brine after hydrohalite precipitation $(([Mg^{2+}]/[Cl^-])_{hydrohalite\text{-}depleted})$ are given as shown below.

$$\left(\frac{[Mg^{2+}]}{[Cl^-]}\right)_{hydrohalite\text{-}depleted} = \frac{[Mg^{2+}]_{seawater}}{[Cl^-]_{hydrohalite\text{-}depleted}} \quad \cdots(eq6)$$

$[Mg^{2+}]_{seawater}$ represents the $Mg^{2+}$ concentration in seawater or brine. Actully, $[Cl^-]_{hydrohalite}$ in eq. 6 can be estimated by substitution of the ambient $Mg^{2+}/Cl^-$ ratio in frost flowers on 28 February $(Mg^{2+}/Cl^- \approx 0.103)$ to $[Mg^{2+}]/[Cl^-])_{hydrohalite-depleted}$. Then, the other sea-salt ratios after hydrohalite precipitation $(([X]/[Cl^-])_{hydrohalite-depleted})$ can be given as the equation below.

$$\left(\frac{[X]}{[Cl^-]}\right)_{hydrohalite-depleted} = \frac{[X]_{seawater}}{[Cl^-]_{hydrohalite-depleted}} \quad \cdots(eq7)$$

The amount of $Na^+$ in hydrohalite $([Na^+]_{hydrohalite})$ was the same as that in $[Cl^-]_{hydrohalite}$. Therefore, the other sea-salt ratios relative to $Na^+$ after hydrohalite precipitation $(([X]/[Na^+])_{hydrohalite-depleted})$ can be estimated using the same procedure. As presented in Figure 6, the ratios of $Na^+/Cl^-$ in frost flowers were close to the estimated ratios in hydrohalite precipitation.

### 3.5 Morphology of sea-salt particles

Figure 7 depicts SEM images of aerosol particles collected above the sea-ice area with frost flowers. Most coarse aerosol particles collected on 1 March 2014 had structures with cuboid-crystal like materials (bright color in SEM image) and non-crystal materials (gray – dark gray color in SEM image) around the cuboid-crystal like materials. Furthermore, most aerosol particles had stains around the particles. The presence of stains is direct evidence that the aerosol particles had a liquid surface in the atmosphere. Na, Cl, and Mg were detected in aerosol particles of this type. Therefore, the particles might be identified as sea-salt particles. Strong peaks of Na and Cl were identified from cuboid-crystal like materials, whereas strong peaks of minor sea salts such as Mg, K, and S were also obtained from non-crystal materials. Depending on the amount (mass) of sea salts and water in a sea-salt particle, salts with lower solubility can exist in a state with a solid core. In SEM observations, however, aerosol particles were exposed to high-vacuum conditions. They dried up. The localization of each sea salt in a particle might proceed in a high-vacuum chamber. Therefore, it is noteworthy that the salt distribution in an SEM image differed from the state in the ambient atmosphere.

### 3.6 Elemental compositions of sea salts and relating salts in each aerosol particle

Figure 8 depicts EDX spectra of sea-salt particles and sea-salt-related particles in aerosol particles collected over the sea ice. In accordance with procedures of aerosol classification presented by Hara et al. (2013, 2014) and atomic ratios of the respective particles, the mixing states of sea-salt particles and related salt particles were classified into the following types: (1) sea-salt particles having atomic ratios similar to those of seawater (Fig. 8a); (2) Mg-rich sea-salt particles (Fig. 8b); (3) K-rich sea-salt particles (Fig. 8c); (4) modified sea-salt particles with a slight Cl-loss (Fig. 8d); (5) wholly modified sea-salt particles (Fig. 8e); (6) sea-salt particles internally mixed with mineral elements such as Al and Si (Fig. 8f); (7) $Na_2SO_4$ particles without Mg (Fig. 8g, and Fig. S6); (8) $MgCl_2$ particles (Fig. 8h); (9) $MgSO_4$ particles (Fig. 8i); and (10) KCl particles (Fig. 8j). Figure S7 shows that Mg in sea-salt particles might be present as $MgCl_2$. Actually, Mg and S were detected from aerosol particles in Fig. 8i. Atomic ratios of Mg and S of the aerosol particles containing only Mg and S were approximately compatible with $MgSO_4$. Actually, Mg-rich sea-salt particles, K-rich sea-salt particles, Mg-salt particles, and K-salt particles were identified also in the boundary layer over Syowa Station, Antarctica during winter–spring (Hara et al., 2013) and near the surface on the Antarctic plateau during summer (Hara et al., 2014). Additionally, aerosol particles with atomic ratios similar to $CaCO_3$ were observed in the same aerosol samples (shown in Supplementary Materials, Fig. S8). In this study, these particles were observed only in aerosol samples collected near new sea ice. In addition, aerosol particles containing sulfates, minerals, soot, and anthropogenic metals were observed in this study.

## 3.7 Abundance of sea-salt particles and sea-salt-related particles

For quantitative discussion, the relative abundance was estimated from the results of EDX analyses (Fig. 9). In this study, sea-salt and modified sea-salt particles were major components of the coarse mode. High abundance of sea-salt particles corresponded to strong winds, high aerosol number concentrations, and appearance of low clouds (fog) above open sea off Siorapaluk. Although a few samples showed low relative abundance of sea-salt particles and modified sea-salt particles, this result corresponded to high relative abundance of minerals and minerals containing sulfates. However, it is noteworthy that the low number concentrations in coarse mode in the atmosphere engender low number density on the sample substrates and engender high uncertainty. For number concentrations of less than 10 L$^{-1}$ in $D_p$> 2.0 μm, the number of the analyzed particles in coarse mode was several to approximately 16 particles.

Aerosol particles containing sulfates were dominant in fine mode, although sulfate particles were observed rarely in coarse mode. The relative abundance of nss-sulfate particles in fine mode was roughly equivalent to that of the Arctic boundary layer around Svalbard (Hara et al., 2003). The relative abundance of sulfate particles in fine mode was higher under conditions with low wind speed and low aerosol number concentrations. The relative abundance of sea-salt particles and modified sea-salt particles in fine mode showed higher abundance than 40% under conditions with strong winds or high aerosol number concentrations in coarse mode.

## 3.8 Sea-salt fractionation of aerosol particles in coarse and fine modes

Figure 10 presents ternary plots of sea-salts (Na, Mg, and S) and Mg-rich sulfates in coarse and fine modes. Internal mixtures of sea salts and minerals were excluded from the ternary plots. The sum of the atomic ratios of Na, S and Mg in each sea-salt particle was not 100% in the most cases. Therefore, we converted the sum of the atomic ratios to 100% for ternary plots. Labels A, B, C, and D respectively denote the bulk seawater ratio, sea-salt particles (close to $MgCl_2$) in which chloride was completely displaced by sulfate, sea-salt particles in which Na was completely replaced by Mg, sea-salt particles (close to $MgSO_4$) in which chloride and Na were completely displaced by sulfate and Mg, respectively. When the sea-salt particles are modified by sulfates and are not fractionated, they are distributed around the stoichiometric line of A–B. Mg in sea-salt particles can be enriched gradually with sea-salt fractionation by precipitation of mirabilite ($Na_2SO_4$ 10$H_2O$) and hydrohalite (NaCl 2$H_2O$) (Hara et al., 2012). When sea-salt fractionation (replacement between Na and Mg) occurs without sea-salt modification by sulfate, sea-salt particles are distributed around the stoichiometric line of A–C. When sea-salt fractionation and sea-salt modification by sulfate occur stoichiometrically and simultaneously, sea-salt particles are distributed around the stoichiometric line of A–D. Na-Mg-S ratios of frost flowers and brine were distributed around bulk seawater ratio (a), although slight Mg-enrichment was recognized in this study.

The Mg ratios in coarse aerosol particles collected near new sea ice (Site IIIa) on 3 March were distributed mainly around the bulk seawater ratio and $NaSO_4$ ratio (Fig. 10a). Mg was enriched slightly in some sea-salt particles, even in coarse modes. In contrast to the sea-salt particles, Mg-free particles were distributed around $Na_2SO_4$ ratio, as depicted in Fig. 8g. As described above, these particles distributed around $Na_2SO_4$ ratio (B) might be mirabilite particles. In fine mode, most of the sea-salt particles were distributed between the stoichiometric lines between seawater ratio – $MgSO_4$ and seawater – $MgCl_2$. Compared to the Mg ratio in coarse mode, Mg enrichment was obtained remarkably in fine sea-salt particles.

Although most of the sea-salt particles in coarse mode were distributed around the bulk seawater ratio on 14 January and 21 February 2014 (Figs. 10b–10c), some coarse sea-salt particles had strong Mg-enrichment and were distributed around the stoichiometric line of seawater ratio – $MgCl_2$. A few particles showed atomic ratios that were roughly equal to that of $MgCl_2$.

Similarly to coarse sea-salt particles, Mg enrichment was also identified in fine mode on 14 January and 21 February 2014. However, Mg-rich sea-salt particles lay approximately midway between stoichiometric lines of seawater ratio-$MgCl_2$ and seawater ratio-$MgSO_4$. Moreover, $MgSO_4$ particles were identified occasionally in this study (e.g., 21 February 2014).

In contrast to sea-salt particles on 14 January, 21 February, and 3 March 2014, sea-salt particles in both modes were distributed mostly around the seawater ratio under storm conditions with blowing and drifting snow on 1 March 2014 (Fig. 10d), although some sea-salt particles in coarse and fine modes had slight Mg enrichment. Winds came not from young sea-ice area with frost flowers (Sites I and II) but from old and very old sea-ice areas.

### 3.9 Variations of the fractionated sea-salt particles during winter

Figure 11 presents variations of Mg/Na ratios in sea-salt particles and wind speed. Sea-salt particles internally mixed with mineral particles were excluded from Fig. 11 to avoid misunderstanding of the sea-salt chemistry. Mg/Na ratios were higher than the bulk seawater ratio (Mg/Na $\approx$ 0.09 in seawater at Siorapaluk, 0.11 in reports of seawater of the literature (Lide, 2005, Merrlet et al. 2008)) in coarse and fine sea-salt particles during measurements. Higher Mg/Na ratios and their large variation in sea-salt particles were observed in both coarse and fine modes under calm wind conditions. In conditions with blowing snow

or strong winds (>5 m s$^{-1}$), the Mg/Na ratios and their standard deviation tended to decrease in both modes (particularly in fine mode). For instance, median Mg/Na ratios in strong winds were ca. 0.18 in both modes on DOY = 40 (10 February), and ca. 0.16 in coarse mode and 0.22 in fine mode on DOY = 59 (1 March). Furthermore, Mg/Na ratios in coarse sea-salt particles increased gradually on DOY = 52–57 (22–27 February), when the air temperature was below -25 °C and the wind speed was lower than 4 m s$^{-1}$. After the storm on DOY = 59 (1 March), Mg/Na ratios of sea-salt particles were distributed around the

seawater ratio, although the ratios varied. In contrast to coarse sea-salt particles, Mg/Na ratios were higher in fine modes in this study. Similar tendencies were observed for aerosol particles over Syowa Station, coastal Antarctica (Hara et al., 2013), and on the Antarctic continent (Hara et al., 2014). This size dependence of Mg enrichment of sea-salt particles is important to elucidate the processes of sea-salt particle release from sea ice and frost flowers.

### 4. Discussion

### 4-1. Sea-salt fractionation on sea-ice

The molar ratios of Cl$^-$/Na$^+$ in seawater collected at Siorapaluk were roughly equal to the ratios reported in the literature (Lide, 2005, Merrlet et al. 2008), although slight differences were identified in other ratios such as K$^+$/Na$^+$ and Ca$^{2+}$/Na$^+$. Considering the reproducibility of determination of sea-salt constituents in our analytical procedures, the differences (more than 10%) between the seawater ratios collected at Siorapaluk and the literature values cannot be explained by analytical errors. According

to Millero (2016), seawater concentrations and seawater ratios were varied in each sampling site. Indeed, differences in the seawater ratios (lower K$^+$/Na$^+$ ratio) were observed at the Antarctic coasts (Hara et al., 2012 and references therein). Therefore, the difference might result from differences of seawater ratios in sampling location.

Coincidence of the ambient molar ratios in frost flowers and snow with the estimated molar ratios in mirabilite precipitation

implies that the molar ratios relative to Na$^+$ in frost flowers and snow were affected strongly by mirabilite precipitation. Salt precipitation in sea-salt fractionation depends on temperature (Marion et al., 1999; Koop et al., 2000). Therefore, we attempted to compare the relations of respective sea salts to Cl$^-$ and Mg$^{2+}$ (Supplementary, Figures S1 and S2) for identification of sea-salt fractionation other than mirabilite precipitation. The molar ratios to Cl$^-$ were not changed by mirabilite precipitation without Cl$^-$ loss by heterogeneous reactions. Relations among Mg$^{2+}$, K$^+$, Ca$^{2+}$, and Cl$^-$ in frost flowers well matched those in

brine. Indeed, t-test results (Table 2) suggest that molar ratios of Mg$^{2+}$, K$^+$, and Ca$^{2+}$ relative to Cl$^-$ in frost flowers were not significantly different from those in brine. Although frost flowers were more likely to be richer in Br$^-$ and I$^-$ than in Cl$^-$ and

$Mg^{2+}$ in most samples (Table 1, Figs. S3-S5), Student t-test results indicated no significant difference because of high ratios in a few brine samples. Considering that the first step of precipitation of Na salt is mirabilite precipitation approximately at -9 °C, the coincidence of the relations of among $Mg^{2+}$, $K^+$, $Ca^{2+}$, and $Cl^-$ suggests that enrichment of $Mg^{2+}$, $K^+$, and $Ca^{2+}$ in frost flowers was driven mainly by mirabilite precipitation. It is noteworthy that t-tests were applied overall to frost flowers and snow with different ages. Therefore, we must consider changes of molar ratios carefully if sea-salt fractionation other than mirabilite precipitation had proceeded with aging of frost flowers (details are discussed in the next section).

Considering the presence of $MgCl_2$ and $MgSO_4$ in aerosol particles, Mg salts might be present in frost flowers and the slush layer. According to earlier laboratory and model studies (e.g., Mairon et al., 1999), $MgCl_2$ $6H_2O$ and KCl (sylvite) can be precipitated respectively at approximately -36 °C and -34 °C. During the measurements, the minimum air temperature (-34.1 °C) and temperature at the surface of the slush layer ($T_{FF}$) were higher than the temperature at $MgCl_2$ $6H_2O$ precipitation. Therefore, $MgCl_2 \bullet 6H_2O$ precipitation might not have occurred during the measurements. In other words, $Mg^{2+}$ might be distributed in the residual brine on frost flowers and in the slush layer and snow.

## 4-2. Aging of frost flowers and sea-salt fractionation

Molar ratios of $SO_4^{2-}/Cl^-$ in frost flowers at Sites I and II (Fig. 5b) were considerably lower than the seawater ratio. This change of $SO_4^{2-}/Cl^-$ ratios might be attributed to sea-salt fractionation by sulfate depletion (i.e., mirabilite precipitation). By contrast, $SO_4^{2-}/Cl^-$ ratios at Site III were roughly equivalent to the seawater ratio. Considering direct evidence indicating that mirabilite-like particles were identified in aerosols only at Sites IIIa and IIIb, mirabilite might be precipitated in the early sea-ice stage. Details of mirabilite-like particles are discussed in section 4-3.

At Site I, sea-salt ratios changed gradually with growth and aging of frost flowers, as shown in Figure 6. As discussed in section 4-1, lower $SO_4^{2-}/Na^+$ ratios might result from the mirabilite precipitation. Moreover, the ratios of $Mg^{2+}/Cl^-$, $K^+/Cl^-$, and $Ca^{2+}/Cl^-$ increased simultaneously and $Na^+/Cl^-$ decreased slightly on 26–28 February when $T_{FF}$ dropped approximately to the temperature for hydrohalite precipitation (ca. -22 °C). This simultaneous change might not result from analytical errors because the differences were 2–3 times larger than analytical errors ($1\sigma$). The ambient molar ratios of $Na^+/Cl^-$ in frost flowers were similar to the estimated ratios in hydrohalite precipitation, although the ratios were slightly lower than the estimated ratios. $Ca^{2+}$ can be fractionated by precipitation of ikaite (-2.2 °C, Dieckmann et al., 2008, 2010) and gypsum (-15 °C, Marion et al., 1999). Therefore, lower ratios of $Ca^{2+}/Cl^-$ might result from precipitation of these salts. Presence of K-rich sea-salt particles and K-salt particles in sea-salt particles in the atmosphere implies that sea-salt fractionation with $K^+$ occurred in our measurements, although sylvite can be precipitated at -33 °C (Marion et al., 1999). The coincidence implies that the molar ratios in frost flowers on 26–28 February were attributed to hydrohalite precipitation. In contrast to $T_{FF}$, $T_{air}$ increased slightly on 26–27 February. Then it increased greatly on 28 February – 1 March. In spite of the slight increase of $T_{AWS}$ and $T_{air}$, $T_{FF}$ tended to decrease slightly during 24–27 February. This decrease of $T_{FF}$ might be attributed to reduction of heat conduction by sea-ice growth (larger thickness). Consequently, it is expected that the sea-ice thickness was a fundamentally important factor for sea-salt fractionation on sea ice, in addition to $T_{air}$.

It is noteworthy that molar ratios in frost flowers cannot change if sea-salt fractionations such as mirabilite and hydrohalite precipitation occur on frost flowers after brine migration onto frost flowers. Whether mirabilite and hydrohalite were precipitated on frost flowers or not, the total amount (mass) of precipitated salts and sea salts in residual brine did not change without liberation by heterogeneous reactions. $Mg^{2+}$, $K^+$, and $Ca^{2+}$ cannot be released into the atmosphere by heterogeneous reactions. Results show that $T_{FF}$ was lower than the temperature at precipitation of mirabilite and hydrohalite and higher than

those at precipitation of sylvite, $MgCl_2 \cdot 6H_2O$, and $NaBr \cdot 5H_2O$ (e.g., Marion, 1999; Koop et al., 2000). Therefore, Mg might be enriched in the residual brine. Precipitation of mirabilite and hydrohalite might be driven near the surface of brine on the sea ice, as suggested by (1) a change of the molar ratios in frost flowers, (2) non-significant change of those in brine, and (3) $T_{FF}$ close to ca. -21 °C. Then, the residual brine might be migrated vertically onto frost flowers. The ratios of $Br^-/Cl^-$ and $I^-/Cl^-$ in frost flowers were mostly higher than those in brines, except for a few brine samples. It is expected that $Br^-$ and $I^-$ were richer in frost flowers because of sea-salt fractionation. However, the concentrations of $Br^-$ and $I^-$ in seawater sampled at Siorapaluk were not determined in this study.

## 4-3. Fractionated sea-salt particles in the atmosphere

From single-particle analysis of aerosols, several types of salt particles, likely related to sea-salt fractionation, were identified in aerosols: (1) Mg-rich sea-salt particles, (2) $Na_2SO_4$ particles, (3) Mg-salt particles ($MgCl_2$ and $MgSO_4$), (4) K-rich sea-salt particles, (5) KCl particles, and (6) $CaCO_3$ particles. With sea-salt fractionation in frost flowers, brine, and surface snow on sea-ice, $Mg^{2+}$, $K^+$ and $Ca^{2+}$ were enriched in frost flowers and surface snow, except $Mg^{2+}$ in surface snow. Therefore, Mg-rich sea-salt particles (Fig. 8b), K-rich sea-salt particles (Fig. 8c), Mg-salt particles (Figs. 8h–8i), and K-salt particles (Fig. 8j) might originate from the sea-ice area and might be associated with sea-salt fractionation. As presented in Fig. 8g, Mg was not detected in aerosol particles containing Na and S. Mg ratios in coarse sea-salt particles usually exceed the detection limit of single particle analysis by EDX. Therefore, the aerosol particles might have an extremely low Mg ratio. The atomic ratios of Na and S of the Mg-poor particles imply strongly that the particles were in the form of $Na_2SO_4$. If the sea-salt particles were modified with $SO_4^{2-}$ by heterogeneous reactions (Supplementary information, Figs. S9–S10), then the modified sea-salt particles contained sea-salt Mg. Consequently, the presence of $Na_2SO_4$ particles cannot be explained by their release from the sea surface and then sea-salt modification. $Na_2SO_4$ particles were observed only at new sea-ice, Sites IIIa and IIIb where $CaCO_3$-like particles were also identified. In the early stage of sea-ice formation, ikaite ($CaCO_3$ $6H_2O$) and mirabilite ($Na_2SO_4$ $10H_2O$) can be precipitated respectively at -2 °C and -8.8 °C on/in sea ice (Dieckmann et al., 2008, 2010; Marion et al., 1999). Therefore, mirabilite-like and ikaite-like particles might be released to the atmosphere in the new sea-ice area.

Next, we specifically examined Mg/Na ratios in sea-salt particles to elucidate the sea-salt cycles in seasonal sea-ice areas. Mg-rich sea-salts and Mg-salts cannot be evaporated or vaporized under ambient conditions: these particles must be released through physical processes. Sea-salt fractionation can occur if sea-salt particles are fractured in the atmosphere. However, direct evidence of fracture of sea-salt particles in the atmosphere has not been obtained (Lewis and Schwartz, 2004). The following evidence is important to elucidate the origins of Mg-rich sea-salt particles and Mg-rich salt particles in the atmosphere: (1) the presence of highly Mg-rich particles (Mg-rich sea-salts, $MgCl_2$, and $MgSO_4$), (2) $T_{FF}$ lower than the temperature at precipitation of mirabilite and hydrohalite, (3) higher Mg/Na ratio in fine mode, (4) small variation of the Mg/Na ratio in strong winds and blowing snow, (5) high $Mg^{2+}/Na^+$ ratios in frost flowers, and (6) Mg depletion in the aged surface snow on sea-ice.

The Mg/Na ratios in sea-salt particles differed greatly depending on the sampling site and meteorological conditions (e.g., winds and temperature) as presented in Fig. 11. It is noteworthy that sea-salt particles at Sites IIIa and IIIb were distributed around seawater ratios since DOY = 60 (2 March). Therefore, most of the sea-salt particles except mirabilite-like and ikaite-like particles at Sites IIIa and IIIb might be released from the sea surface. Moreover, the presence of ikaite-like particles and mirabilite-like particles in the atmosphere implies that these particles were released into the atmosphere from the early stage of sea ice after precipitation of ikaite and mirabilite on sea-ice and frost flowers. In contrast to Mg/Na ratios in sea-salt particles at Sites IIIa and IIIb, higher Mg/Na ratios than seawater ratios were identified at Sites I and II. Higher Mg/Na ratios in sea-salt

particles suggest strongly that Mg-rich sea-salt particles in the atmosphere were supplied from the sea-ice area with sea-salt fractionation. As shown in Figure 4, $Mg^{2+}/Na^+$ ratios in frost flowers at Site I increased gradually on 20 February – 1 March under colder conditions. The correspondence between high Mg/Na ratios in coarse mode and the coldest conditions implies strongly that Mg/Na ratios in coarse sea-salt particles responded rapidly to sea-salt fractionation on sea ice and frost flowers.

Because of the aerosol number concentrations, relative abundance of sea-salt particles, and high Mg/Na ratios relative to seawater ratios in strong winds, the fractionated sea-salt particles might be dispersed from sea-ice to the atmosphere by strong winds. Although high aerosol number concentrations were observed occasionally at Siorapaluk under calm winds, the features might result from transport of (1) sea-salt particles released elsewhere by strong winds and (2) anthropogenic aerosols (i.e., sulfates and Arctic haze). Because of the high abundance of sea-salt particles, most cases of higher aerosol number concentrations in calm winds were likely associated with the release from sea-ice area and transport of sea-salt particles. Similar phenomena (aerosol enhancement) were identified also at the Antarctic coasts (Hara et al., 2010). As discussed above, Mg was likely richer in frost flowers and the residual brine on sea-ice. Earlier investigations (Obbard et al., 2009; Roscoe et al., 2011) revealed, however, that no aerosol particles were released by the breakage of frost flowers under strong winds. Considering the direct evidence of Mg depletion in aged surface snow on sea-ice, Mg-rich sea-salt particles and Mg-salt particles were likely released from surface snow mixed with the residual brine on sea-ice. The variations of Mg/Na ratios in sea-salt particles were smaller in both coarse and fine modes under storm conditions (DOY = 40 and 59), although the Mg/Na ratios were higher than the seawater ratio. Winds passed from the old and very-old sea-ice area to the sampling sites in the storm conditions. Consequently, Mg-rich sea-salt particles in the storms might be released also from the snow layer on old and very-old sea ice through erosion of snow by strong winds because the slush layer was absent on old and very-old sea ice. By contrast, Mg/Na ratios were varied largely under calm wind conditions. To explain the presence of highly Mg-rich sea-salt particles and Mg-rich salt particles, we inferred that these particles were released from the aged surface snow and the residual brine on slush layer and frost flowers through erosion of snow with the residual brine and splashing and shattering of the residual brine film. Higher Mg/Na ratios in fine sea-salt particles are eminently explainable if the processes proceeded on seasonal sea-ice areas. To elucidate these points, we must accumulate more information related to the salt distribution on and in frost flowers and sea ice at the nanometer–micrometer scale.

### 4-4. Sea-salt cycles in seasonal sea-ice area

From the evidence and results from this work and earlier works, we propose the following as hypotheses for the sea-salt fractionation processes on sea-ice and the release of sea-salt particles into the atmosphere (Fig. 12).

### Initial stage: open sea surface

Before sea-ice formation, sea-salt particles are released from the sea surface through bubble bursting and breaking waves (e.g., Lewis and Schwartz, 2004). Sea-salt ratios in the particles released by bubble bursting are similar to the seawater ratio (Keene et al., 2010).

### First stage: seawater freezing

Seawater starts freezing at lower air temperatures. In this stage, sea-ice was likely present in conditions of grease-ice, frazil ice, and sludge (Comiso and Steffen, 2001; Brandt et al., 2005; Comiso, 2010). Considering that sea-salt particles with ratios similar to seawater were found to be present only at Sites IIIa and IIIb, these particles must be released from the sea surface in the initial stage and first stage. Depending on the temperature at the sea-ice surface, ikaite can start precipitation at temperatures lower than -2 °C (Dieckmann et al., 2008, 2010).

**Second stage: sea-ice formation and sea-salt fractionation**

Then, the sea-surface was covered with thin sea-ice (i.e., nilas) at Sites IIIa and IIIb. The presence of sea ice prevents the release of sea-salt particles from the sea surface to the atmosphere. A strong vertical gradient of air temperature near the sea-ice surface might cause frost flower formation on sea-ice (e.g., Perovich and Richter-Menge, 1994; Martin et al., 1995, 1996; Style and Worster, 2009; Roscoe et al., 2011). Some brine can be migrated vertically on frost flowers. Cooling of surface of the frost flower and brine on sea-ice can engender precipitation of ikaite and mirabilite. The presence of ikaite-like particles and mirabilite-like particles in the atmosphere suggests that these particles are released into the atmosphere through physical processes. Mirabilite-like and ikaite-like particles were identified in aerosols collected only at Sites IIIa and IIIb. Therefore, these particles might be released from fresh sea-ice areas. However, specific release processes remain unclear.

**Third stage: frost flower growth and sea-salt fractionation**

With sea-ice growth, the temperature on sea-ice ($T_{FF}$) might decrease gradually by reduction of heat conduction from seawater to the sea-ice surface. Lower temperatures on and in the slush layer can induce sea-salt fractionation by precipitation of mirabilite and hydrohalite. Precipitation of mirabilite and hydrohalite can engender sea-salt enrichment (e.g., $Mg^{2+}$, $K^+$, and $Ca^{2+}$) in frost flowers and the residual brines. The residual brine having Mg enrichment is migrated vertically on frost flowers.

**Fourth stage: strong winds and snowfall on Frost flower**

Under conditions with strong winds, snowfall, and blowing snow, snow particles were attached on frost flowers and slush layers. As suggested by laboratory experiments (Roscoe et al., 2010), no aerosol particles are released from frost flowers. However, Mg-rich sea-salt particles and Mg-salts might be released from the slush layer and surface snow on sea-ice.

**Fifth stage: frost flower and slush layer covered with snow**

When snowfall and blowing snow are sufficient to cover frost flowers and the slush layer on new – young sea-ice, frost flowers and slush layer are buried completely in snow after the storm. After snow deposition onto new – young sea-ice, the residual brine with $Mg^{2+}$ enrichment might be migrated vertically and gradually into the snow layer. As a result, the snow layer on new – young sea-ice was wet, as observed in this study. Sea-salts in the migrated brine, frost flowers and snow can be redistributed through snow metamorphosis, although distributions of sea-salts might be heterogeneous in the snow layer. Mg-rich sea-salt particles and Mg-salt particles might be released from the surface snow on sea-ice because of slight loss of $Mg^{2+}$ in the aged surface snow on sea-ice as shown in Figs. S1 and S4. Therefore, we speculate that splash and erosion of the residual brine on snow and frost flowers by winds are plausible release processes of Mg-rich sea-salt particles and Mg-salt particles.

**Sixth stage: snow erosion by strong winds**

With sea-ice growth, snow and slush layers can be frozen gradually. Then, strong winds (i.e., storm condition) engender erosion of aged surface snow on sea-ice and release of Mg-rich sea-salt particles into the atmosphere. A dry and hard surface of sea ice appears after snow layers are removed from old and very-old sea-ice. Because of wet conditions in the snow and slush layer, a large amount of surface snow remained on the young sea-ice.

We observed frost flowers and aerosols on sea-ice in a fjord near Siorapaluk, Greenland. Therefore, we were able to compare respective sea-ice stages easily. If sea-ice areas having frost flowers are present in the locations affected strongly by winds, waves, tides, and ocean currents (e.g., Arctic Ocean and Antarctic coasts), then sea-ice can flow out and have many cracks, which can occur often in polar regions. Under such conditions, some stages in Fig. 12 might duplicate and proceed simultaneously.

## 5. Concluding remarks

Simultaneous sampling and observations of frost flowers, brines, and atmospheric aerosol particles were conducted around Siorapaluk, northwestern Greenland at the end of December, 2013 – early March 2014. We obtained the following direct evidence from our field observations: (1) presence of ikaite-like particles, mirabilite-like particles, and sea-salt particles with ratios similar to seawater in the atmosphere at new sea-ice areas (Sites IIIa and IIIb); (2) changes of sea-salt ratios of frost flowers by sea-salt fractionation such as mirabilite and hydrohalite precipitation at Site I; (3) sea-salt enrichment in frost flowers and surface snow; (4) presence of Mg-rich sea-salt particles and Mg-salt particles in the atmosphere; (5) higher Mg/Na ratio in aerosol particles in fine mode; (6) small variation of the Mg/Na ratio in aerosols under conditions with strong winds and blowing snow; and (7) Mg depletion in the aged surface snow on sea-ice. From this evidence and results from earlier investigations, we proposed sea-salt cycles in seasonal sea-ice area during the winter. At the moment, some processes are inferred from our field evidence. In addition to analysis of frost flowers, brines and snow, more field studies and laboratory experiments must be applied to aging frost flowers, sea-salt fractionation, and the release of fractionated sea-salts into the atmosphere, and halogen chemistry in polar regions.

### Acknowledgments

We thank Ikuo Oshima and Siorapaluk residents for extremely useful comments and help related to operations on sea ice, sea-ice conditions, and the appearance of frost flowers. This study was supported by a Grant-in-Aid for Challenging Exploratory Research (PI: K. Hara, No. 25550018).

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

**Table 1** Statics of molar ratios of sea-salts relative to Na$^+$ and Cl$^-$ concentrations in frost flower, brine and seawater collected in this study ("n" indicates sample number).

| | | K$^+$/Na$^+$ | Mg$^{2+}$/Na$^+$ | Ca$^{2+}$/Na$^+$ | Cl$^-$/Na$^+$ | SO$_4^{2-}$/Na$^+$ | Br$^-$/Na$^+$ | I/Na$^+$ | K$^+$/Cl | Mg$^{2+}$/Cl | Ca$^{2+}$/Cl | SO$_4^{2-}$/Cl$^-$ | Br$^-$/Cl$^-$ | I/Cl$^-$ |
|---|---|---|---|---|---|---|---|---|---|---|---|---|---|---|
| Frost flower n=23 | Ave | 0.022 | 0.113 | 0.023 | 1.326 | 0.029 | 0.0025 | $1.83\times10^{-6}$ | 0.017 | 0.086 | 0.017 | 0.022 | 0.0019 | $1.38\times10^{-6}$ |
| | STD | 0.001 | 0.008 | 0.001 | 0.052 | 0.020 | 0.0003 | $3.38\times10^{-7}$ | 0.000 | 0.004 | 0.000 | 0.016 | 0.0002 | $2.42\times10^{-7}$ |
| | median | 0.023 | 0.116 | 0.023 | 1.351 | 0.019 | 0.0025 | $1.83\times10^{-6}$ | 0.017 | 0.086 | 0.017 | 0.014 | 0.0019 | $1.35\times10^{-6}$ |
| | min | 0.020 | 0.091 | 0.022 | 1.217 | 0.008 | 0.0015 | $1.26\times10^{-6}$ | 0.016 | 0.074 | 0.017 | 0.006 | 0.0012 | $9.65\times10^{-7}$ |
| | max | 0.023 | 0.121 | 0.024 | 1.387 | 0.070 | 0.0031 | $2.95\times10^{-6}$ | 0.017 | 0.093 | 0.018 | 0.058 | 0.0023 | $2.20\times10^{-6}$ |
| Brine n=11 | Ave | 0.021 | 0.105 | 0.022 | 1.170 | 0.056 | 0.0020 | $1.46\times10^{-6}$ | 0.019 | 0.096 | 0.020 | 0.049 | 0.0019 | $1.35\times10^{-6}$ |
| | STD | 0.001 | 0.008 | 0.001 | 0.246 | 0.026 | 0.0002 | $2.47\times10^{-7}$ | 0.008 | 0.035 | 0.008 | 0.022 | 0.0008 | $5.77\times10^{-7}$ |
| | median | 0.021 | 0.109 | 0.022 | 1.230 | 0.056 | 0.0020 | $1.37\times10^{-6}$ | 0.017 | 0.085 | 0.017 | 0.049 | 0.0017 | $1.13\times10^{-6}$ |
| | min | 0.019 | 0.091 | 0.020 | 0.451 | 0.021 | 0.0018 | $1.21\times10^{-6}$ | 0.016 | 0.081 | 0.016 | 0.016 | 0.0014 | $9.56\times10^{-7}$ |
| | max | 0.022 | 0.115 | 0.024 | 1.346 | 0.101 | 0.0024 | $1.85\times10^{-6}$ | 0.044 | 0.202 | 0.045 | 0.089 | 0.0043 | $2.97\times10^{-6}$ |
| Snow n=15 | Ave | 0.023 | 0.076 | 0.026 | 1.321 | 0.037 | | | 0.018 | 0.058 | 0.020 | 0.030 | | |
| | STD | 0.001 | 0.030 | 0.002 | 0.188 | 0.029 | | | 0.002 | 0.024 | 0.002 | 0.026 | | |
| | median | 0.024 | 0.070 | 0.026 | 1.303 | 0.025 | | | 0.018 | 0.054 | 0.020 | 0.019 | | |
| | min | 0.020 | 0.032 | 0.021 | 1.065 | 0.010 | | | 0.012 | 0.024 | 0.014 | 0.007 | | |
| | max | 0.027 | 0.122 | 0.029 | 1.953 | 0.108 | | | 0.020 | 0.095 | 0.022 | 0.101 | | |
| Seawater n=2 | | 0.020 | 0.091 | 0.020 | 1.227 | 0.0613 | 0.0018 [a] | $2.76\times10^{-7}\sim$ $1.37\times10^{-6}$ [b] | 0.016 | 0.074 | 0.016 | 0.0500 | 0.0015 [a] | $2.32\times10^{-7}\sim$ $1.15\times10^{-6}$ [b] |

a) Molar ratios of Br$^-$ were listed using bulk seawater ratio (Lide, 2005).

b) Iodine (I$^-$ + IO$_3^-$) concentration in seawater was estimated from the concentrations of I$^-$ and IO$_3^-$ measured by Ito (1997, 1999), Hirooka et al., (2003), Ito et al. (2003), Chen et al. (2007), and Horikawa et al. (2016). The estimated iodine concentrations in seawater are 0.130 - 0.647 µmol L$^{-1}$. Upper and lower molar ratios of I/Na and I/Cl were calculated using the estimated iodine concentrations and the concentrations of N$^+$ and Cl$^-$ by Lide (2005).

Table 2. Results of t-test of the molar ratios of sea-salt constituents among frost flowers, brines and snow.

**Frost flowers – Brines***

| Ratios | $Cl^-/Na^+$ | $K^+/Na^+$ | $Mg^{2+}/Na^+$ | $Ca^{2+}/Na^+$ | $SO_4^{2-}/Na^+$ | $Br^-/Na^+$ | $I/Na^+$ |
|---|---|---|---|---|---|---|---|
| T value | 2.939 | 3.585 | 2.920 | 4.113 | -3.449 | 4.668 | 3.186 |
| P value | 0.006 | 0.001 | 0.006 | $2.54\times10^{-4}$ | $1.60\times10^{-3}$ | $5.20\times10^{-5}$ | 0.003 |
| Ratios | $Na^+/Cl^-$ | $K^+/Cl^-$ | $Mg^{2+}/Cl^-$ | $Ca^{2+}/Cl^-$ | $SO_4^{2-}/Cl^-$ | $Br^-/Cl^-$ | $I/Cl^-$ |
| T value | -3.925 | -1.501 | -1.470 | -1.515 | -4.035 | 0.079 | 0.176 |
| P value | 0.00045 | 0.143 | 0.151 | 0.139 | $3.18\times10^{-4}$ | 0.937 | 0.861 |

**Frost flowers – Snows****

| Ratios | $Cl^-/Na^+$ | $K^+/Na^+$ | $Mg^{2+}/Na^+$ | $Ca^{2+}/Na^+$ | $SO_4^{2-}/Na^+$ |
|---|---|---|---|---|---|
| T value | 0.114 | -2.981 | 5.835 | -6.698 | -1.083 |
| P value | 0.910 | 0.005 | $1.16\times10^{-6}$ | $8.20\times10^{-8}$ | 0.286 |
| Ratios | $Na^+/Cl^-$ | $K^+/Cl^-$ | $Mg^{2+}/Cl^-$ | $Ca^{2+}/Cl^-$ | $SO_4^{2-}/Cl^-$ |
| T value | -0.648 | -3.248 | 5.492 | -5.844 | -1.120 |
| P value | 0.521 | 0.003 | $3.32\times10^{-6}$ | $1.122\times10^{-6}$ | 0.270 |

**Snows – Brines*****

| Ratios | $Cl^-/Na^+$ | $K^+/Na^+$ | $Mg^{2+}/Na^+$ | $Ca^{2+}/Na^+$ | $SO_4^{2-}/Na^+$ |
|---|---|---|---|---|---|
| T value | 1.271 | 4.936 | -3.175 | 6.228 | -1.730 |
| P value | 0.217 | $4.88\times10^{-5}$ | 0.004 | $1.95\times10^{-6}$ | 0.096 |
| Ratios | $Na^+/Cl^-$ | $K^+/Cl^-$ | $Mg^{2+}/Cl^-$ | $Ca^{2+}/Cl^-$ | $SO_4^{2-}/Cl^-$ |
| T value | -1.332 | -0.621 | -3.328 | -0.079 | -1.962 |
| P value | 0.196 | 0.541 | 0.003 | 0.938 | 0.062 |

*) Degrees of freedom were 32.

**) Degrees of freedom were 36.

***) Degrees of freedom were 24.

5   Dark-grey and light-grey areas indicated "insignificant" and "slightly significant", respectively.

Figure captions

Figure 1 Locations of sampling and sea-ice conditions around Siorapaluk and photographs of new sea-ice conditions off Siorapaluk (taken from helicopter on 7 March, 2014). Black, red, and blue broken lines show locations of sea-ice break in November, 2013, and on 10-14 Feberuary, 2014, and 1 March, 2014, respectively. Marks of A, B, and C denote broad locations in maps and photographs.

Figure 2 Variation of air temperature, relative humidity, and wind speed at Siorapaluk.

Figure 3 Photographs of (a) condition of sea-ice and frost flowers on 20 February at Site I , (b) frost flowers at Site I on 22 February, (c) frost flowers at Site II, (d) sea-ice condition at Site IIIa, (e) frost flowers on 4 March at Site IIIa, and (f) condition of old sea-ice on 2 March immediately after the storm. Red dotted lines indicate approximate location of margin between old sea-ice and young sea-ice.

Figure 4 Relation among each constituent to $Na^+$ concentration of frost flowers, brine, and seawater taken in this study. Black and red lines respectively present regression lines of frost flowers and brine. Open black and red circles respectively present concentrations in frost flowers and brine. Filled blue stars represent the concentrations of seawater taken around Siorapaluk. Open blue stars represent the concentrations of seawater in the literature presented in Table 1. Error bars indicate standard deviation ($1\sigma$) of analytical errors.

Figure 5 Variations of (a) concentration factor of $Na^+$ ($CF_{Na}$) and (b) molar ratios of $SO_4^{2-}/Na^+$ in frost flowers at each sampling site. Red line indicates the bulk seawater ratio. Error bars indicate standard deviation ($1\sigma$) of analytical errors.

Figure 6 Short-term features of (a) air temperature measured by AWS ($T_{AWS}$), air temperature above frost flowers ($T_{air}$, ca. 10 cm above the sea-ice surface), temperature of base of frost flowers ($T_{FF}$), and (b–h) molar ratios of sea-salts in frost flowers and brine at Site I. $T_{air}$ and $T_{FF}$ were not measured on 20–22 February. Error bars indicate standard deviation ($1\sigma$) of analytical errors. Blue and green dash lines the molar ratios after precipitation of mirabilite and hydrohalite, respectively.

Figure 7 SEM image of aerosol particles collected on 1 March, 2014 above the sea-ice area with frost flowers.

Figure 8 EDX spectra of sea-salt particles and sea-salt relating particles collected over the sea-ice area. Asterisks denote background peaks derived from the sample substrate.

Figure 9 Variations of (a) wind speed, (b) aerosol number concentrations, relative abundance of each aerosol type in (c) coarse mode and (d) fine mode. Red + marks indicate the date when the low clouds (fogs) were identified off Siorapaluk.

Figure 10 Ternary plots of Na–Mg–S of sea-salt particles and the modified sea-salt particles collected over the sea-ice area. A, B, C, and D in this figure denote respectively ratios of seawater, fully modified with $SO_4^{2-}$, $MgCl_2$, and $MgSO_4$,. Red open circles and blue triangles respectively present ratios of particles in coarse and fine modes.

Figure 11 Variations of wind speed and atomic ratios of Mg/Na in sea-salt particles and the modified sea-salt particles collected over the sea-ice area. In box plots, the top bar, top box line, black middle box line, bottom box line, and bottom bar respectively denote values of 90, 75, 50 (median), 25, and 10%. The red and lines show mean values and seawater ratio of Mg/Na.

5    Figure 12 Schematics of sea-salt cycles in sea-ice area. Dotted arrows indicate the speculated processes.

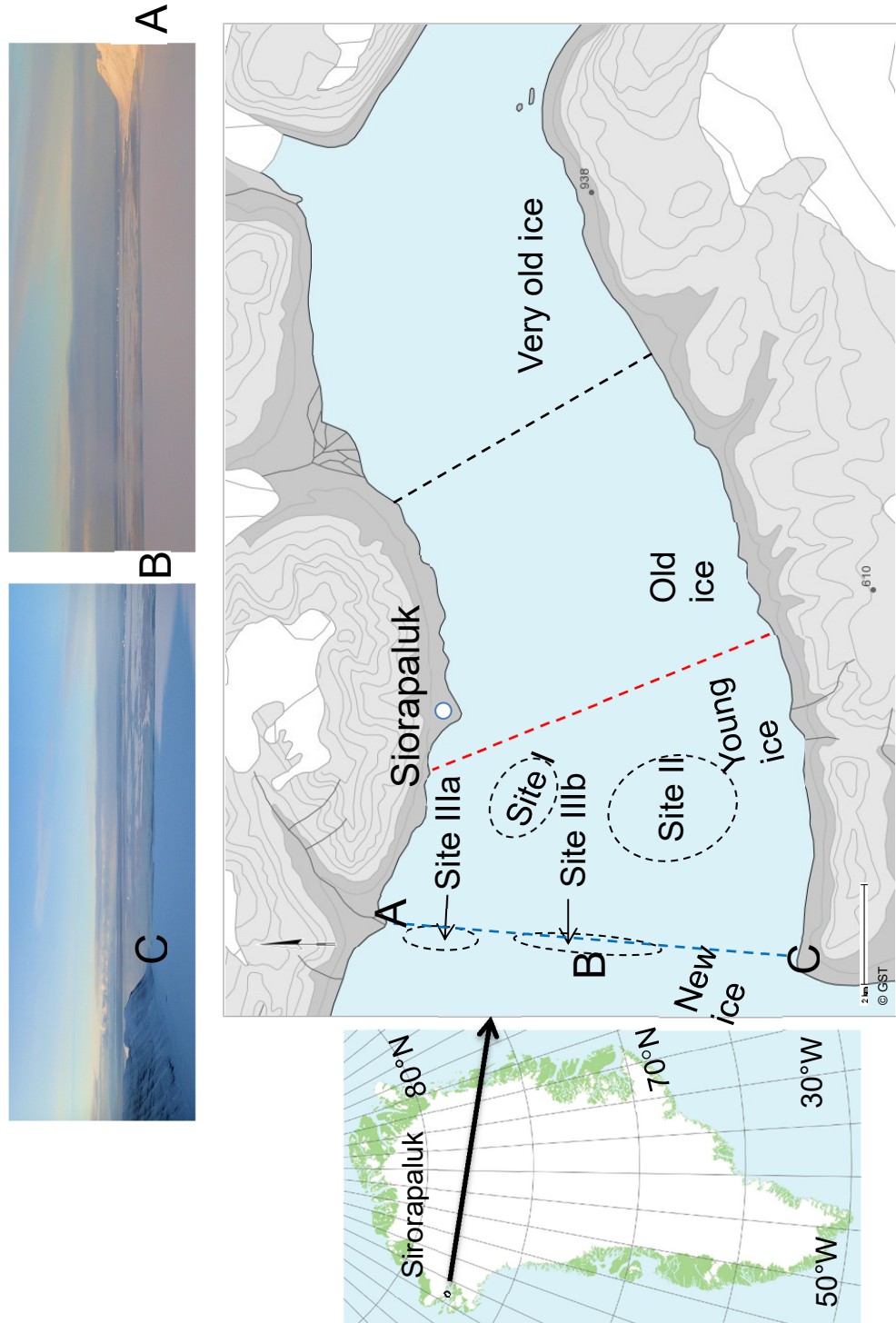

**Figure 1 Locations of sampling and sea-ice conditions around Siorapaluk and photographs of new sea-ice conditions off Siorapaluk (taken from helicopter on 7 March, 2014). Black, red, and blue broken lines show locations of sea-ice break in November, 2013, and on 10-14 Feberuary, 2014, and 1 March, 2014, respectively. Marks of A, B, and C denote broad locations in maps and photographs.**

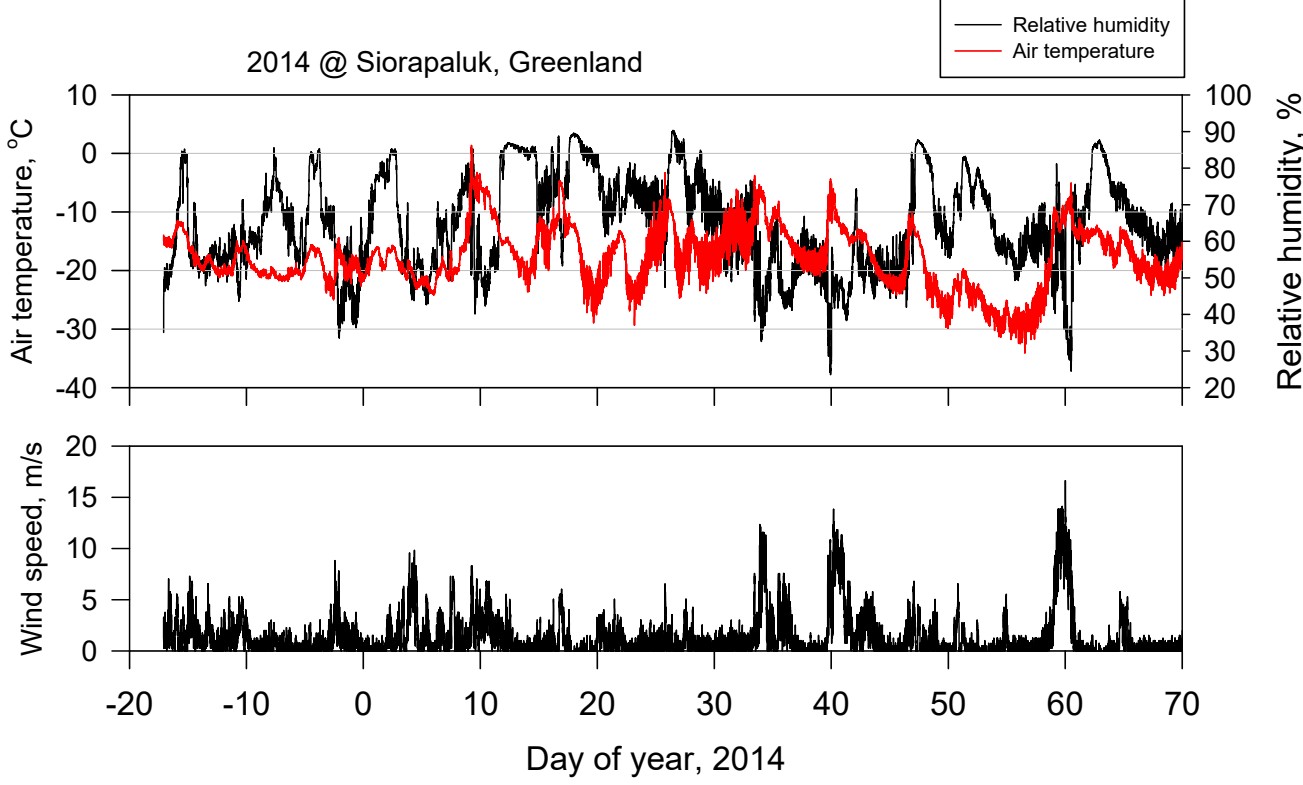

**Figure 2 Variation of air temperature, relative humidity, and wind speed at Siorapaluk.**

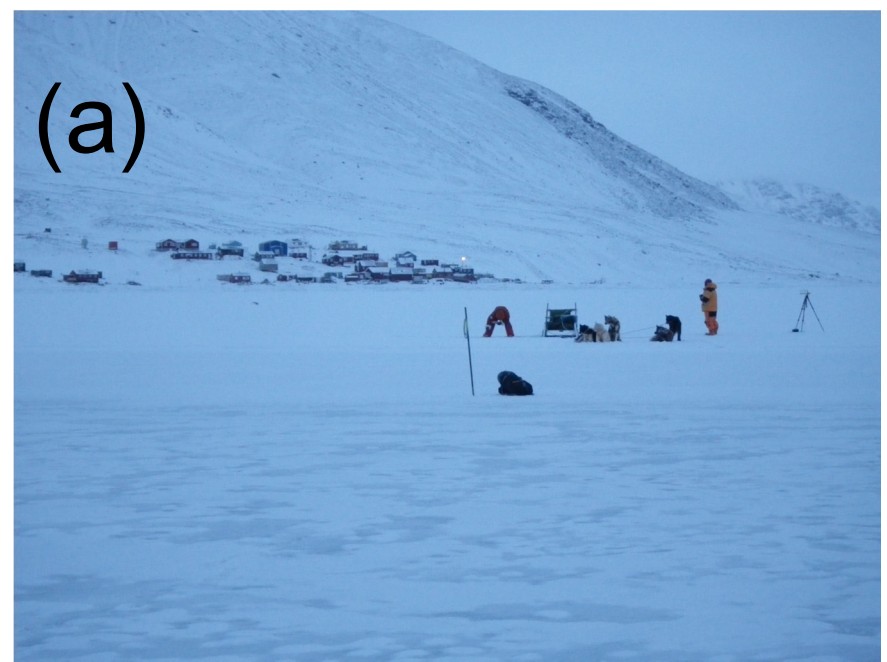

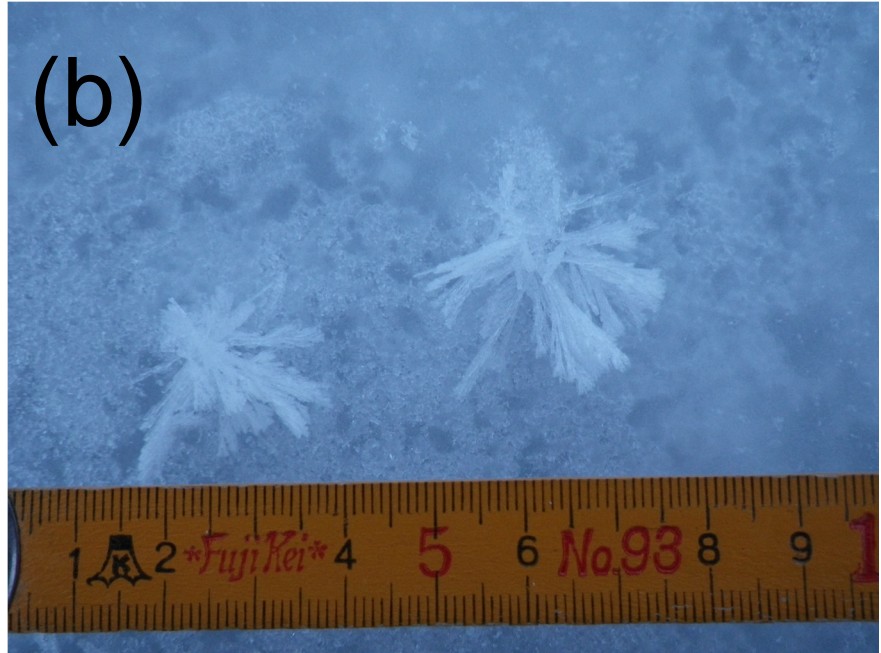

Fig.3

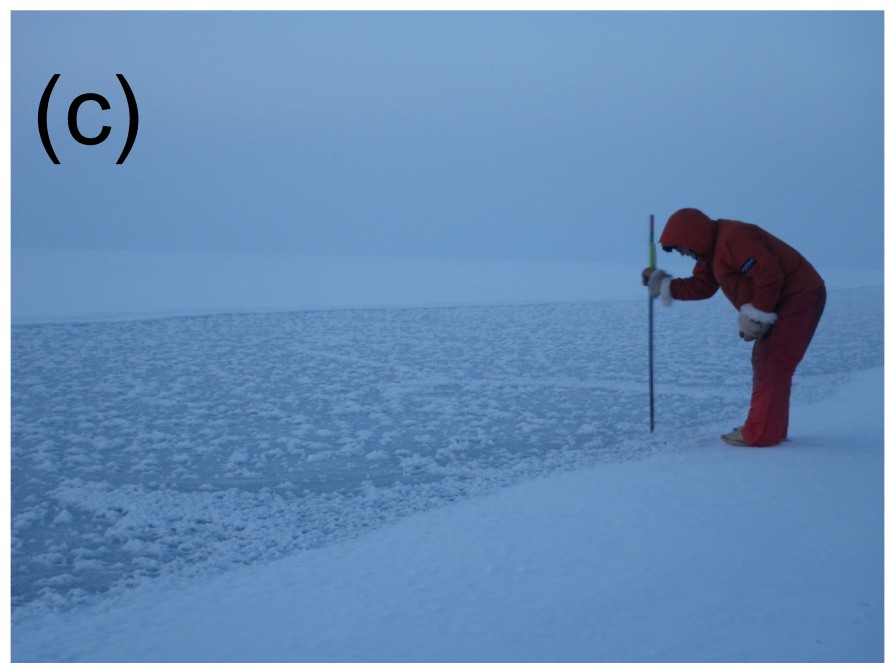

(c)

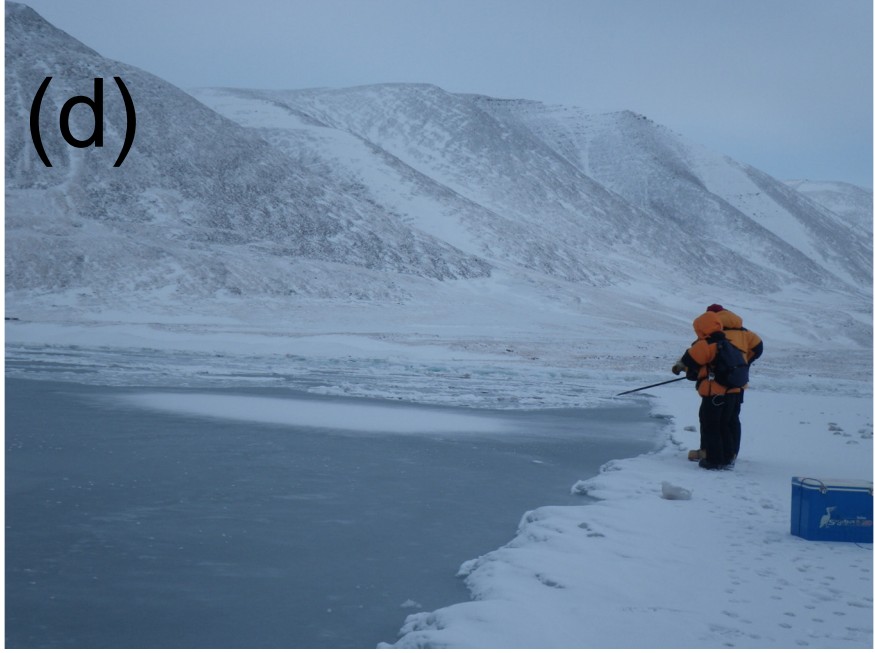

(d)

Fig.3

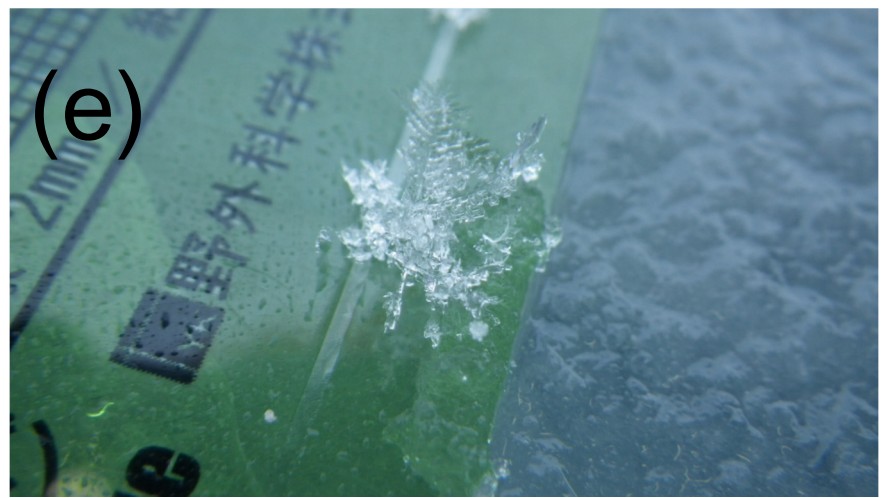

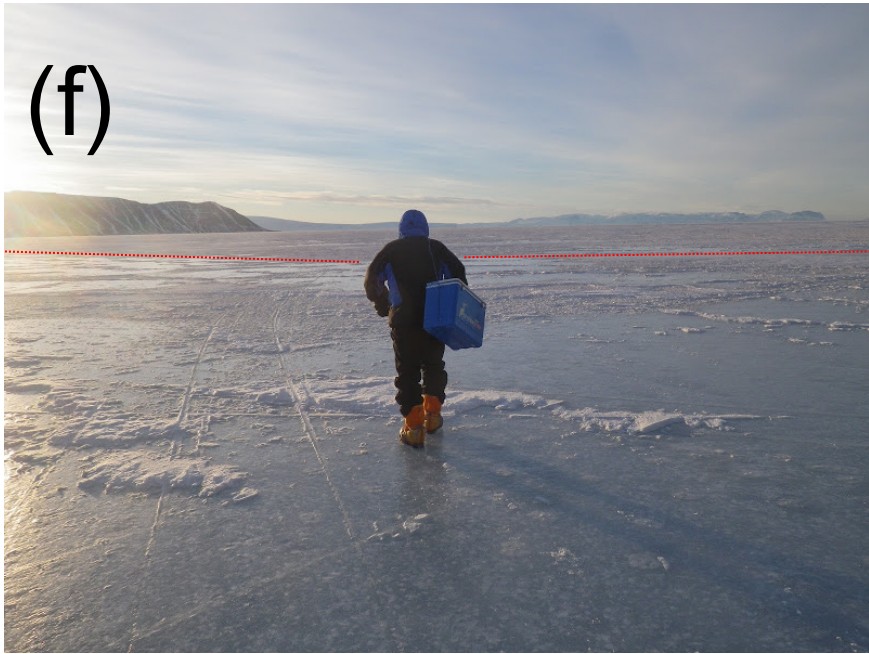

**Figure 3 Photographs of (a) condition of sea-ice and frost flowers on 20 February at Site I , (b) frost flowers at Site I on 22 February, (c) frost flowers at Site II, (d) sea-ice condition at Site IIIa, (e) frost flowers on 4 March at Site IIIa, and (f) condition of old sea-ice on 2 March immediately after the storm. Red dotted lines indicate approximate location of margin between old sea-ice and young sea-ice.**

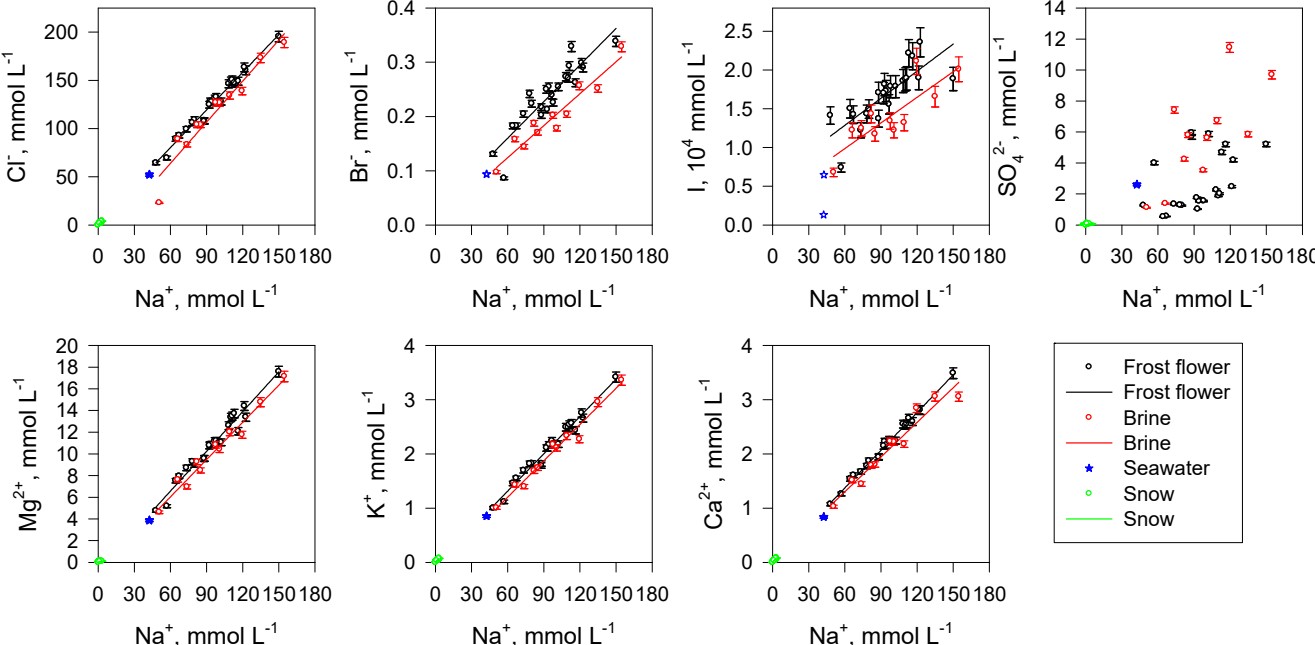

**Figure 4 Relation among each constituent to Na$^+$ concentration of frost flowers, brine, and seawater taken in this study. Black and red lines respectively present regression lines of frost flowers and brine. Open black and red circles respectively present concentrations in frost flowers and brine. Filled blue stars represent the concentrations of seawater taken around Siorapaluk. Open blue stars represent the concentrations of seawater in the literature presented in Table 1. Error bars indicate standard deviation (1σ) of analytical errors.**

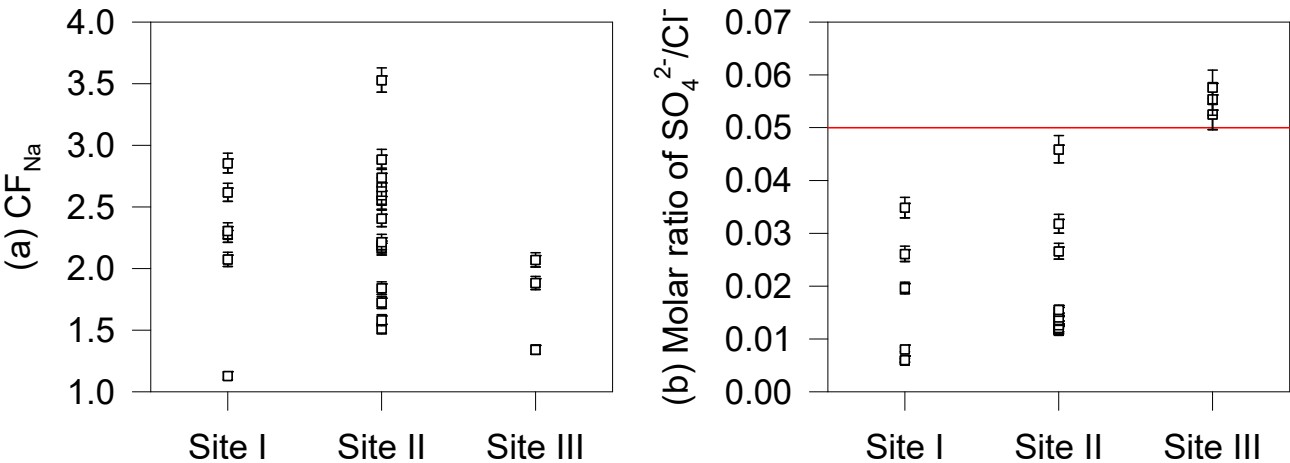

**Figure 5 Variations of (a) concentration factor of Na$^+$ (CF$_{Na}$) and (b) molar ratios of SO$_4^{2-}$/Na$^+$ in frost flowers at each sampling site. Red line indicates the bulk seawater ratio. Error bars indicate standard deviation (1σ) of analytical errors.**

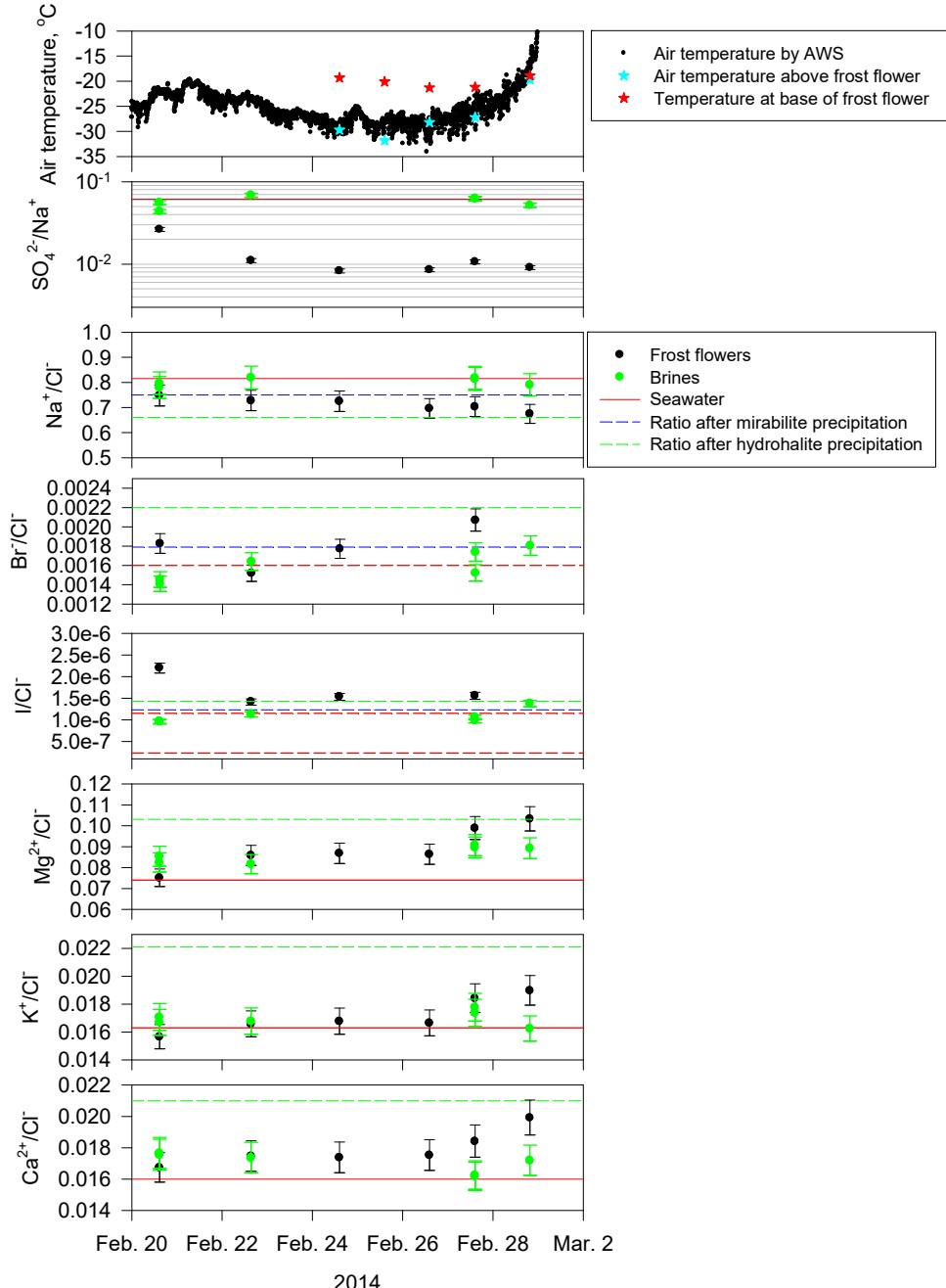

**Figure 6 Short-term features of (a) air temperature measured by AWS ($T_{AWS}$), air temperature above frost flowers ($T_{air}$, ca. 10 cm above the sea-ice surface), temperature of base of frost flowers ($T_{FF}$), and (b–h) molar ratios of sea-salts in frost flowers and brine at Site I. $T_{air}$ and $T_{FF}$ were not measured on 20–22 February. Error bars indicate standard deviation (1σ) of analytical errors. Blue and green dash lines the molar ratios after precipitation of mirabilite and hydrohalite, respectively.**

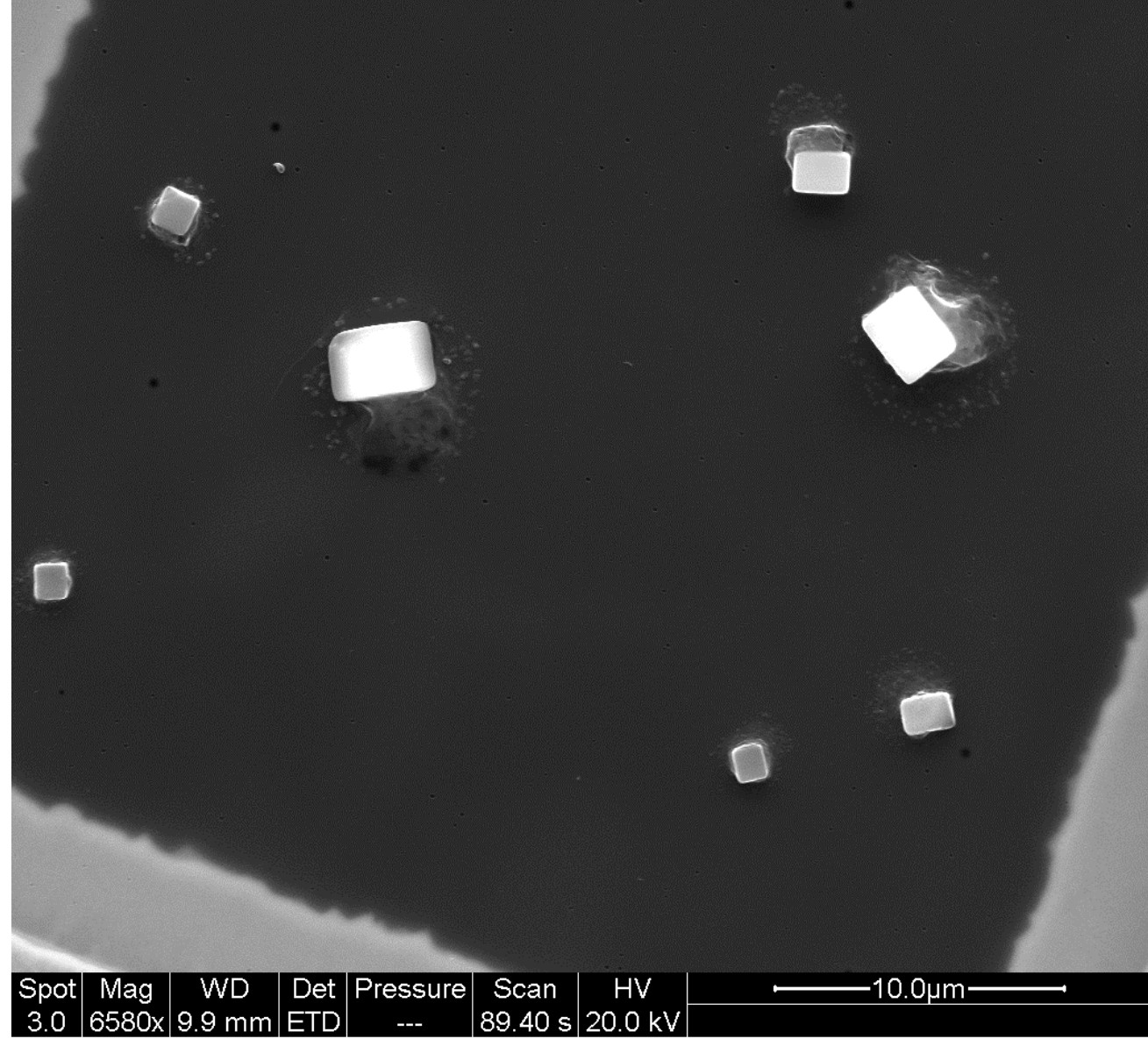

| Spot | Mag | WD | Det | Pressure | Scan | HV | ├────10.0μm────┤ |
|------|-----|-----|-----|----------|------|-----|---|
| 3.0 | 6580x | 9.9 mm | ETD | --- | 89.40 s | 20.0 kV | |

**Figure 7 SEM image of aerosol particles collected on 1 March, 2014 above the sea-ice area with frost flowers.**

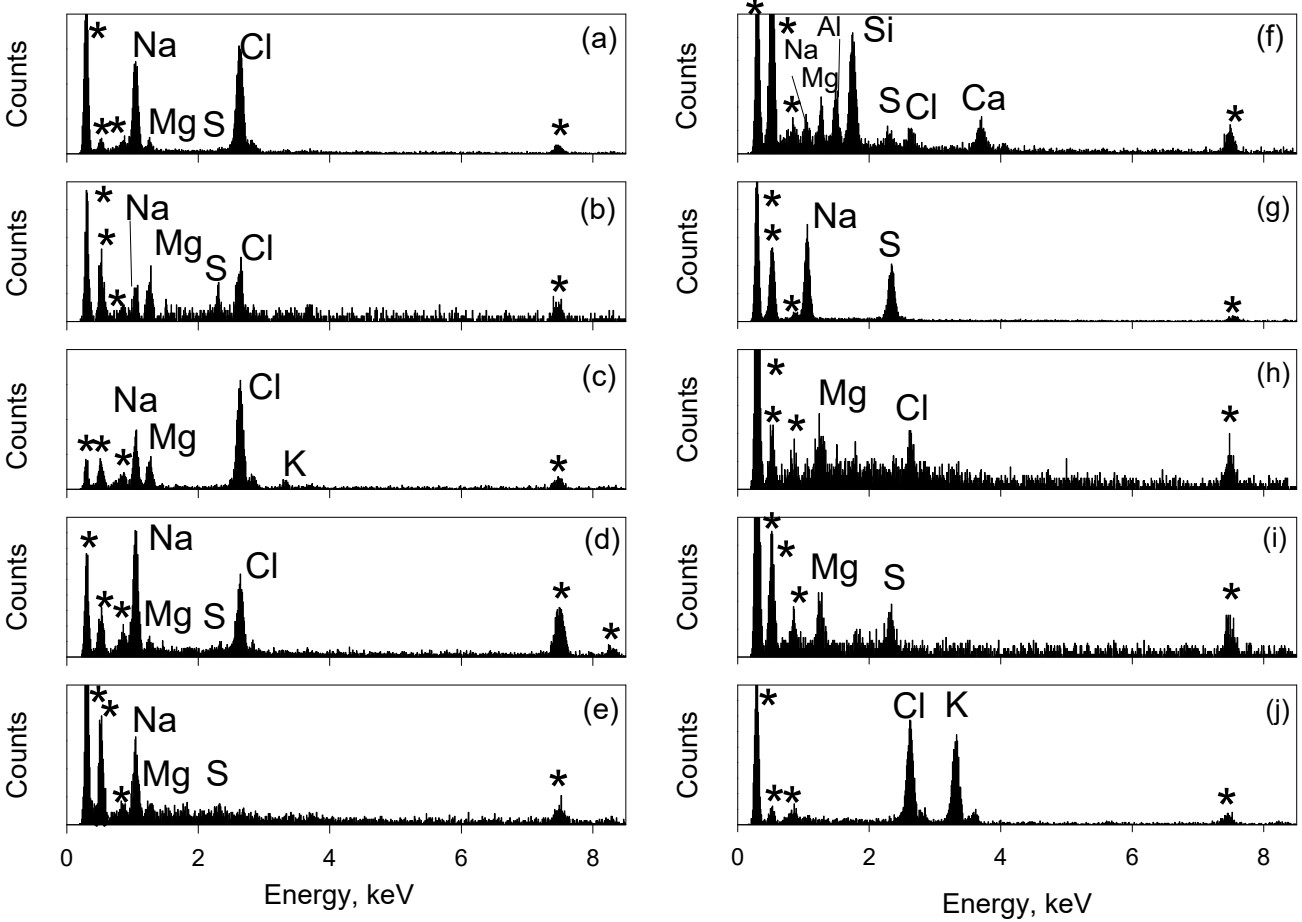

**Figure 8 EDX spectra of sea-salt particles and sea-salt relating particles collected over the sea-ice area. Asterisks denote background peaks derived from the sample substrate.**

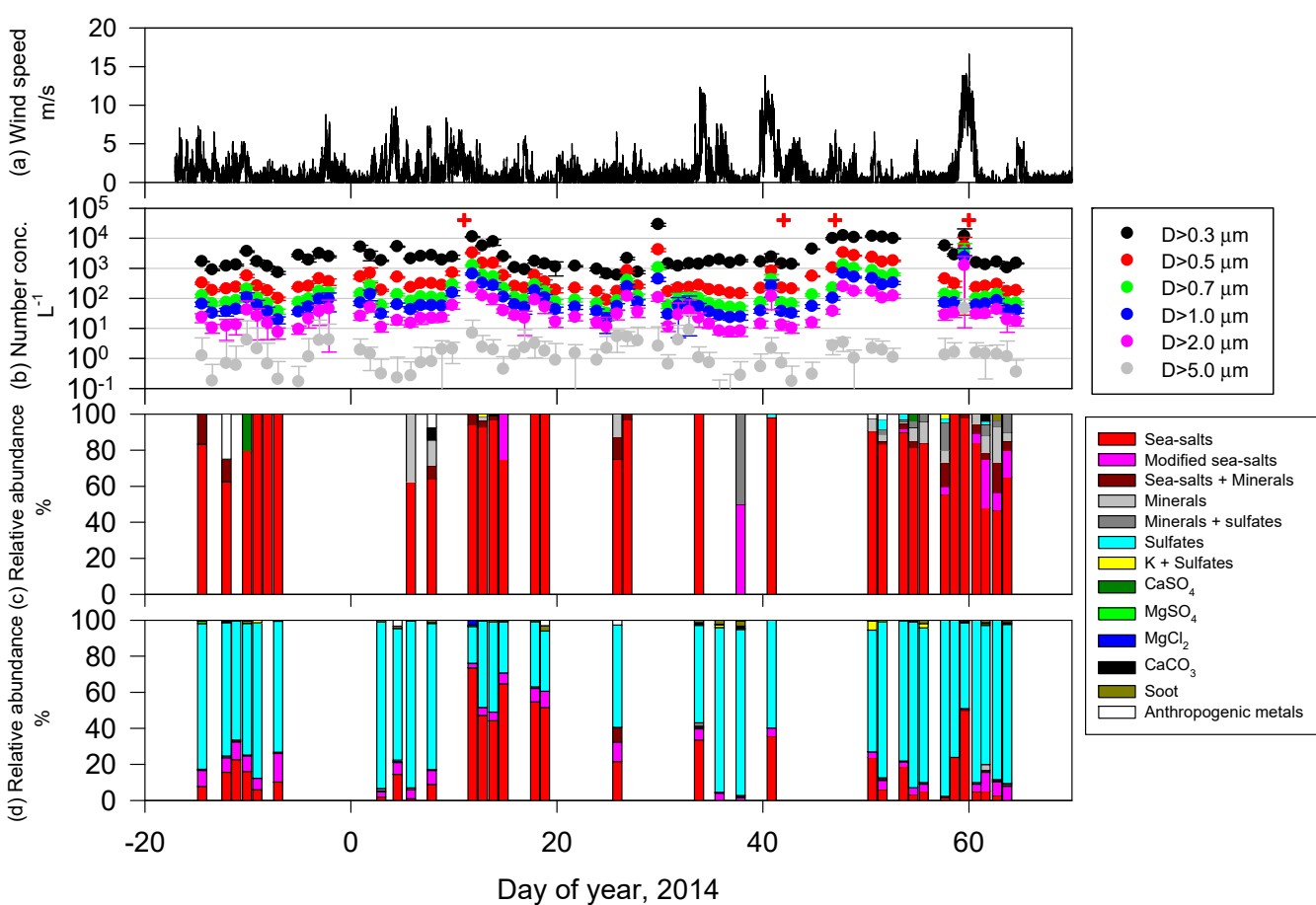

**Figure 9 Variations of (a) wind speed, (b) aerosol number concentrations, relative abundance of each aerosol type in (c) coarse mode and (d) fine mode. Red + marks indicate the date when the low clouds (fogs) were identified off Siorapaluk.**

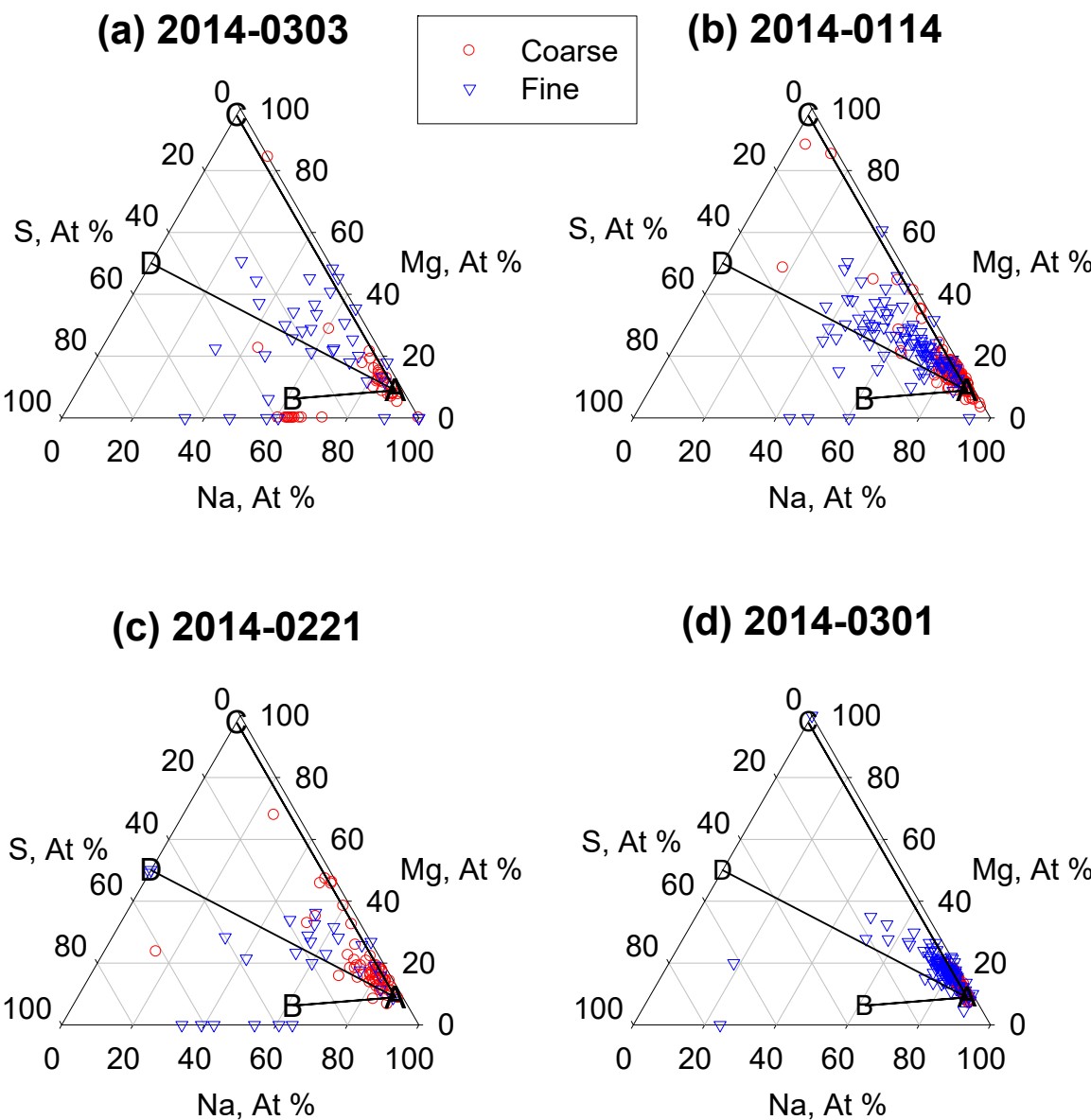

**Figure 10 Ternary plots of Na–Mg–S of sea-salt particles and the modified sea-salt particles collected over the sea-ice area. A, B, C, and D in this figure denote respectively ratios of seawater, fully modified with $SO_4^{2-}$, $MgCl_2$, and $MgSO_4$,. Red open circles and blue triangles respectively present ratios of particles in coarse and fine modes.**

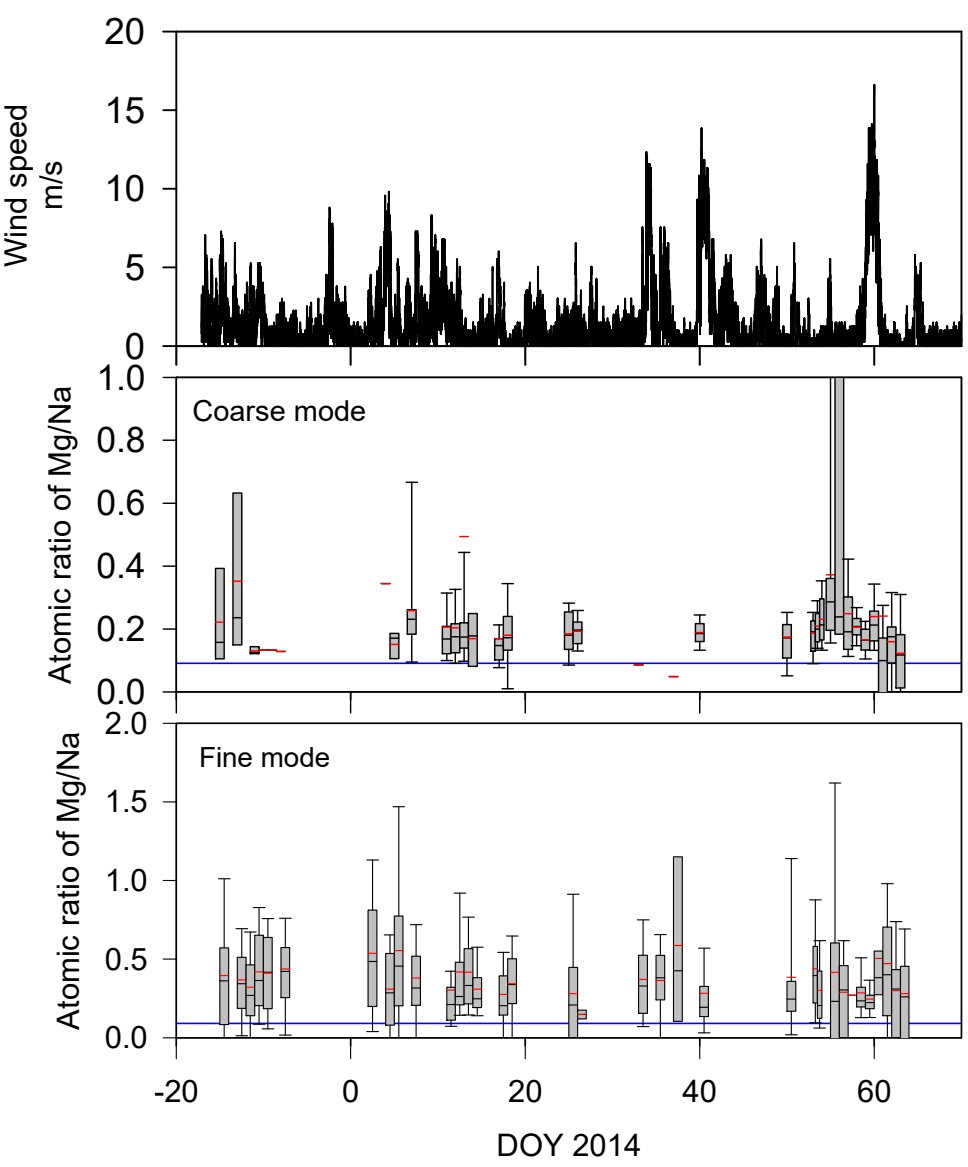

**Figure 11 Variations of wind speed and atomic ratios of Mg/Na in sea-salt particles and the modified sea-salt particles collected over the sea-ice area. In box plots, the top bar, top box line, black middle box line, bottom box line, and bottom bar respectively denote values of 90, 75, 50 (median), 25, and 10%. The red and lines show mean values and seawater ratio of Mg/Na.**

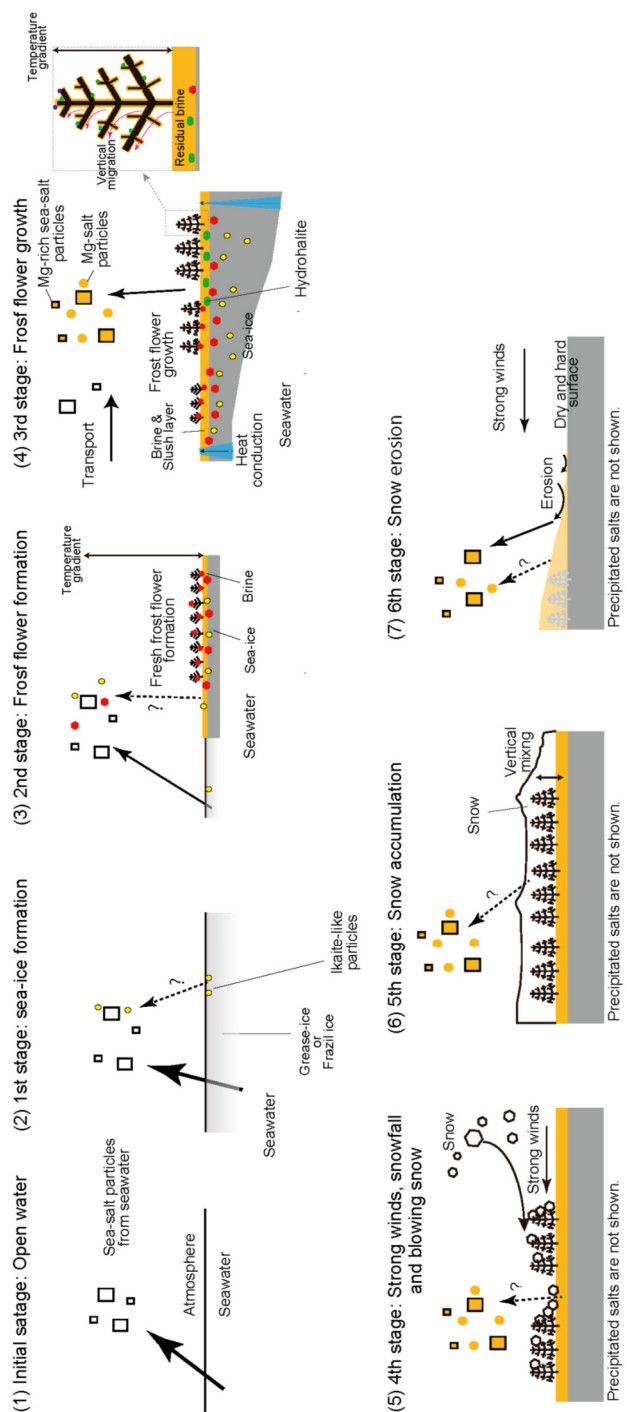

**Figure 12 Schematics of sea-salt cycles in sea-ice area. Dotted arrows indicate the speculated processes.**