# Peer review of "Frost flowers and sea-salt aerosols over seasonal sea-ice areas in north-western Greenland during winter—spring"

_Atmospheric Chemistry and Physics, 2016_

## Referee Comment (RC1) · H. K. Roscoe (Referee) · 21 Dec 2016

This paper presents a wealth of information on composition of frost flowers and aerosol from an Arctic field campaign, with a detailed discussion of results. It is very well written except for the trivia listed under Editorial Comments below.

Major Comments:

A. Except for a couple of lines buried on p13, the authors ignore the discovery in 2008 and 2010, in field measurements and separately in laboratory measurements, that frost flowers can be of non-fragile structure not dispersed by wind, and in the laboratory not producing aerosol (Obbard et al. 2009, Roscoe et al. 2011). This discovery was

consistent with the then new theory that wind-blown snow played a major role in the mobilisation of Br to the atmosphere in the sea ice zone (Yang et al. 2010). Although many previous field measurements had been consistent with the scenario presented by the authors in their Introduction, others had not - instead they had been consistent with the discovery in 2008 and later. Before publication, the Introduction must be revised to discuss the ambiguous nature of frost flowers and their mobilisation; Yang et al. (2010) must be included in the References; and the discussion on p13 lines12-16 must be enlarged.

B. There is no major scientific conclusion in the paper - it is an extensive and comprehensive report of the measurements and what they might mean in detail, but not what they might mean to Atmospheric Chemistry and Physics in general. Without resorting to a new set of work with a model, I doubt that there could be such a conclusion. It is up to the Editor whether or not this is a bar to publication.

Minor Comments:

1. Despite the text on p3 and elsewhere throughout the manuscript, we discover from Figure 1 that Site III is actually split into two. It would help if some reason for this was given, and if later text and captions said whether the result being described was from Site IIIa or IIIb .

2. The authors describe the good practice of storing brine samples in Japan frozen until analysis (p5 line11), but it seems that they were transferred to Japan unfrozen - presumably because of the difficulty of carrying frozen samples by air. How long were the samples left unfrozen - 4 days, or several weeks?

3. It is also good practice to keep aerosol samples collected in polar regions frozen for shipment and storage until analysis. This is not mentioned on p4 lines27-29, so we assumed they were not frozen, in which case more should be made of the potential for change during storage than the one line given here, and the unfrozen time between collection and analysis should be stated.

4. The description on p6 of Figures 3(a) to (e) bears little relation to the order of (a) to (e) in the Figure itself. Nor does the caption of Figure 3, which does not even mention Figure 3(e). This should be sorted out.

5. From the scenario of frost flower formation given in the introduction, we would expect sulphate depletion in surface brine to be equal to that in the frost flowers - the flowers are expected to wick this brine, so if sulphate deposition occurs only in the colder temperatures above the surface then the frost flower total sample would contain the original brine sulphate. The fact of depletion in frost flowers but not brine in Figure 6 is a strange and new result, which deserves comment in p9 para1. Might it be related to the sampling protocol for surface brine given in p4 lines5-6? This suggests a large amount of solid was removed together with any liquid, thereby possibly incorporating ice into which sulphate had precipitated. This should be discussed, as it may also affect some of the discussion about changes in Mg, Br and Ca after 26 Feb.

6. The discussion about possible removal of Br and I from frost flowers by heterogeneous reactions on 27 Feb (p10 lines21-21) stretch the apparent accuracy of the measurements - the difference between 0.00206 and 0.00214 for Br is surely not significant, given the scatter in Figure 4b. If the Br and I differences are significant, error bars should be derived.

7. The statement on p17 line23 that sea salts (Mg etc) were "remarkably enriched" is not borne out by Figure 6. Depending on the error bars, they may not be enriched at all. If "remarkably" is removed, the sentence can probably stand, but better would be a derivation of error bars.

8. The caption to Figure 1 must contain some details of the broken lines described on p3. It is not acceptable to have figures with important features that are not described in the caption.

Editorial Comments:

[Figure]

**[ACPD](https://www.atmos-chem-phys-discuss.net)**

Interactive
comment

p1 line12 - replace "sea salts" by "sea-salt constituents".

p3 line23 - either "Sea ice forms gradually from October" or "Sea ice formed gradually from October 2013".

p3 line25 - delete the second "the fjord".

p5 line1 - insert "a" or "the" before "data logger"; replace "in" by "with a".

p5 lne3 - replace "slash" by "slush".

p8 line22 - replace "Site I" by "Site".

Fig1 - must be rotated and "Siorapaluk" made legible.

Fig5 - shows values at Sites A, B and C which should presumably be Sites I, II and III.

Fig5 caption - define the red line in Fig5(b).

Fig9 caption - does not say which of the two lower figures is coarse and which is fine.

Fig S2 caption - does not agree with the axis legends of the figures.

Figs S3, S4, S5 - must be rotated.

Fig S5 caption - only describes one of the two images and spectra shown in the figure.

Additional Reference:

Yang, X., J.A. Pyle, R.A. Cox, N. Theys, M. Van Roozendael, "Snow-sourced bromine and its implications for polar tropospheric ozone", Atmos. Chem. Phys., 10, 7763–7773, doi:10.5194/acp-10-7763-2010 (2010).

---

## Referee Comment (RC2) · F. Dominé (Referee) · 12 Jan 2017

This paper contains an impressive amount of data on frost flower and surface brine composition on Arctic sea ice, as well as a large amount of varied data on aerosol size distribution and chemical composition at the same sites. To my knowledge, such a data set is novel and deserves to be made available to the scientific community and to be fully exploited in terms of physical and chemical processes.

I however have to say that I was disappointed by the very limited use of the data made by the authors. No real scientific plan seems to have driven their campaign other than data collection, and this paper falls short of providing significant new insights into processes at the surface of sea ice, processes of aerosol generation above sea ice, in the

presence and absence of frost flowers (FF), and processes of aerosol chemical evolution, although this last aspect is somewhat discussed. Clearly, the scientific discussion is not up to the level of the data set and at present, I just do not think that the paper is good enough for publication in ACP. The somewhat tedious point-by-point description of results should be completely replaced with a discussion focussed on solving a few selected scientific questions such as for example : "how does the presence of FF affect aerosol composition?" or "what are the processes leading to halogen enrichment in FF and surface brine?". More aggressive attempts to make deductions from the observations are mandatory. For example the authors do not even consider the surface to volume ratio of aerosols to attempt to quantitatively examine chemical evolution. An all too frequent phrase is "Similar observations were made at Syowa" and I must confess that after reading this over 10 times without any subsequent deduction of any process, I started to become a bit frustrated, perhaps even irritated. The authors may cross their data with GOME2 BrO data to perhaps reach some interpretation on halogen activation.

http://atmos.eoc.dlr.de/gome/product_bro.html

http://atmos.eoc.dlr.de/gome/image_browsing.html

http://www.iup.uni-bremen.de/doas/scia_data_browser.htm

In summary, before I can recommend serious consideration of publication, I strongly suggest to completely reorganize the paper as follows:

1- Select a couple of novel scientific questions to be addressed by the data set.

2- Select the data to be presented to address the selected questions. A couple of case studies focused on a few events may be interesting.

3- Separate results and discussion and write in a much more concise form to produce a much shorter paper.

4- Reach some strong and novel conclusion. For example, finding out that the presence

of FF does not significantly affect aerosol composition would be quite interesting.

5- Place the data not used here but of potential interest to others in supplementary material or any other accessible place.

The authors should feel free to adopt any other strategy, the objective being to make a good and concise use of the data to derive strong conclusions. At present, the manuscript is more a detailed preliminary campaign report than an actual scientific paper.

Specific points.

Page 1, line 28. Vapor is supplied TO the atmosphere, not FROM. See the references on the same line. 2, 31. Specific surface areas are now expressed in m2 kg-1. Please convert.

Section 2.2. Was snow present in FF and brine samples? This should be mentioned as it dilutes the samples. Re. section 3.4.

6, 7. Please add a + sign: +1.8°C, to avoid any ambiguity.

Throughout: replace liberated with released

7, 18. Replace correlation with determination.

7, 32. The structure of the paper is such that the mention of solar radiation here is a bit odd and unexpected, and maybe not readily understood by all. Separating results and discussion would have helped.

End of section 3.3. A more in-depth discussion of the causes of Br and I enrichment is in order.

9, top. Air T is a useful variable for many purposes, but the actual variable of interest here is surface T. All the speculation between air T and processes is really not useful, unless a surface T can be produced. Several lengthy discussions could just be
removed.

9, 28. Replace larger by greater. The sea ice thickness may be important for the relationship between air and surface T, but not for surface processes. This is a good example that the variable of interest is the surface T, not the air T or the sea ice thickness.

10, 4-5. This statement does not lead to any useful conclusion. Please delete.

11, 5. Since or until ?

12, 1-3. What useful conclusion do we derive from these Mg-rich and K-rich particles? Data description just is not enough for a scientific paper.

13, 9. Could you please discuss the presence of non-sea salt sulphate?

14, 11-13. This is where your impressive data set could be put to good use to address these points. "At the moment, release processes of mirabilite-like and ikaite-like particles from the sea-ice surface without frost flowers remain unknown". Sure, but is not this campaign supposed to contribute to solving this problem ?

14, 17. "Therefore, most of the aerosol particles around $Na_2SO_4$ ratio in fine mode might be the modified sea-salt particles by heterogeneous reactions with $nss-SO_4^{2-}$." Can't you get to a stronger statement than just "might" by more in-depth examination of your data?

15, 6. "Therefore, sea-salt modification (Cl loss) might be most likely to occur in fine mode." Sure, you may even use your data quantitatively and examine the role of aerosol surface to volume ratio on reaction kinetics. Again, "might" is not sufficient here.

16, 33 to 17, 6. Again, a more in-depth use of the data should allow useful conclusions, not vague suppositions.

17, last lines. These mentions of just observations, without any scientific deductions, are very disappointing. The authors have a unique data set and hardly do anything

novel with it. Honestly, I expected a breakthrough paper, which is what those data deserve, and only get descriptions. How frustrating ! I am certain that you can do much more, please just do it!

---

## Referee Comment (RC3) · Anonymous Referee #3 · 1 Feb 2017

This manuscript describes measurements of frost flowers and sea salt aerosol particles during winter/spring 2014 on the coast of Greenland. The goal of simultaneous measurement of potential sea salt aerosol sources (e.g. frost flowers) and aerosol is commendable. However some potential sources of aerosol (e.g. surface snowpack) do not appear to have been measured, and the data presentation and argumentation in the manuscript is not very clear. The manuscript does show some results that appear significant. The enrichment of sea salts in frost flowers compared to sea water and depletion of sulfate in frost flowers compared to sea water ratios are both observed in agreement with past literature. However, the subsequent points of the abstract are not well argued for or are not clear.

[Figure]

Major points

The manuscript lacks an error analysis that would allow one to determine the significance of results. Words like "significant" are used, but that is not clearly related to a rigorous statistical definition or simply a qualitative word. To make statements about composition being different from sea water ratios, the authors would need to discuss error analysis more rigorously. For example, figures 4, 5, and 6 should all have error bars. Particularly in Figure 6, the lack of error bars makes it challenging to tell if deviations from the sea water composition is significant or not. The text seems to indicate that sea water was sampled, which could be used to assess random and systematic errors in analysis of these ionic species, but no such analysis is shown. This is particularly concerning for species that are analyzed by different methods (e.g. I- and Br- are analyzed by different methods than Cl-, and cations in different runs from Cl-).

The manuscript contains many technical errors and is difficult to read due to non-standard use of English. An example of this problem is on page 6, line 19, where the text says "Figure 3b presents a close up photograph of frost flowers...". Figure 3b is a photo of people on sea ice and houses on the shore. Many other examples of technical errors are in the text, another being page 15, line 18 indicates A, B, and C are on a diagram, but that diagram actually has A through D. The use of ternary plots is also not clear. Normally ternary plots are useful when the sum of the three components is 100%. However, in Figure 10, Cl, S, and Na are plotted as atomic percentages. These samples also have other atoms in their composition (e.g. oxygen that is a part of SO4, Mg, etc.), so I guess that the plots show atomic percentage of Cl, S, and Na atoms? Some points are then very strange, such as on Figure 10b, three course points have >80% Cl and little Na. Something must charge balance the Cl-, but that is not clear on the plot.

The paper often describes frost flowers as "fragile", while laboratory studies of frost flowers in a wind tunnel failed to produce aerosol, and field studies often show frost flowers get buried under blowing snow (e.g. snow blows while frost flowers remain

intact). Therefore, there is not clarity in the literature that frost flowers are the only source of sea salt aerosol, and to the contrary, blowing snow and/or aerosol production from open water are often discussed in the literature. This manuscript doesn't describe the chemical composition of snow, which could be relevant to the production of aerosol, nor does it consider nearby open water and potential of aerosol production from that source.

The manuscript claims in the abstract that "Aerosol number concentrations, particularly in coarse mode, were increased considerably by release from sea-ice surface under strong wind conditions." However, the figures and text really do not back up that claim. Figure 9 is presumably the data for this claim, but the authors do not indicate what periods to look at on that figure to see they effect they claim. In general, I see high coarse-mode aerosol on about DOY 12-18, 30, and 50-55. Those periods often have some winds, but not peak winds. Two points at peak wind periods (DOY 41 and 59) do show spikes in aerosol, but other high wind periods don't have spikes (e.g. DOY 36 and 38). Overall, the authors need to make a clearer argument to their claim.

The claim that bromide is being released from frost flowers made at the bottom of page 9 and page 10 would seem to imply a large release of bromine from frost flowers to the atmosphere. The authors should do a mass balance argument to indicate how much bromine would be released from this proposed release and compare the value to observations of atmospheric bromine (e.g. BrO). If that calculation led to unreasonably large BrO concentrations, then it would be evidence against this hypothesized direct halogen release. The lack of error analysis also makes it challenging to tell what is significant on these plots. Lastly, field evidence (Pratt et al., 2013) and multiple laboratory studies indicate that the pH of surfaces should be acidic for efficient halogen release, while highly saline samples were not efficient at releasing halogens.

The referencing of the paper is not accurate. An example of this is on line 3-4 of page 3, which the authors say "Reportedly, Br- enrichment occurs in frost flowers at Barrow, Alaska (Douglas et al., 2012)...". However, the text of that citation says "There is no

enhancement in bromide to chloride ratios in the frost flowers compared to brine or seawater".

Overall

This paper needs major revisions to be acceptable for publication. There is a great deal of interesting data in the publication, but it is not presented in a form that gains scientific understanding from the data.

---

## Author Comment (AC1) · 28 Feb 2017

**Reply to Referee #1 (Dr. Roscoe)**

We would like to thank your helpful comments to improve our manuscript. We responded to the general and specific comments. All comment are addressed in the revised manuscript. The updated parts by your comments indicates red words in the revised manuscript (pdf file). We separated Results and Discussion on basis of comment from Referee #2 (Dr. Dominé).

*Comment from Referee*

> *A. Except for a couple of lines buried on p13, the authors ignore the discovery in 2008 and 2010, in field measurements and separately in laboratory measurements, that frost flowers can be of non-fragile structure not dispersed by wind, and in the laboratory not producing aerosol (Obbard et al. 2009, Roscoe et al. 2011). …*

**Reply from authors**

> We agree with your comment. Description in "Introduction" and "Discussion" were modified as follows.
>
> Introduction
> Frost flowers are ice crystals containing brine and sea salts.
>
> Frost flowers have a fine structure. Previous field and laboratory experiments indicated that frost flowers were less-fragile even under strong winds in spite of the fine structure (Obbard et al., 2009; Roscoe et al., 2010). In addition, model studies implied that blowing snow had significant contribution to atmospheric halogen chemistry (Yang et al., 2010; Abbatt et al., 2012; Lieb-Lappen and Obbard, 2015).
>
> Concluding remarks
> Under the conditions with strong winds, snowfall and blowing snow, snow particles were attached on frost flowers and slush layer. As suggested by laboratory experiments (Roscoe et al., 2010), aerosol particles are released insignificantly from frost flowers. However, Mg-rich sea-salt particles and Mg-salts are released from the brine and surface snow by winds.
>
> Also, Yang et al. (2010) was added to the statements in the manuscript and "References".

*Comment from Referee*

*B. There is no major scientific conclusion in the paper - it is an extensive and comprehensive report of the measurements and what they might mean in detail, but not what they might mean to Atmospheric Chemistry and Physics in general. Without resorting to a new set of work with a model, I doubt that there could be such a conclusion.*

**Reply from authors**

Thank you for your suggestion. We changed completely the description in "Concluding remarks". From field evidences in this study and previous works, we proposed sea-salt cycles in the seasonal sea-ice area in "Concluding remarks". We believe that our proposal (hypothesis) is useful and new set for sea-salt chemistry in the polar regions.

*Comment from Referee*

*1. Despite the text on p3 and elsewhere throughout the manuscript, we discover from Figure 1 that Site III is actually split into two. It would help if some reason for this was given, and if later text and captions said whether the result being described was from Site IIIa or IIIb.*

**Reply from authors**

Age of sea-ice at Site IIIa and IIIb was almost same, because the storm on 1 March, 2014 caused sea-ice breakage at sea-ice margin near Site III (blue line in Figure 1). We choose Site IIIa and IIIb as sampling sites near new sea-ice by traverse by dog-sledge team. Consequently, we denote both sites as "Site III". From your suggestion, we used Site IIIa and IIIb in the revised manuscript. Descriptions about sea-ice conditions and sea-ice age were added to section of "2.1 Sampling sites and conditions".

*Comment from Referee*

*2. The authors describe the good practice of storing brine samples in Japan frozen until analysis (p5 line11), but it seems that they were transferred to Japan unfrozen presumably because of the difficulty of carrying frozen samples by air. How long were the samples left unfrozen - 4 days, or several weeks?*

**Reply from authors**

The samples were kept below -20°C in Greenland and our laboratory. During the

transport from Greenland to Japan, samples were kept in "unfrozen" for ca. 3 days. The description was added to "2.2 Sampling of frost flowers, brine, snow and sea-ice".

**Comment from Referee**

*3. It is also good practice to keep aerosol samples collected in polar regions frozen for shipment and storage until analysis. This is not mentioned on p4 lines27-29, so we assumed they were not frozen, in which case more should be made of the potential for change during storage than the one line given here, and the unfrozen time between collection and analysis should be stated.*

**Reply from authors**

Aerosol samples for single particle analysis are not kept in freezer to avoid morphology change and heterogeneous reactions by humidification. Some information and the unfrozen periods were added to "2.3 Aerosol sampling and measurements".

**Comment from Referee**

*4. The description on p6 of Figures 3(a) to (e) bears little relation to the order of (a) to (e) in the Figure itself. Nor does the caption of Figure 3, which does not even mention Figure 3(e). This should be sorted out.*

**Reply from authors**

This point is our editorial error. We correct order of the pictures.

**Comment from Referee**

*5. From the scenario of frost flower formation given in the introduction, we would expect sulphate depletion in surface brine to be equal to that in the frost flowers - the flowers are expected to wick this brine, so if sulphate deposition occurs only in the colder temperatures above the surface then the frost flower total sample would contain the original brine sulphate. The fact of depletion in frost flowers but not brine in Figure 6 is a strange and new result, which deserves comment in p9 para1. Might it be related to the sampling protocol for surface brine given in p4 lines5-6? This suggests a large amount of solid was removed together with any liquid, thereby possibly incorporating ice into which sulphate had precipitated. This should be discussed, as it may also affect some of the discussion about changes in Mg, Br and Ca after 26 Feb.*

**Reply from authors**

In our sampling, all body of frost flowers taken from sea-ice. On the other hands, brine, ice, and solid salts were samples ad "brine", because it is difficult to segregate brine from slush (ice and solid salts). When some salts such as mirabilite and hydrohalite were precipitated in the slush layer, we cannot identify the evidence of salt precipitation from sea-salt ratios (e.g., $SO_4^{2-}/Na^+$). Temperature at surface of brine (base of frost flowers), $T_{FF}$, dropped to temperature at precipitation of mirabilite and hydrohalite, so that these salts might be precipitated and the n the residual brine might be migrated onto frost flowers. These description were added to "2.2 Sampling of frost flowers, brine, snow and sea-ice" and "4.2 Aging of frost flower and sea-salt fractionation".

*Comment from Referee*

*6. The discussion about possible removal of Br and I from frost flowers by heterogeneous reactions on 27 Feb (p10 lines21-21) stretch the apparent accuracy of the measurements - the difference between 0.00206 and 0.00214 for Br is surely not significant, given the scatter in Figure 4b. If the Br and I differences are significant, error bars should be derived.*

**Reply from authors**

We agree with your comments. We added analytical error bars in determination in IC, IC-MS and ICP-MS into the plots of Fig. 6. Therefore, $Br^-$ might be released slightly of insignificantly in our research conditions. On the other hands, iodine appeared to be released from frost flowers. Consequently, we updated description about $Br^-/Cl^-$ and $I/Br^-$ in frost flowers in the revised manuscript, as follows.

To elucidate halogen chemistry in frost flower, we compared between the features of $Br^-/Cl^-$ and $I/Cl^-$ in frost flowers at Site I. High correlation of $Cl^- - Na^+$ and $Cl^-$ enrichment strongly suggests that $Cl^-$ release was insignificant from frost flowers and brine. Thus, we focus on the features of $Br^-$ and I, here. we attempt to estimate the molar ratios of $Br^-/Cl^-$ and $I/Cl^-$ in frost flowers using the ratios of $Mg^{2+}/Cl^-$, $Br^-/Cl^-$ and $I/Cl^-$ on 22–26 February, assuming that hydrohalite was not precipitated yet on 22–26 February, and that $Br^-$ and I did not liberate from frost flowers through heterogeneous reactions. When the molar ratios on 22–26 February changed by the assumptions above, the molar ratios of $Br^-/Cl^-$ and $I/Cl^-$ in frost flowers after hydrohalite precipitation were estimated respectively as 0.00214 and $1.82 \times 10^{-6}$.

Although Br$^-$/Cl$^-$ ratio (0.00206) was slightly lower than the estimated ratio, this difference might be very slightly or insignificant. Therefore, Br$^-$ release from frost flowers might be insignificant at Site I under the dusk conditions. On the other hands, the estimated I/Cl$^-$ ratio was higher than the ratios in frost flower ($1.562 \times 10^{-6}$) on 27 February. The following likelihood should be considered: (1) reduction of I enrichment by precipitation of salts containing iodine, and (2) I release from frost flowers through heterogeneous reactions. Although NaBr•5H$_2$O can be precipitated at -28 °C (Koop et al., 2000), no report in the relevant literature discusses a study of precipitation of iodine salts in sea salts. If iodine salts were not precipitated under the conditions at Site I, this difference implies the likelihood that I was released from frost flowers. Iodine can be released from frost flowers through the following heterogeneous reactions (Thompson and Zafiriou, 1983; Carpenter, 2003; Simpson et al., 2007; Saiz-Lopez et al., 2015).

**Comment from Referee**

*7. The statement on p17 line23 that sea salts (Mg etc) were "remarkably enriched" is not borne out by Figure 6. Depending on the error bars, they may not be enriched at all. If "remarkably" is removed, the sentence can probably stand, but better would be a derivation of error bars.*

**Reply from authors**

We addressed all of descriptions in "Concluding remarks". Therefore, the sentences with "remarkably enriched" were removed from the revised manuscript.

**Comment from Referee**

*8. The caption to Figure 1 must contain some details of the broken lines described on p3. It is not acceptable to have figures with important features that are not described in the caption.*

**Reply from authors**

We added some details on the dash lines into the caption in Fig.1 Black, red, blue dash lines indicate locations of sea-ice breakage in November, 2013 and on 10-14 February, 2014, and 1 March, 2014, respectively.

All editorial comments were addressed in the revised manuscript.

---

## Author Comment (AC2) · 28 Feb 2017

**Reply to Referee #2 (Dr. Dominé)**

We would like to thank your helpful comments to improve our manuscript. We responded to the general and specific comments. All comment are addressed in the revised manuscript. The updated sentences by your comments indicates blue words.

*Comment from Referee*

*The somewhat tedious point-by-point description of results should be completely replaced with a discussion focused on solving a few selected scientific questions such as for example: "how does the presence of FF affect aerosol composition?" or "what are the processes leading to halogen enrichment in FF and surface brine?". More aggressive attempts to make deductions from the observations are mandatory.*

**Reply from authors**

Thank you for your suggestions. We focused on sea-salt cycles in seasonal sea-ice area including sea-salt fractionation, aging processes of frost flowers, the fractionated sea-salt aerosols, and release processes of sea-salt aerosols from sea-ice surface on basis of our field evidences and previous works.

*Comment from Referee*

*The authors may cross their data with GOME2 BrO data to perhaps reach some interpretation on halogen activation.*

**Reply from authors**

We agree with your suggestions. However, our measurement periods were too early to measure BrO and IO densities around northern areas of Greenland using satellite, because of lower solar angle. Thus, we did (could) not compare to BrO and IO data in the manuscript.

*Comment from Referee*

*1- Select a couple of novel scientific questions to be addressed by the data set. and 2- Select the data to be presented to address the selected questions. A couple of case studies focused on a few events may be interesting.*

**Reply from authors**

As mentioned above, we focused on sea-salt cycles in seasonal sea-ice area including

sea-salt fractionation, aging processes of frost flowers, the fractionated sea-salt aerosols, and release processes of sea-salt aerosols from sea-ice surface on basis of our field evidences and previous works.

**Comment from Referee**

3- Separate results and discussion and write in a much more concise form to produce a much shorter paper. ,

4- Reach some strong and novel conclusion. For example, finding out that the presence of FF does not significantly affect aerosol composition would be quite interesting. and

5- Place the data not used here but of potential interest to others in supplementary material or any other accessible place. The authors should feel free to adopt any other strategy, the objective being to make a good and concise use of the data to derive strong conclusions. At present, the manuscript is more a detailed preliminary campaign report than an actual scientific paper.

**Reply from authors**

We separated Results and Discussion in the revised manuscript. Also we made an effort to concise form. Actually, we removed the description of sea-salt modification to "Supplementary" and remove some repetition of the statements and discussion. Additionally, we added our proposal (hypothesis) on sea-salt cycles in seasonal sea-ice area in "Concluding remarks". Some data and descriptions (e.g., sea-salt concentrations in snow) were added into the section of Results and Discussion in the revised manuscript, because of comments from Referee#3.

**Comment from Referee**

Vapor is supplied TO the atmosphere, not FROM. See the references on the same line. 2, 31. Specific surface areas are now expressed in $m^2 \, kg^{-1}$. Please convert.

**Reply from authors**

These editorial points were addressed in the revised manuscript.

**Comment from Referee**

Section 2.2. Was snow present in FF and brine samples? This should be mentioned as it dilutes the samples. Re. section 3.4.

**Reply from authors**

Yes. During the campaign, snowfall and blowing snow occurred. On some frost flowers at Site I and II, snow was present slightly present to extent to that fine structure of frost flower was identified clearly. This statement was wadded to "2.2 Sampling of frost flowers, brine, snow, and seawater".

*Comment from Referee*

*Please add a + sign: +1.8 °C, to avoid any ambiguity.*

*Throughout: replace liberated with released*

*Replace correlation with determination.*

**Reply from authors**

We added "+" sign in +1.8 °C. Words of "liberated" and "liberation" replaced to "released" and "release", respectively in the revised manuscript.

*Comment from Referee*

*The structure of the paper is such that the mention of solar radiation here is a bit odd and unexpected, and maybe not readily understood by all. Separating results and discussion would have helped.*

**Reply from authors**

With modification of structure in the manuscript, results and discussion were separated in the revised manuscript, as already noted.

*Comment from Referee*

*A more in-depth discussion of the causes of Br and I enrichment is in order.*

**Reply from authors**

All description was moved to section of "4-1. Sea-salt fractionation on sea-ice". Also, we added plausible processes for enrichment of $Br^-$ and I to the section. Then, we discussed each process, as follows.

Therefore, sea-salt fractionation by hydrohalite might be promoted in some samples of brine and frost flowers. Similar to $Mg^{2+}$, $K^+$, $Ca^{2+}$, and $Cl^-$, $Br^-$ and I can be enriched in frost flowers and brine by precipitation of mirabilite and hydrohalite. The plausible processes for enrichment of $Br^-$ and I can be listed as follows; (1) sea-salt fractionation

by precipitation of salts containing $Mg^{2+}$ or $Cl^-$ and (2) surface enrichment of $Br^-$ and I in liquid phase (e.g., brine). $MgCl_2$ and $MgSO_4$ were identified in aerosol particles as shown in Figs. 8 and S3. Considering presence of $MgCl_2$ and $MgSO_4$ in aerosol particles, Mg salts might be localized or precipitated in frost flowers and slush layer. According to previous laboratory and model studies (e.g., Mairon et al., 1999), $MgCl_2$ $6H_2O$ and KCl (sylvite) can be precipitated approximately at -36 °C and -34 °C, respectively. During the measurements, minimum air temperature (-34.1 °C) and temperature at surface of slush layer ($T_{FF}$) were higher than temperature at $MgCl_2$ $6H_2O$ precipitation. Therefore, $MgCl_2 \bullet 6H_2O$ precipitation might not occur during the measurements, although precipitation of mirabilite and hydrohalite can occur. If strong vertical gradient of temperature near surface engender that temperature at brine surface or around top of frost flower dropped to temperature at precipitation of sylvite and $MgCl_2$ $6H_2O$, these salts can be precipitated. However, temperature at brine surface or around top of frost flower might be higher than temperature for precipitation of sylvite and $MgCl_2$ $6H_2O$ during the measurements. Thus, precipitation of sylvite and $MgCl_2$ $6H_2O$ might not occur near surface of brine on sea-ice. Consequently, enrichment of $Br^-$ and I in frost flowers might derive from sea-salt fractionation to a greater degree than precipitation of mirabilite and hydrohalite. In addition to sea-salt fractionation by precipitation of mirabilite, hydrohalite, $MgCl_2$ $6H_2O$ and sylvite, a molecular dynamics (MD) simulation conducted by Jungwirth and Tobias (2001) predicted considerable surface enhancement of $Br^-$ and I in alkaline halide solutions. If $Br^-$ and I prefer to be enhanced also at the brine surface, enrichment of $Br^-$ and I might proceed in frost flower and brine by this surface enhancement and sea-salt fractionation.

These descriptions were added to "4-1. Sea-salt fractionation on sea-ice".

**Comment from Referee**

*Air T is a useful variable for many purposes, but the actual variable of interest here is surface T. All the speculation between air T and processes is really not useful, unless a surface T can be produced. Several lengthy discussions could just be removed.*

**Reply from authors**

Although we showed air temperature measured by AWS ($T_{AWS}$), our discussion was based on $T_{air}$ (temperature above 10 cm from sea-ice surface) and $T_{FF}$ (temperature at base of frost flowers) in the revised manuscript. We updated description in discussion

using air temperature in the sections of 4.1 and 4.2. Some sentences were removed from discussion in the revised manuscript.

**Comment from Referee**

*Replace larger by greater. The sea ice thickness may be important for the relationship between air and surface T, but not for surface processes.*

**Reply from authors**

We replaced "larger" to "greater" in the revised manuscript. Also, we rearrange the relation between $T_{FF}$ and sea-ice thickness in "Concluding remarks".

**Comment from Referee**

*10, 4-5. This statement does not lead to any useful conclusion. Please delete.*

**Reply from authors**

We removed this statement from the manuscript.

**Comment from Referee**

*Since or until ?*

**Reply from authors**

"Since" is correct.

**Comment from Referee**

*What useful conclusion do we derive from these Mg-rich and K-rich particles? Data description just is not enough for a scientific paper.*

**Reply from authors**

Presence of Mg-rich and K-rich sea-salt particles was direct evidence that the fractionated sea-salt particles were released from sea-ice area. We discussed more details about presence of these particles in the atmosphere, sea-salt fractionation, and release of the fractionated sea-salt particles to the atmosphere in the revised manuscript. The following discussion was added to "4-1. Sea-salt fractionation on sea-ice".

Mg was enriched in sea-salt particles collected in this study. The following evidences

are important to discuss the origins of Mg-rich sea-salt particles and Mg-rich salt particles in the atmosphere; (1) presence of highly Mg-rich particles (Mg-rich sea-salts, $MgCl_2$, and $MgSO_4$), (2)$T_{FF}$ lower than temperature at precipitation of mirabilite and hydrohalite, (3) higher Mg/Na ratio in fine mode, and (4) small variability of Mg/Na ratio in strong winds and blowing snow. Because Mg-rich sea-salts and Mg-salts cannot be evaporated and vaporized under the ambient conditions, these particles must be released through physical processes. If sea-salt particles were fractured in the atmosphere, sea-salt fractionation can occur. However, direct evidence of fracture of sea-salt particles in the atmosphere has not been obtained (Lewis and Schwartz, 2004). With sea-salt fractionation in brine and frost flowers, sea-salt particles released from sea-ice had different sea-salt ratios from those of seawater, as discussed above. Actually, $Mg^{2+}$, $K^+$ and $Ca^{2+}$ were enriched in frost flowers. Therefore, Mg-rich sea-salt particles (Fig.8b), K-rich sea-salt particles (Fig.8c), Mg-salt particles (Figs.8h-i), and K-salt particles (Fig.8j) might be originated from the sea-ice area and that they are associated with sea-salt fractionation.

**Comment from Referee**

*Could you please discuss the presence of non-sea salt sulphate?*

**Reply from authors**

Presence of nss-sulphate particles and some explanation were added to "3.7 Abundance of sea-salt particles and sea-salt-related particles". Presence of nss-sulphates in sea-salt particles were already discussed in discussion of sea-salt modification, which we remove to Supplementary. Short discussion was added to "4-3. Fractionated sea-salt particles in the atmosphere", because we focused on sea-salt cycles in the seasonal sea-ice areas in this study as follows.

Although the high aerosol number concentrations were observed occasionally at Siorapaluk under calm winds, the features might be caused by transport of (1) sea-salt particles released elsewhere by strong winds and (2) anthropogenic aerosols (i.e., sulphates and Arctic haze). Because of high abundance of sea-salt particles, most cases of higher aerosol number concentrations in calm winds were associated likely with release and transport of sea-salt particles. Similar phenomena were identified also in the Antarctic coasts (Hara et al., 2010).

**Comment from Referee**

*This is where your impressive data set could be put to good use to address these points. "At the moment, release processes of mirabilite-like and ikaite-like particles from the sea-ice surface without frost flowers remain unknown". Sure, but is not this campaign supposed to contribute to solving this problem?*

**Reply from authors**

We consider that presence of ikaite-like and mirabilite-like particles at Site III is an important evidence of sea-salt fractionation on sea-ice area and release of sea-salt particles from sea-ice. On basis of the evidence, we added more discussion about sea-salt fractionation near new sea-ice (Site III) to "4-3. Fractionated sea-salt particles in the atmosphere", as follows.

Mg was enriched in sea-salt particles collected in this study. The following evidences are important to discuss the origins of Mg-rich sea-salt particles and Mg-rich salt particles in the atmosphere; (1) presence of highly Mg-rich particles (Mg-rich sea-salts, $MgCl_2$, and $MgSO_4$), (2)$T_{FF}$ lower than temperature at precipitation of mirabilite and hydrohalite, (3) higher Mg/Na ratio in fine mode, and (4) small variability of Mg/Na ratio in strong winds and blowing snow. Because Mg-rich sea-salts and Mg-salts cannot be evaporated and vaporized under the ambient conditions, these particles must be released through physical processes. If sea-salt particles were fractured in the atmosphere, sea-salt fractionation can occur. However, direct evidence of fracture of sea-salt particles in the atmosphere has not been obtained (Lewis and Schwartz, 2004). With sea-salt fractionation in brine and frost flowers, sea-salt particles released from sea-ice had different sea-salt ratios from those of seawater, as discussed above. Actually, $Mg^{2+}$, $K^+$ and $Ca^{2+}$ were enriched in frost flowers. Therefore, Mg-rich sea-salt particles (Fig.8b), K-rich sea-salt particles (Fig.8c), Mg-salt particles (Figs.8h-i), and K-salt particles (Fig.8j) might be originated from the sea-ice area and that they are associated with sea-salt fractionation.

*Comment from Referee*

*"Therefore, most of the aerosol particles around $Na_2SO_4$ ratio in fine mode might be the modified sea-salt particles by heterogeneous reactions with nss-$SO_4^{2-}$." Can't you get to a stronger statement than just "might" by more in-depth examination of your data?*

*"Therefore, sea-salt modification (Cl loss) might be most likely to occur in fine mode." Sure, you may even use your data quantitatively and examine the role of aerosol*

*surface to volume ratio on reaction kinetics. Again, "might" is not sufficient here.*

**Reply from authors**

We agree with your suggestions. We change "might" to "may" in the revised manuscript. Because we focused on sea-salt fractionation, description of sea-salt modification was moved to "Supplementary". Role of aerosol surface area to volume ratio in sea-salt modification was already discussed in our previous studies (Hara et al., 2003, 2005, 2013). Thus, we mentioned this in Supplementary of the revised manuscript.

*Comment from Referee*

*Again, a more in-depth use of the data should allow useful conclusions, not vague suppositions.*

*These mentions of just observations, without any scientific deductions, are very disappointing.*

**Reply from authors**

Using our field evidences and results by previous works, we proposed sea-salt cycles in the seasonal sea-ice area in "Concluding remarks". Because this proposal is hypothesis, this includes some speculation. We believe that our proposal (hypothesis) is useful and new set for sea-salt chemistry in the polar regions.

---

## Author Comment (AC3) · 28 Feb 2017

**Reply to Referee #3**

We would like to thank your helpful comments to improve our manuscript. We responded to the general and specific comments. All comment are addressed in the revised manuscript. We separated Results and Discussion on basis of comment from Referee #2 (Dr. Dominé).

*Comment from Referee*

*The manuscript lacks an error analysis that would allow one to determine the significance of results. Words like "significant" are used, but that is not clearly related to a rigorous statistical definition or simply a qualitative word. To make statements about composition being different from sea water ratios, the authors would need to discuss error analysis more rigorously.*

**Reply from authors**

Thank you for your helpful suggestion. We agree with your suggestion. We estimated analytical errors in each analytical method from reproducibility of determination of standard solutions with the concentrations with similar to the field samples. Error bars were added to Figures 4-6 and also plots in Supplementary. The description about the estimation of analytical errors was added to "2.4.1 Analysis of frost flower, brine, snow, and seawater".

*Comment from Referee*

*The manuscript contains many technical errors and is difficult to read due to nonstandard use of English.*

**Reply from authors**

We addressed editorial and technical errors in the revised manuscript. Our manuscript was checked by native English speaker (FASTEKJAPAN, URL http://www.fastekjapan.com/greetings.htm) before submission and revise.

*Comment from Referee*

*The use of ternary plots is also not clear. Normally ternary plots are useful when the sum of the three components is 100%. However, in Figure 10, Cl, S, and Na are plotted as atomic percentages. These samples also have other atoms in their composition (e.g. oxygen that is a part of $SO_4$, Mg, etc.), so I guess that the plots show atomic percentage*

*of Cl, S, and Na atoms?*

**Reply from authors**

As you pointed out, sum of atomic ratios of Na, S, and Mg (or Na, S, and Cl) was not 100 % in the most cases. Thus, we converted sum of the atomic ratios (e.g., Na-S-Mg) to 100% for ternary plot. This data conversion is common procedure for the ternary plot. These descriptions were added to "3.8 Sea-salt fractionation of aerosol particles in coarse and fine modes".

*Comment from Referee*

*Some points are then very strange, such as on Figure 10b, three course points have >80% Cl and little Na. Something must charge balance the Cl⁻, but that is not clear on the plot.*

**Reply from authors**

This was already mentioned in description in sea-salt modification in the manuscript, as follows.

Some sea-salt particles had lower Na ratio and higher Cl ratio than the bulk seawater ratio. Mg was enriched considerably in the aerosol particles with lower Na ratios. Details of Mg enrichment are discussed ...

In the revised manuscript, the section about sea-salt modification was moved to Supplementary, because of suggestion by Referee #2 (Dr. Dominé).

*Comment from Referee*

*The paper often describes frost flowers as "fragile", while laboratory studies of frost flowers in a wind tunnel failed to produce aerosol, and field studies often show frost flowers get buried under blowing snow (e.g. snow blows while frost flowers remain intact). Therefore, there is not clarity in the literature that frost flowers are the only source of sea salt aerosol, and to the contrary, blowing snow and/or aerosol production from open water are often discussed in the literature. This manuscript doesn't describe the chemical composition of snow, which could be relevant to the production of aerosol, nor does it consider nearby open water and potential of aerosol production from that source.*

**Reply from authors**

Thank you for your comments. We agree with you. We removed the word of "fragile" from the statements in the revised manuscript. In this study, we samples surface snow on sea-ice in this study, although surface snow was present patchily and snow sampling sites were several meters away from the sites of sampling of frost flowers and brine at Site I and II. Snow sampling procedures were added to "2.2 Sampling of frost flowers, brine, snow, and seawater", as follows.

Snow on sea-ice was also taken using a clean stainless steel shovel from the location with snow accumulation (< 3 cm depth) without frost flowers at Site I, and II. Snow sampling was made approximately several –ten meters away from the site of sampling of frost flowers and brine. The pieces of frost flowers, brine (slush) , and snow samples, and snow were moved into each polyethylene bag (Whirl-pak; Nasco).

The data and description of surface snow were added to "3.3 Concentrations of sea salts in frost flowers, brine, and snow on sea ice" in the revised paper as follows.

By contrast, $Na^+$ concentration in snow samples collected on sea-ice was $1 - 2$ orders lower than that of seawater. $Na^+$ concentration of fresh snow on sea-ice was lower than 0.1 mmol $L^{-1}$. $Na^+$ concentration ranged in $0.4 - 3.2$ mmol $L^{-1}$ in the aged snow on sea-ice.

Snow:

$[Cl^-] = 1.315 [Na^+] + 0.02$ ($R^2 = 0.972$)

$[Mg^{2+}] = 0.035 [Na^+] + 0.02$ ($R^2 = 0.751$)

$[K^+] = 0.024 [Na^+] - 0.002$ ($R^2 = 0.997$)

$[Ca^{2+}] = 0.026 [Na^+] + 2 \times 10^{-5}$ ($R^2 = 0.994$)

In this study, we did not determine the concentrations of $Br^-$ and I in the snow samples. (There are some descriptions)

Similar to the slopes of frost flower, slopes of $K^+$-$Na^+$, and $Ca^{2+}$-$Na^+$ in snow on sea-ice were higher than seawater ratios. Slope of $Mg^{2+}$-$Na^+$, however, was lower than the seawater ratio, although fresh snow samples with the $Na^+$ concentration lower than 0.1 mmol $L^{-1}$ were distributed on the seawater ratios.

Also, we added discussion and explanation about release of sea-salt aerosols from sea-ice, snow, and open sea surface in discussion, as follows.

In "4-3. Fractionated sea-salt particles in the atmosphere":

The atomic ratios of Na and S of the Mg-poor particles imply strongly that the particles were in the form of $Na_2SO_4$. If the sea-salt particles were modified with $SO_4^{2-}$ by heterogeneous reactions, then the modified sea-salt particles contained sea-salt Mg. Thus, the presence of $Na_2SO_4$ particles cannot be explained by their release from the sea surface.

Mg/Na ratios in sea-salt particles varied greatly depending on sampling site and meteorological conditions (e.g., winds and temperature) as shown in Fig. 11. Variations of Mg/Na ratios in sea-salt particles are very interesting to understand release processes of sea-salt particles from sea-ice surface. It should be noted that sea-salt particles at Site IIIa and IIIb were distributed around seawater ratios. Hence, sea-salt particles except mirabilite-like and ikaite-like particles at Site IIIa and IIIb might be released from sea surface, which were present off Site IIIa and IIIb on 2-3 March.

Additionally, we proposed sea-salt cycles in the seasonal sea ice areas from our field evidences and results by previous works. Schematic figure (Fig. 12) was added in the revised manuscript.

In "Concluding remarks":

*First stage: seawater freezing*

Seawater starts freezing with decrease of air temperature. In this stage, sea-ice was present as conditions of grease-ice, frazil ice, and sludge. Considering that sea-salt particles with ratios similar to seawater were present only at Site IIIa and IIIb, these particles must be released from sea-surface in the initial stage and first stage. Depending on temperature at surface of sea-ice, ikaite starts precipitation, because temperature of ikaite precipitation is higher than that of mirabilite.

*Second stage: sea-ice formation and sea-salt fractionation*

Then, sea-surface was covered with thin sea-ice like Site IIIa and IIIb. Presence of sea-ice prevents release of sea-salt particles from sea surface to the atmosphere. Strong vertical gradient of air temperature near sea-ice surface resulted in frost flower formation. Some brine is migrated on frost flowers. Cooling of surface of frost flowers and brine can engender precipitation of ikaite and mirabilite. Ikaite-like and

mirabilite particles are released from frost flower and brine on sea-ice into the atmosphere. Ikaite-like particles and mirabilite-like particles are released into the atmosphere.

**Third stage: Frost flower growth and sea-salt fractionation**

With sea-ice growth, temperature on sea-ice ($T_{FF}$) decrease gradually by reduction of heat conduction from seawater to sea-ice surface. Lower temperature on and in slush layer induce sea-salt fractionation by precipitation of mirabilite and hydrohalite. Sea-salt enrichment (e.g., $Mg^{2+}$, $K^+$, $Ca^{2+}$, $Br^-$ and iodine) proceed gradually in the residual brine by the Na-salt precipitation. The residual brine having Mg enrichment is migrated vertically on frost flowers. If air temperature at top of frost flowers decrease approximately to -33 °C by the vertical gradient, sylvite and $MgCl_2$ can be precipitated. Mg-rich sea-salt particles and Mg-salts are released from the brine and surface snow. Iodine is released into the atmosphere through heterogeneous reactions and $Br^-$ were released slightly or insignificantly under the dusk conditions.

**Forth stage: Strong winds and snowfall on Frost flower**

Under the conditions with strong winds, snowfall and blowing snow, snow particles were attached on frost flowers and slush layer. As suggested by laboratory experiments (Roscoe et al., 2010), aerosol particles are released insignificantly from frost flowers. However, Mg-rich sea-salt particles and Mg-salts are released from the brine and surface snow by winds.

**Fifth stage: Frost flower covered with snow**

When snowfall and blowing snow are much enough to cover frost flowers and slush layer, frost flowers are buried completely in snow after the storm. Brine on sea-ice and frost flowers can be mixed into the snow layer. Sea-salt concentrations (salinity) in brine layer decrease gradually by migration and mixing into the snow layer. As a result, sea-ice surface (bottom of snow layer) freeze gradually. Sea-salts in the migrated brine and frost flowers can be redistributed through snow metamorphosis, although distributions of sea-salts might be heterogeneous in snow layer.

**Sixth stage: Snow erosion by strong winds**

Then, strong winds engender erosion of the snow layer. In other words, Mg-rich sea-salt particles are released into the atmosphere. Dry and hard surface of sea-ice appears after snow layer are removed completely.

*Comment from Referee*

*The manuscript claims in the abstract that "Aerosol number concentrations, particularly in coarse mode, were increased considerably by release from sea-ice surface under strong wind conditions." However, the figures and text really do not back up that claim. Figure 9 is presumably the data for this claim, but the authors do not indicate what periods to look at on that figure to see they effect they claim. In general, I see high coarse-mode aerosol on about DOY 12-18, 30, and 50-55. Those periods often have some winds, but not peak winds.*

**Reply from authors**

During occurrence of blowing snow and strong winds, aerosol number concentrations increased by release of sea-salt aerosols from sea-ice area. As you pointed out, some high aerosol number concentrations without blowing snow and strong winds were identified on DOY 12-18, 30, 50-55. In the Arctic regions, high aerosol number concentrations are associated with (1) release of sea-salt particles by the strong winds and (2) transport of anthropogenic aerosols such as sulphates (i.e., Arctic haze). In our observations, the high aerosol number concentrations corresponded to (1) appearance of low clouds (fog) above open sea surface off Siorapaluk, (2) high abundance of sea-salt particles in both coarse and fine modes. Therefore, the high aerosol concentrations might be associated with sea-salt release from sea-ice and sea surface before transport to the sampling sites. Indeed, similar phenomena were observed at Syowa Station, Antarctica (Hara et al., JGR, 2010). These results and discussion were added to results and discussion.

In "3.7 Abundance of sea-salt particles and sea-salt-related particles":
Appearance of low clouds (fog) above open sea was added to Figure 9 using red + symbols.

High abundance of sea-salt particles corresponded to strong winds, high aerosol number concentrations, and appearance of low clouds (fog) above open sea off Siorapaluk.

In "4-3. Fractionated sea-salt particles in the atmosphere":
High aerosol number concentrations and high abundance of sea-salt particles with

Mg enrichment under strong winds implies that sea-salt particles were dispersed from sea-ice surface. Although the high aerosol number concentrations were observed occasionally at Siorapaluk under calm winds, the features might be caused by transport of (1) sea-salt particles released elsewhere by strong winds and (2) anthropogenic aerosols (i.e., sulphates and Arctic haze). Because of high abundance of sea-salt particles, most cases of higher aerosol number concentrations in calm winds were associated likely with release and transport of sea-salt particles. Similar phenomena were identified also in the Antarctic coasts (Hara et al., 2010).

**Comment from Referee**

*The claim that bromide is being released from frost flowers made at the bottom of page 9 and page 10 would seem to imply a large release of bromine from frost flowers to the atmosphere. The authors should do a mass balance argument to indicate how much bromine would be released from this proposed release and compare the value to observations of atmospheric bromine (e.g. BrO). If that calculation led to unreasonably large BrO concentrations, then it would be evidence against this hypothesized direct halogen release. The lack of error analysis also makes it challenging to tell what is significant on these plots. Lastly, field evidence (Pratt et al., 2013) and multiple laboratory studies indicate that the pH of surfaces should be acidic for efficient halogen release, while highly saline samples were not efficient at releasing halogens.*

**Reply from authors**

On basis of your comments, we added the error bars to Figs. 4-6 and also plots in Supplementary in the revised manuscript. Considering analytical errors, $Br^-$ release from frost flowers might be slightly or insignificantly. Therefore, $Br^-$ release from frost flowers might be insignificant at Site I under the dusk conditions. On the other hands, the estimated $I^-/Cl^-$ ratio appeared to be released somewhat from frost flowers. From $I^-/Cl^-$ ratios, we estimated the released amount of iodine from frost flowers to ca. 16%. Therefore, we modified description about halogen release from frost flowers in "4-2. Aging of frost flower and sea-salt fractionation", as follows.

To elucidate halogen chemistry in frost flower, we compared between the features of $Br^-/Cl^-$ and $I^-/Cl^-$ in frost flowers at Site I. High correlation of $Cl^-$-$Na^+$ and $Cl^-$ enrichment strongly suggests that $Cl^-$ release was insignificant from frost flowers and brine. Thus, we focus on the features of $Br^-$ and $I^-$, here. we attempt to estimate the

molar ratios of $Br^-/Cl^-$ and $I/Cl^-$ in frost flowers using the ratios of $Mg^{2+}/Cl^-$, $Br^-/Cl^-$ and $I/Cl^-$ on 22–26 February, assuming that hydrohalite was not precipitated yet on 22–26 February, and that $Br^-$ and I did not liberate from frost flowers through heterogeneous reactions. When the molar ratios on 22–26 February changed by the assumptions above, the molar ratios of $Br^-/Cl^-$ and $I/Cl^-$ in frost flowers after hydrohalite precipitation were estimated respectively as 0.00214 and $1.82 \times 10^{-6}$. Although $Br^-/Cl^-$ ratio (0.00206) was slightly lower than the estimated ratio, this difference might be very slightly or insignificant. Therefore, $Br^-$ release from frost flowers might be insignificant at Site I under the dusk conditions. On the other hands, the estimated $I/Cl^-$ ratio was higher than the ratios in frost flower ($1.562 \times 10^{-6}$) on 27 February. The following likelihood should be considered: (1) reduction of I enrichment by precipitation of salts containing iodine, and (2) I release from frost flowers through heterogeneous reactions. Although $NaBr \bullet 5H_2O$ can be precipitated at -28 °C (Koop et al., 2000), no report in the relevant literature discusses a study of precipitation of iodine salts in sea salts. If iodine salts were not precipitated under the conditions at Site I, this difference implies the likelihood that I was released from frost flowers. Iodine can be released from frost flowers through the following heterogeneous reactions (Thompson and Zafiriou, 1983; Carpenter, 2003; Simpson et al., 2007; Saiz-Lopez et al., 2015).

$$HOI + Br^- + H^+ \rightarrow IBr + H_2O \qquad \text{(R1)}$$
$$I^- + H_2O_2 \rightarrow HOI + OH^- \qquad \text{(R2)}$$
$$HOI + I^- + H^+ \rightarrow I_2 + H_2O \qquad \text{(R3)}$$
$$I^- + O_3 + H_2O \rightarrow HOI + O_2 + OH \qquad \text{(R4)}$$

Reactions of R4 can proceed under nighttime conditions. Other reactions, however, are enhanced under conditions with solar radiation because HOI can be formed efficiently through atmospheric photochemical reactions. Frost flowers at Site I had been exposed to direct solar radiation since 18 February, 2014, although it had been dusk at noon since early February. Therefore, the heterogeneous iodine loss from frost flowers can engender reduction of I enrichment after hydrohalite precipitation in frost flowers. From the $I/Cl^-$ ratios, amount of the released iodine from frost flowers can be estimated to ca. 16 % under our research conditions. The more solar radiation, the more reactive halogens might be released from frost flowers and brines. Comparing between the short-term features of $Br^-/Cl^-$ and $I/Cl^-$ in frost flowers, heterogeneous I loss appears likely to occur from frost flowers relative to heterogeneous $Br^-$ loss.

*The referencing of the paper is not accurate. An example of this is on line 3-4 of page 3, which the authors say "Reportedly, Br⁻ enrichment occurs in frost flowers at Barrow, Alaska (Douglas et al., 2012)...". However, the text of that citation says "There is no enhancement in bromide to chloride ratios in the frost flowers compared to brine or seawater".*

**Reply from authors**

Douglas et al. (2012) estimated $Br^-/Cl^-$ ratios in frost flowers and brine. Then, they evaluated $Br^-$ enrichment from the ratios. However, we calculated molar ratios of $Br^-/Na^+$ from the data listed in Table 1 of Douglas et al. (2012). $Br^-$ appeared to be enriched relative to $Na^+$ slightly in some frost flower samples, although many frost flowers had insignificant $Br^-$. Therefore, we addressed the statements in "Introduction", as follows.

Reportedly, $Br^-$ enrichment occurs slightly in frost flowers in the Weddell Sea, Antarctica (Rankin et al., 2002). Slight $Br^-$ enrichment to $Na^+$ was observed in a few samples collected at Barrow, Alaska, although there was no $Br^-$ enrichment in same samples of frost flowers and brine (Douglas et al., 2012). Additionally, results of some earlier studies have indicated non-significant $Br^-$ enrichment in frost flowers at Barrow and Hudson Bay (Alvarez-Aviles et al., 2008; Obbard et al., 2009).

---

## Author Response (AR2)

**Reply to Referee #1 (Dr. Roscoe)**

We would like to thank your helpful comments to improve our manuscript. All comments are responded and addressed in the current revise. The updated parts are indicated by red words in the revised manuscript (pdf file).

*Comment from Referee:* *Section 5 and Figure 11 present an excellent new speculative scientific conclusion, but there is no demonstration of how either relates to the discussion in Section 4. Also, the verbiage that is the revised Section 4 is so confused that it cannot yet be used to prove even elements of the scenario in Section 5 and Figure 11, let alone all of them.*

**Reply from authors:** We arranged descriptions in Section 4 (discussion) on basis of interpretation of "what the field measurements meant". In addition, we divide each process between shown and speculated from field evidences in Figure 12.

Details of our reply to each comment are as follows.

*Comment from Referee:* The discussions in the sub-sections of Section 4 are now hopelessly lengthy, tangled and confused. Section 4 needs a major rethink, and needs to be shortened.

**Reply from authors:** We shortened and rearranged the discussion section. In section 4, we attempted to focus on sea-salt fractionation on sea-ice and sea-salt cycles which can be shown statistically. Also, statistical analysis (t-test) was done in Section 3. Schematics about sea-salt cycles (Fig. 12) and descriptions moved from "conclusion" to Section 4 by suggestion from the referee #2.

*Comment from Referee:* *The points in Section 5 need to relate, in a very concise way, to the results of the discussion in Section 4. The authors admit that some of Section 5 is speculative, and this is no bad thing for such a comprehensive scenario, but Section 5 should identify (concisely, please) which elements are speculation, which follow from the work of others, and which follow from Section 4 and where in Section 4.*

**Reply from authors:** Schematics about sea-salt cycles (Fig. 12) were redrawn in Fig. 12

of the current revise. Highly speculated processes were removed from Fig. 12. Processes including some speculation were shown by dotted arrows with "?" marks, whereas processes shown from the field evidences were shown by thick arrows.

*Comment from Referee:* *The early part of Section 4.3 focuses on frost flowers as the source of particles. But the correlation between high particle density and strong winds, brought up later in Section 4.3, suggests that the earlier discussion in 4.3 is irrelevant. This earlier discussion should be discarded, or at least relegated to the end of the Section 4.3 and downplayed.*

**Reply from authors:** The description in early part of Section 4-3 was removed in the current revise.

*Comment from Referee:* Page 1 line 24 wrongly states that frost flowers are ice crystals that contain salts. In fact, they grow perfectly well on fresh-water ice. Perhaps the authors meant "In the sea-ice zone, frost flowers are usually ice crystals that contain salts ..." ?

**Reply from authors:** The sentence changed to "Frost flowers on sea-ice are ice crystals that contain brine and sea salts".

*Comment from Referee:* *Page 14 line 39 makes an incorrect claim about the frost-flower paper that I led in 2010. Our conclusion was that no particles were released from frost flowers, not that they were released non-significantly.*

**Reply from authors:** The statement was changed to "As suggested by laboratory experiments (Roscoe et al., 2010), no aerosol particles are released from frost flowers.".

*Comment from Referee:*  *The material added in the revisions contains a large number of errors of grammar and syntax, even before Section 4. The revised sentences should be re-read carefully, and corrected - they have the hallmarks of being inserted in haste, and of not being read by all the co-authors.*

**Reply from authors:** We, all coauthors, checked the current revise. Additionally, the manuscript was checked by native English speaker (FASTEK, http://www.fastekjapan.com/). Our manuscript was written and checked in US English.

**Reply to Referee #2**

We would like to thank your helpful comments to improve our manuscript. All comments are responded and addressed in the current revise. The updated parts are indicated by red words in the revised manuscript (pdf file).

*Comment from Referee:* *While some effects are large (e.g. sulfate is depleted by a large factor (~5) compared to sea water ratios), many of them are small. Therefore, it is necessary to give a well described error analysis that accurately uses appropriate words. Error estimates are included, but for example in Figure 6, the error bars are not specified as 1-sigma or otherwise.*

**Reply from authors:** We used 1-sigma of reproducibility of determination in our analytical conditions as analytical errors in text and error bars in Figures. The descriptions were added to the text and figure captions. Furthermore, statements with strong wording (e.g., remarkably and markedly) were changed to "weaker wording".

*Comment from Referee:* *The analyses of sea water (discussed further below) are not compared to literature values, and a quick comparison seems to show deviations of 10-20% from literature, which is on the order of potential analysis error, particularly when propagated into ratios.*

**Reply from authors:** As suggested by you (Referee #2), we compared seawater ratios in this study (seawater at Siorapaluk) and literatures (Lide, 2005; Millero et al., 2008; Millero, 2016). Although the molar ratios of seawater collected at Siorapaluk differed slightly in some species from the literature values (Lide, 2005; Millero et al., 2008; Millero, 2016), this difference was larger than analytical errors. Because seawater ratios were different at sampling sites (Millero, 2016), the difference might result from locality of seawater ratios. These statements were added into P.6 L. 11-19 in the current revise.

*Comment from Referee:* *For the weaker effects, a rigorous discussion of analytical errors is needed. A t-test could be used, but often errors are not truly normally distributed (they are non-Gaussian), so effects that are indicated to be "significant" in a statistical senses but are not far from the level of significance are not to be over interpreted.*

**Reply from authors:** We agree with your comment. T-test was applied for the molar ratios of frost flowers, brine, and snow. Results of t-test (t-values, p-values, and degree of

freedom) were shown in Table 2 in the current revise. Description in Section 4 were rearranged on basis of the results of t-test. Therefore, large parts of discussion about Br- and I were removed from the text.

*Comment from Referee: The idea that blowing snow may be a source of sea salt aerosol is still not truly considered in this manuscript. Instead the authors stay with the idea of the prior manuscript that sea salt aerosol essentially come from frost flowers. The manuscript appears to show photographic evidence for blowing snow. Figure 3, panel f shows the "condition of old sea ice on 2 March, immediately after the storm", which is clearly scoured of snow. Photographs on earlier days (Figure 3, panels (a), (c), and (d)) clearly show snow on the sea ice. Apparently this snow was blown away, and as snow blows, it sublimes, producing aerosol particles. Open sea water can also lead to production of sea salt aerosol.*

**Reply from authors:** As shown in Fig. 12, we proposed sea-salt emission by blowing snow. At Sites I and II with slush layer, snow layer was not blown away completely in contrast to surface condition on old and very-old sea-ice after storm conditions. Because slush layer and surface snow on new and young sea-ice were wet, snow erosion might be less efficient on new and young sea-ice with slush layer than on old and very-old sea-ice. Of course, sea-salt particles can be released from surface snow on new and young sea-ice through snow erosion. This difference is likely important to elucidate sea-salt cycles on seasonal sea-ice areas. Some descriptions about sea-ice/snow conditions were added into Sections of 3.2, 4.3, and 4.4 in the current revise. Emission of sea-salt particles from open sea water was identified dominantly at Sites IIIa and IIIb in this study. The statements were shown in Sections of 4.3 and 4.4.

*Comment from Referee: Figure (12), has some good ideas in it, but does not belong in the conclusions (it is a discussion point and some of the text related to it are not reasonable conclusions of the present study). Some aspects of this discussion that are taken too far by the authors include the following. In the "Initial stage", bubble bursting releases particles from the surface microlayer of the ocean. This microlayer is organic-rich, and could potentially include inorganic counter ions selected by the organic species. In the second stage, it is posited that "ikaite-like and mirability particles are released from frost flowers and brine on sea ice into the atmosphere". What physical release mechanism is being proposed here? For the frost flowers to be depleted in sulfate with respect to chloride, there must be a physical separation of brine (depleted in sulfate) from*

*mirabalite crystals. The normally discussed mechanism is that the mirabilite is left in the brine on the sea ice, and the sulfate-depleted brine migrates up the frost flower. That would leave the mirabilite further from the atmosphere and seems to not be compatible with production of mirabilite particles in the atmosphere. The text needs to posit a mechanism for this effect, and it should be discussed, as the results of the manuscript don't appear to prove such a mechanism exists. The "third stage" indicates that it is concluded that "iodine is released into the atmosphere through heterogeneous reactions and Br- is released slightly or non-significantly under dusk conditions. The more solar radiation, the more reactive halogens can be released from frost flowers and brines." This statement is way too far for the results shown. Remember that the results show slight enhancements in Br- and I- in the older frost flowers. The argument for I- release is that I- is less enhanced than Br-, but again these enhancements are small and analytically suspect in themselves, their differences are even smaller. Lastly, how can the study that says bromide is not released conclude that with more solar radiation there is release of bromide? In the fourth stage, the lack of release of particles by wind from frost flowers is discussed to somehow conclude that Mg-enriched particles are released by wind?*

**Reply from authors:** As suggested from you (referee #2), Fig. 12 and the statements moved from "Concluding remarks" to Section 4.4 (discussion).

In your comment on "Initial stage", microlayer and organics were pointed out. It is true that presence of microlayer and organics have some potentials to modify constituents of sea-salt particles released from sea surface, but Keene et al. (2010) showed sea-salt ratios in particles released from bubble bursting were similar to the seawater ratios. Because we did not analyze and discuss organics in this study, description about microlayer and organics was not added in the revised manuscript. Instead, early work by Keene et al. (2010) was cited in the text.

In Second stage, ikaite-like and mirabilite-like particles were present at Sites IIIa and IIIb. Because these particles cannot be vaporized in ambient conditions, these particles must be released through physical processes. Considering these particles were identified only at IIIa and IIIb, these particles might be released from fresh and new sea-ice area. As shown in our results and pointed by you, mirabilite might be precipitated on sea-ice and brine. Then the residual brine (i.e., sulfate-depleted brine) can be migrated onto frost flower at Sites I and II. However, $SO_4^{2-}/Na^+$ ratios at Sites IIIa and IIIb were higher than those at Sites I and II. This difference indicated that less-sulfate-depleted brine were migrated onto frost flowers at Sites IIIa and IIIb. Therefore, mirabilite might be distributed on both brine and frost flowers at new sea-ice area. However, specific release

processes of ikaite-like and mirabilite-like particles were still unknown. These statements were added to Sections 4.3 and 4.4.

Third stage: The molar ratios of $Br^-/Cl^-$ and $I/Cl^-$ in frost flower were mostly higher than those in brines, except for a few brine samples. It is expected that $Br^-$ and I were richer in frost flowers because of sea-salt fractionation. However, the differences of change of $Br^-/Cl^-$ and $I/Cl^-$ were not significant in t-test. Therefore, statements about likelihood of release of $Br^-$ and I were excluded in the current revise.

Fourth stage: Similar to ikaite-like and mirabilite-like particles, Mg-rich sea-salt particles and Mg-salt particles must be released though physical processes from frost flowers, brine, and snow. Considering the direct evidence of Mg depletion in aged surface snow on sea-ice, Mg-rich sea-salt particles and Mg-salt particles were likely released from surface snow mixed with the residual brine on sea-ice. The variations of Mg/Na ratios in sea-salt particles were smaller in both coarse and fine modes under storm conditions (DOY = 40 and 59), although the Mg/Na ratios were higher than the seawater ratio. Winds passed from the old and very-old sea-ice area to the sampling sites in the storm conditions. Consequently, Mg-rich sea-salt particles in the storms might be released also from the snow layer on old and very-old sea ice through erosion of snow by strong winds because the slush layer was absent on old and very-old sea ice. By contrast, Mg/Na ratios were varied largely under calm wind conditions. To explain the presence of highly Mg-rich sea-salt particles and Mg-rich salt particles, we inferred that these particles were released from the aged surface snow and the residual brine on slush layer and frost flowers through erosion of snow with the residual brine and splashing and shattering of the residual brine film. Higher Mg/Na ratios in fine sea-salt particles are eminently explainable if the processes proceeded on seasonal sea-ice areas. These statements were added to Sections 4.3 and 4.4.

***Comment from Referee:*** *The text has been modified, but the abstract has been modified very little. Line 18 of page 1 discussed heterogeneous $SO_4^{2-}$ formation that is not really discussed in the text anymore.*

**Reply from authors:** Although sea-salt modification was shown in supplementary information, the statement in abstract was changed to "Sulfate depletion by sea-salt fractionation was found to be slight in sea-salt aerosols because of the presence of non-sea-salt $SO_4^{2-}$".

***Comment from Referee:*** *Page 2, line 13: This says that sea salt is an ice nucleus (IN),*

*which seems surprising given that most IN are not soluble. Give a reference or cut this text. page 2, line 23: Again it is said that salt is an IN.*

**Reply from authors:** Some recent studies pointed out that sea-salts have potential to ice nuclei. Some references were added in the text.

*Comment from Referee: page 2, lines 32-38: This section is about fractionation in a section nominally about "modification". Move to the other section about fractionation. A clearer description of how "fractionation" differs from "modification" would also be useful.*

**Reply from authors:** We agree with your comment. These descriptions moved to section on "sea-salt fractionation". In this study, sea-salt fractionation and modification indicate, respectively, change of molar ratios in sea-salts through salt precipitation and heterogeneous reactions. The mention was shown in P.2 L1-2 and L31-32.

*Comment from Referee: page 3, section 2.1: Give a clearer definition of new, young, old, very old ice.*

**Reply from authors:** New, young, old, and very old sea-ice were defined by sea-ice age in this study. Some description was added in P.3 L. 12-18.

*Comment from Referee: page 5, line 7: There is a shift here from mass based concentrations to molar based ratios. Make that shift more clear in the text.*

**Reply from authors:** For more clear mention, we show "molar concentration" in the text (p.5 L.11).

*Comment from Referee:   The slopes of the correlations are unitless, but the intercept has the unit of the concentration, and thus is important. This intercept is often trivially small, but for two cases, the intercept may have an effect, specifically the Mg2+ correlation to Na+ in snow shows a very different slope than the other species, but also an intercept that is similar to actual snow Mg2+ snow concentrations. The brine Cl- to Na+ comparison may also be affected by this offset. It also needs to be noted that ion ratios (calculated in other places) are equivalent to slopes of correlation plots only when the intercept is zero. When the intercept is non-zero, the slope of a correlation plot will differ from the ratio of the ion concentrations. On page 6, line 40, the authors indicate*

*that ratios to Na+ are consistent with sea salt fractionation (by mirabalite precipitation), which they are, but these changes are relatively small because there is much more Na+ than SO4-- in sea water, so sulfate limits the removal of Na+. Therefore, what should be pointed out is that SO2-- is depleted compared to either Na+ or Cl-, which is clearer evidence of mirabilite precipitation. The magnitude of these changes appear consistent.*

**Reply from authors:** We agree with your comments. As pointed out from you, the slopes in the relations might be close to the ambient molar ratios, when the intercepts are close to zero. The slopes of the relations can be biased positively in cases of contamination/mixing of non-sea-salt species such as minerals and anthropogenic species, which can be deposited onto surfaces of frost flowers, brines, and snows. In contrast, the ratios can be biased negatively in cases of sea-salt fractionation on sea-ice and depletion/release of the continents in frost flowers, brines, and snows into the atmosphere. The molar ratios in frost flowers, brines, and snow are presented in Table 1. With the exception of $Mg^{2+}/Na^+$ in snow and $I/Na^+$ in frost flowers and brine, the molar ratios conform to the slopes. The intercept values and the coefficients of determination in these ratios are, respectively, larger and smaller than the other ratios. These descriptions were added to Section 3.3 (p.7 L.8-15). Also, explanation about mirabilite precipitation was added in Section 3.3 (p. 7, L.17-37).

**Comment from Referee:** *The treatment of sea water ratios and their errors is still lacking in this discussion. Specifically, Table 1 shows the ratio of Br- to Na+ and Cl- from literature (Lide, 2005). However, the analyses of sea water are used for the other ratios. When I look up sea water ratios to Na+, I find the following values, which are then compared to the analysis results from seawater (n=2) in Table 1, which is in parentheses following the literature values: K+/Na+ = 0.022 (0.020), Mg2+/Na+ = 0.113 (0.091), Ca2+/Na+ = 0.022 (0.020), Cl-/Na+ = 1.164 (1.227), and SO4--/Na+ = 0.060 (0.0613). These differences are on the order of 10-20%. They could be true differences between the sea water sampled as compared to standard sea water, or they could be small analytical errors. However, it is important that the present study used only a literature value for the Br-/Cl- ratio and then compares to analytical results to say that Br-/Cl- ratio is larger in frost flowers than in sea water. The magnitude of the enrichment is ~20%, which is comparable to the differences between the sampled sea water and literature sea water. Therefore, it is not clear to me that the "result" of bromide enrichment is a true result. Given the magnitude of errors, this result should be a minor one rather than a serious highlight of the results.*

**Reply from authors:** We compare sea-salt ratios in seawater at Siorapaluk to the literature values (Lide, 2005, Millero et al., 2008; Millero, 2016). As mentioned in above comment, sea-salt ratios differed even in respective literatures (Lide, 2005, Millero et al., 2008; Millero, 2016) because of locality shown by Millero (2016). Details were mentioned in P.6 L.11-19 in current revise. Because of the differences, we used seawater ratios at Siorapaluk in this study, although literature values were used for seawater ratios of $Br^-$ and $I^-$.

*Comment from Referee:* This discussion is interesting and clearly reiterates the point of sulfate being depleted in frost flowers but not in brine. The effects on other ions are relatively small, but the wording of the discussion does not reflect the small magnitude of these changes and still lacks a complete error discussion. For example, page 7 line 32 indicates that $Mg2+/Cl^-$, $K+/Cl^-$, and $Ca2+/Cl^-$ "increased remarkably". These increases are on the order of 10-20%, and not too different from reported error bar magnitudes. The wording needs to be more reflective of the actual magnitude of the effect. Additionally, the errors are indicated as "analytical errors" in the caption of Figure 6, but that is not a clear indication of the type of error. If these are 1-sigma error bars, that should be noted. The results section does not point out the large difference (factor of 5) present in sulfate ratios, but instead focuses on small changes that appear near error estimates. Additionally, in the start of this section (lines 17-18 of page 7), the text should more clearly make a correspondence between the terms "aged, young, fresh" frost flowers and the sampling sites.

**Reply from authors:** As mentioned above, analytical errors and error bars mean 1-sigma values of reproducibility of determination. The mention was added into the text and figure captions. To elucidate the sea-salt ratios changed by precipitation of mirabilite and hydrohalite, we estimated the sea-salt ratios in frost flowers, although ratios of $Mg^{2+}/Cl^-$, $K^+/Cl^-$, and $Ca^{2+}/Cl^-$ do not change by mirabilite precipitation. Procedures and assumption were added into Sections 3.3 (P. 7 L.24-37) and 3.4 (P.9 L.7 – p.10 L.5).

*Comment from Referee:* The observation of Mg2+ enhancement in aerosol particles is an interesting one. Figure 11 should include a sea water ratio line (0.113 from literature, but 0.091 from sea water analysis in this work). Given that Figure 11 shows no wind speed, it is very challenging to confirm the statement "In conditions of blowing or drifting snow and strong winds, the Mg/Na ratios and the standard deviations decreased in both

*modes...". This needs to be more clearly presented. The storm of 1 March is noted, but the figure shows day of year. The text should indicate the DOY of this storm.*

**Reply from authors:** Lines of seawater ratios and plot of wind speed were added into Fig. 11. Short explanations were added to the text in Section 3.9 (p.12 L.4-11). DOY of 1 March was also indicated in the text.

***Comment from Referee:*** *This section indicates that Mg/Na ratios decreased (towards sea salt ratios) during blowing snow and strong winds. However, page 13, line 20 indicates that Mg is enriched during strong winds. These two statements appear to conflict.*

**Reply from authors:** The description in Section 3.9 was modified to avoid confusion and miss-understanding of readers, as follows. In conditions with blowing snow or strong winds (>5 m s$^{-1}$), the Mg/Na ratios and their standard deviation tended to decrease in both modes (particularly in fine mode). For instance, median Mg/Na ratios in strong winds were ca. 0.18 in both modes on DOY = 40 (10 February), and ca. 0.16 in coarse mode and 0.22 in fine mode on DOY = 59 (1 March).

***Comment from Referee:*** *page 10, lines 12-36 are mostly speculation, and their only basis is results that are probably close to analytical error limits.*

**Reply from authors:** The ratios of Br$^-$/Cl$^-$ and I/Cl$^-$ in frost flowers were mostly higher than those in brines, except for a few brine samples. It is expected that Br$^-$ and I were richer in frost flowers because of sea-salt fractionation. However, t-test indicated that the difference between frost flowers and brines was insignificant. Most of discussion about enrichment of Br$^-$ and I were removed in the current revise, although short statement was shown in the text in Section 4.4 (P.12 L.28-37). In Section 4.1, we focused on occurrence of sea-salt fractionation by mirabilite precipitation on basis of the results of t-test.

***Comment from Referee:*** *page 11, line 11 overstates the "drastic change". Also, lower in this section, lines 23 and 24 indicate significant and non-significant changes that seem similar to the eye and have no statistical basis.*

**Reply from authors:** To elucidate change of molar ratios by precipitation of mirabilite and hydrohalite, we estimated the sea-salts. Simultaneous change of the ratios of

$Mg^{2+}/Cl^-$, $K^+/Cl^-$, $Ca^{2+}/Cl^-$, $Na^+/Cl^-$, $Br^-/Cl^-$, and $I/Cl^-$ implies that hydrohalite precipitation proceeded at Site I. Indeed, difference of the ratios between 24-27 February and 26-28 February was 2-3 times greater than analytical errors. Therefore, the differences might be attributed to sea-salt fractionation (hydrohalite precipitation). Discussion was modified in the text of P.13 L.16-30.

*Comment from Referee:* *page 11, line 36: I have no clue how these molar ratios were generated; please explain more, or more likely cut this section.*

**Reply from authors:** As mentioned in above comments, the procedures of the estimation were added to the statements in Sections 3.3 and 3.4.

*Comment from Referee:* *page 11, line 42: A tiny difference is being used to justify a broad statement of "likelihood that I was released". This is really wild speculation. Iodide is not even a conserved species in sea water, so drawing this line is not straightforward from the literature. This wild speculation continues through page 12, line 16.*

**Reply from authors:** This statement about I release were excluded from discussion because ambient $I/Cl^-$ ratios of seawater at Siorapaluk were uncertain in this study.

*Comment from Referee:* page 13, line 20: This statement conflicts with page 9, section 3.9.

**Reply from authors:** This comment was responded in above comment.

*Comment from Referee:* *page 14, lines 1-6 are truly conclusions from this work. Other true conclusions should be added to this section to make a new conclusions section. The reminder of the new "conclusions" should be moved to a discussion, as these points are really a discussion of potential ideas about sea salt aerosol formation related to sea ice. As discussed earlier, this discussion should be narrowed to what is defensible from the observations in this manuscript and/or points already in the literature (with citation of those literature sources). The current discussion lacks appropriate citation.*

**Reply from authors:** As suggested from you (referee #2), Fig. 12 and the statements moved from concluding remarks to Section 4.4 (discussion). Also citations were also added in each site of the text in Section 4.4. More conclusions were added into

"Concluding remarks".

---

## Author Response (AR3)

Reply to Co-editor, Dr. Sergey A. Nizkorodov,

We would like to thank you for your comments. Our manuscript was corrected on basis of your comments. Corrected parts were shown by red words in the text (please show the pdf file).

*Comments*: I encourage the authors to make writing more succinct, especially in the sections describing their observations.

*Reply:* We arranged the sections of "Sampling and analysis" and removed some sentences.

*Comments*: P1, L12 and L14: charge on the iodine ion is missing, perhaps you need to explain that you mean the sum of all forms of iodine (in the abstract and also in the analysis section)

*Reply:* We used ICP-MS for determination of iodine in frost flowers and brines. Because ICP-MS can provide only the elemental concentrations, iodine (or I) is correct here. In order to avoid misunderstanding of readers, we changed from "I" to "iodine" in abstract. In analytical section, we add short explanation about this reply.

*Comments*: Figure 4: I recommend starting all axes from 0 in the in Br- vs. Na+ plot, and I vs Na+ plot.

*Reply:* We modified the plots (starting all axes from 0) in Figures 4, S3 and S5.

*Comments*: P2, L24: "abilities of" -> "abilities to act as"

*Reply:* We updated the wording on basis of your comment.

*Comments*: P3, L12: "presents simultaneous observations of frost flowers, brine" -> "shows the locations where simultaneous observations of frost flowers, brine … were made"

*Reply:* We updated the wording on basis of your comment.

*Comments*: P4, L14: "sucking" -> "flow rate"

*Reply:* Because 1L is not flow rate but air volume. We arranged a few sentences in this explanation.

*Comments*: P5, L2: please verify that the dilution factor of 1E6 is not a typo, it appears to be too high

*Reply:* $10^6$ is not typo. Because concentrations in frost flowers and brine was higher than that in seawater, $10^6$-diluition was suitable for the analytical conditions in this study.

*Comments*: P5, L19: "in accordance with" -> "described by"

*Reply:* We updated the wording on basis of your comment.

*Comments*: P7, L6: "the Br- and I concentrations in the snow samples were not found." – "the Br- and I concentrations were below the limit of detection" (also correct the same P14, L2)

*Reply:* We checked the words. In this study, we did not determined Br- and I in snow samples. Therefore, we changed the sentence to "the Br- and I concentrations were not determined".

*Comments*: The discussion on pages 7 and 8 is difficult to follow. Is the main point of the discussion to prove that mirabilite precipitation was occurring? I think this point can be conveyed using much less text.

*Reply:* In page 7, we focused mainly on mirabilite precipitation and change of the ratios by mirabilite precipitation. In page 8, we attempt to show heterogeneous distribution of sea-salts in frost flowers, brines, snow. We arranged the discussion and added some comments to follow easily.

*Comments*: P10, L11: "had a liquid surface in the atmosphere. In other words, the particles were deliquescent in the atmosphere." -> "were deliquesced in the atmosphere"

*Reply:* This sentence, "In other words~in the atmosphere", was removed from the text, because of other comment (as mentioned below).

*Comments*: Figure 8: explain the meaning of stars in the figure caption

*Reply:* Explanation about asterisks was added in caption of Fig. 8.

*Comments*: P11,L17: "wholly Cl released sea-salt particles by SO42-" -> "sea-salt particles in which chloride was completely displaced by sulfate". Also please fix other awkward definitions of particle classes in this sentence.

*Reply:* We updated the wording on basis of your comment.

*Comments*: P13, L13: This explanation: "It is notewerthy that molar ratios in frost flowers cannot change …" is important for understanding the results of this paper and should appear earlier, perhaps in the introduction section. Otherwise readers will wonder where the precipitates go and will not find out until they get to the end of the paper.

*Reply:* Some explanation about locations where sea-salt precipitation is going on in was added in "Introduction".

*Comments*: P14, L16: This sentence "In other words …" repeats information in the previous sentence and is redundant (such redundancies actually happen in multiple places in the paper, I would try to eliminate them if possible).

**Reply:** We found the sentences using "In other words" in Page 10, 13, 14, and 16. The sentence in page 10 was removed. Because of importance, the sentences in Page 13 remained in the text. The sentences Pages 14 and 16 were arranged in the text.